# SDR enzymes oxidize specific lipidic alkynylcarbinols into cytotoxic protein-reactive species

Pascal Demange[1†], Etienne Joly[1†], Julien Marcoux[1†], Patrick RA Zanon[2,3], Dymytrii Listunov[4,5], Pauline Rullière[4], Cécile Barthes[5], Céline Noirot[6], Jean-Baptiste Izquierdo[1], Alexandrine Rozié[1,7], Karen Pradines[1,7], Romain Hee[1,7], Maria Vieira de Brito[5,8], Marlène Marcellin[1], Remy-Felix Serre[9], Olivier Bouchez[9], Odile Burlet-Schiltz[1], Maria Conceição Ferreira Oliveira[8], Stéphanie Ballereau[4], Vania Bernardes-Génisson[5], Valérie Maraval[5], Patrick Calsou[1,7], Stephan M Hacker[2,3], Yves Génisson[4*‡], Remi Chauvin[5*‡], Sébastien Britton[1,7*]

[1]Institut de Pharmacologie et de Biologie Structurale, IPBS, CNRS, Université de Toulouse, Toulouse, France; [2]Leiden Institute of Chemistry, Leiden University, Leiden, Netherlands; [3]Department of Chemistry, Technical University of Munich, Garching, Germany; [4]SPCMIB, UMR5068, CNRS, Université de Toulouse, UPS, Toulouse, France; [5]LCC-CNRS, Université de Toulouse, CNRS, UPS, Toulouse, France; [6]INRAE, UR 875 Unité de Mathématique et Informatique Appliquées, Genotoul Bioinfo Auzeville, Castanet-Tolosan, France; [7]Equipe labellisée la Ligue contre le Cancer 2018, Toulouse, France; [8]Department of Organic and Inorganic Chemistry, Science Center, Federal University of Ceará, Fortaleza, Brazil; [9]INRAE, US 1426 GeT-PlaGe, F-31326, Castanet-Tolosan, France

*For correspondence:
yves.genisson@univ-tlse3.fr (YG);
remi.chauvin@univ-tlse3.fr (RC);
sebastien.britton@ipbs.fr (SB)

†These authors contributed equally to this work
‡These authors also contributed equally to this work

**Abstract** Hundreds of cytotoxic natural or synthetic lipidic compounds contain chiral alkynylcarbinol motifs, but the mechanism of action of those potential therapeutic agents remains unknown. Using a genetic screen in haploid human cells, we discovered that the enantiospecific cytotoxicity of numerous terminal alkynylcarbinols, including the highly cytotoxic dialkynylcarbinols, involves a bioactivation by HSD17B11, a short-chain dehydrogenase/reductase (SDR) known to oxidize the C-17 carbinol center of androstan-3-alpha,17-beta-diol to the corresponding ketone. A similar oxidation of dialkynylcarbinols generates dialkynylketones, that we characterize as highly protein-reactive electrophiles. We established that, once bioactivated in cells, the dialkynylcarbinols covalently modify several proteins involved in protein-quality control mechanisms, resulting in their lipoxidation on cysteines and lysines through *Michael* addition. For some proteins, this triggers their association to cellular membranes and results in endoplasmic reticulum stress, unfolded protein response activation, ubiquitin-proteasome system inhibition and cell death by apoptosis. Finally, as a proof-of-concept, we show that generic lipidic alkynylcarbinols can be devised to be bioactivated by other SDRs, including human RDH11 and HPGD/15-PGDH. Given that the SDR superfamily is one of the largest and most ubiquitous, this unique cytotoxic mechanism-of-action could be widely exploited to treat diseases, in particular cancer, through the design of tailored prodrugs.

## Editor's evaluation

This manuscript describes an elegant chemical-genetic strategy to discover that human oxidoreductase HSD17B11 is a major contributor to the bioactivation of the dialkinylcarbinol class of cytotoxic natural products. Mechanistic work further revealed that the reactive metabolites generated

by HSD17B11 modify lysine and cysteine side-chains on proteins, leading to an unfolded protein response and apoptotic cell death. This study thus provides a plausible mechanism to explain how dialkynylcarbinol compounds exert their cytotoxic properties and identifies enzyme targets for controlling this process in human cells.

## Introduction

Nature is a rich source of bioactive compounds, some of which can be directly exploited to treat diseases. Some of them reveal sophisticated mechanisms of action which can be mimicked by designing synthetic molecules with specific features (*Newman and Cragg, 2020*). Marine sponges have attracted pharmaceutical interest since the discovery in the 1950s of C-nucleosides in *Crypto-tethia crypta* that led to the development of cytosine arabinoside (ara-C or cytarabine) and analogues as anticancer treatments for acute myelogenous leukemia (*Bergmann, 1950*; *Ellison et al., 1968*). In a different structural series, several cytotoxic acetylenic lipids bearing a terminal alkenylalkynylcarbinol (AAC) pharmacophore have since been isolated from marine sponges, such as petrocortyne A (*Figure 1—figure supplement 1*), isolated from *Petrosia sp.* (*Seo et al., 1998*) and fulvinol isolated from *Haliclona fulva* (*Ortega et al., 1996*). The simplest cytotoxic AAC representative, (*S*)–eicos-(*4E*)-en-1-yn-3-ol ((*S*)–**1**, *Figure 1—figure supplement 1*), was isolated from the marine sponge *Cribrocha-lina vasculum* (*Gunasekera and Faircloth, 1990*). It demonstrated high cytotoxic activity selectively towards non-small cell lung carcinoma cells as compared to normal lung fibroblasts (*Zovko et al., 2014*). Starting from (*S*)–**1**, an extensive structure-activity relationship study in human cancer cell lines established that (*Figure 1—figure supplement 1*): (i) the non-natural enantiomer (*R*)–**1** has higher cytotoxic activity, (ii) homologues with shorter lipidic tails are more cytotoxic, with an optimum total aliphatic backbone of 17 carbon atoms (e.g. (*R*)–**2**), and (iii) replacement of the internal C=C bond by a C≡C bond, giving rise to a terminal dialkynylcarbinol (DAC) pharmacophore, further increases cytotoxicity, to reach an $IC_{50}$ down to 90 nM for the DAC (*S*)–**3** (*El Arfaoui et al., 2013*; *Listunov et al., 2015a*; *Listunov et al., 2018b*). However, despite this significant level of activity, the mode of action of this family of molecules, including the natural compound (*S*)–**1**, remains elusive (*Zovko et al., 2014*).

Here, we use functional genomics and chemoproteomics to decipher how cytotoxic DACs and related molecules mediate their biological effect. We discover that they behave as prodrugs enantiospecifically bioactivated by a member of the Short-chain Dehydrogenase/Reductase (SDR) family. Finally, we design new SDR-bioactivated DACs derivatives, establishing this family of lipidic alkynylcarbinols as a large and untapped reservoir of cytotoxic prodrugs.

## Results

### The SDR HSD17B11 governs (*S*)-DACs cytotoxicity

To determine how cytotoxic DACs mediate their effect on human cells, we applied a genetic approach using the pseudo-haploid human cell line HAP-1 (*Carette et al., 2009*). Given that (*S*)–**3** had the greatest cytotoxic activity of all the DACs previously tested (*El Arfaoui et al., 2013*; *Listunov et al., 2015a*; *Listunov et al., 2018b*), we screened for mutations that could render HAP-1 cells resistant to (*S*)–**3**. We first confirmed in HAP-1 that (*S*)–**3**, but not (*R*)–**3** (*Figure 1A*), exhibits nanomolar cytotoxic activity (*Figure 1B*, $IC_{50}$ 62.4 nM), in agreement with previous results on HCT116 colon cancer cells (*El Arfaoui et al., 2013*). We used Ethyl-Methane Sulfonate (EMS) to generate a mutagenized HAP-1 population and selected resistant clones using a lethal 250 nM (*S*)–**3** concentration. Ten individual (*S*)–**3**-resistant clones (DACR) were isolated, displaying a 38- to 62-fold resistance to (*S*)–**3** (*Figure 1C*) but similar sensitivity as parental cells to two unrelated compounds, bortezomib and doxorubicin (*Figure 1—figure supplement 2A,B*). Based on previous work (*Wacker et al., 2012*; *Bossaert et al., 2021*), and considering that EMS induces mainly point mutations under these conditions (*Forment et al., 2017*), we selected four DACR clones for RNA-seq analysis, to identify mis- or non-sense mutations accounting for the resistance. Around nine mutated genes were identified per clone (*Figure 1—figure supplement 2C*), with *KCTD5* and *HSD17B11* being the only mutated genes shared by more than two clones (*Figure 1—figure supplement 2D*). *KCTD5* encodes for an E3-ubiquitin ligase substrate adaptor identified in a genetic screen as a negative regulator of the Akt

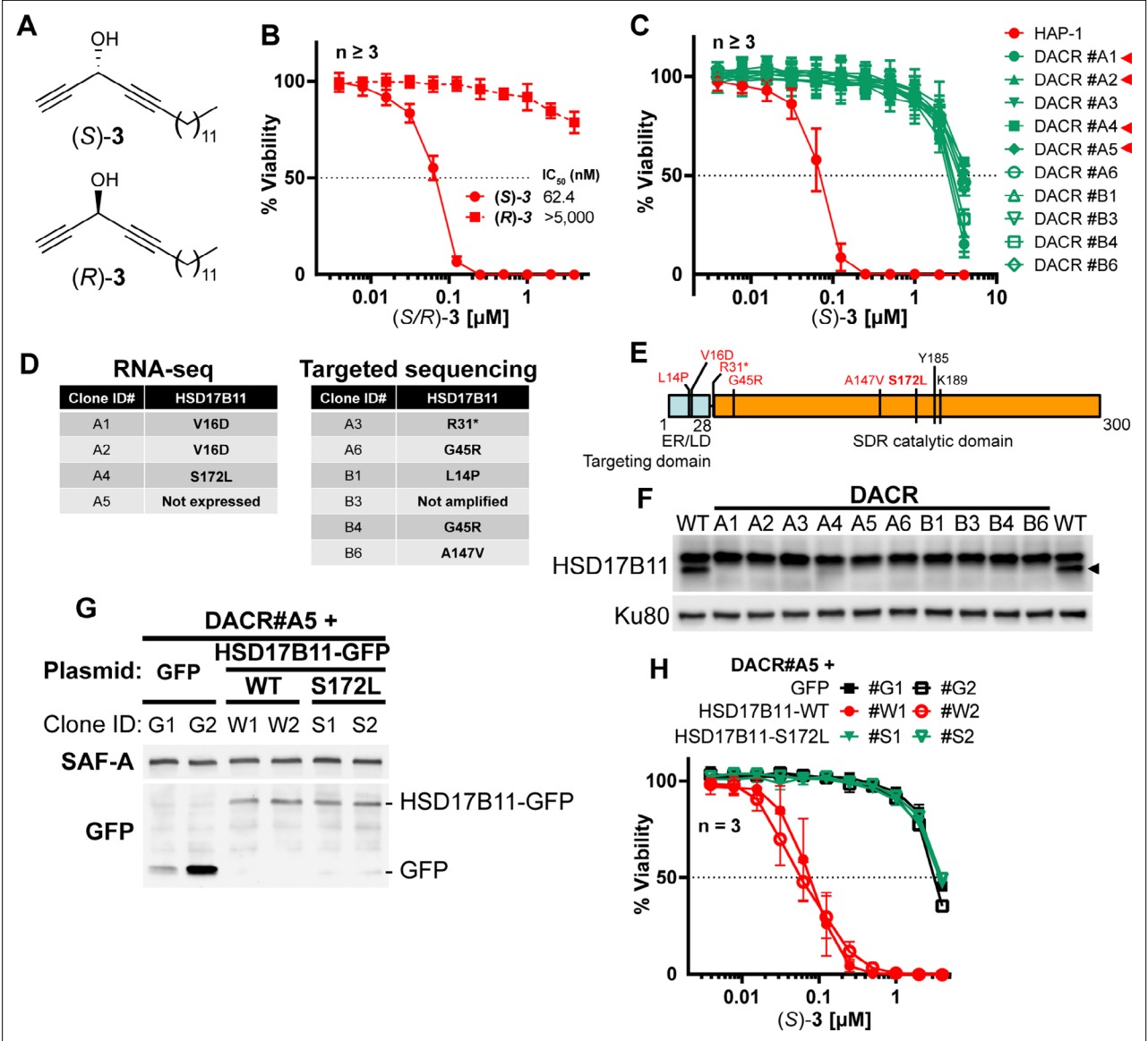

**Figure 1.** HSD17B11 is necessary for DAC (S)–3 cytotoxic activity. (**A**) DAC (S)–3 and (R)–3 structures. (**B**) Cell viability analysis of HAP-1 or U2OS cells treated for 72 h with the indicated concentrations of (S)- or (R)–3. (**C**) Cell viability analysis of individual DAC-resistant clones or wild-type HAP-1 treated for 72 hr with the indicated concentrations of (S)–3. (**D**) List of mutations identified by RNA-seq or targeted sequencing of HSD17B11 in individual DAC-resistant clones. (**E**) Schematic representation of HSD17B11 functional domains. The positions of the identified mutations are indicated in red. The Y185, K189 (indicated in black), and S172 amino acids are critical for catalysis. (**F**) Analysis by immunoblotting of HSD17B11 levels in wild-type HAP-1 and DAC-resistant clones. Ku80 was used as a loading control. The black arrow indicates HSD17B11 position. (**G**) Analysis by immunoblotting of HSD17B11-GFP levels in individual clones of DAC-resistant clone A5 complemented with GFP, wild-type or S172L mutant HSD17B11-GFP. SAF-A and total H2AX were used as loading controls. (**H**) Cell viability analysis of individual clones of DAC-resistant clone A5 complemented with GFP, wild-type or S172L mutant HSD17B11-GFP treated for 72 h with the indicated concentrations of (S)–3.

The online version of this article includes the following source data and figure supplement(s) for figure 1:

**Source data 1.** Source data related to *Figure 1F*.

**Source data 2.** Source data related to *Figure 1G*.

**Figure supplement 1.** Representative structures of natural and bioinspired synthetic alkynylcarbinol-containing cytotoxic molecules.

**Figure supplement 2.** Characterization of DACR clones.

**Figure supplement 3.** HSD17B11 inactivation confers resistance to multiple alkynylcarbinol-containing molecules (ACs) of similar configuration.

**Figure supplement 4.** DAC (S)–3 cytotoxic activity is HSD17B11-dependent in multiple cell lines.

**Figure supplement 4—source data 1.** Source data related to *Figure 1—figure supplement 4B*.

*Figure 1 continued on next page*

*Figure 1 continued*

**Figure supplement 5.** DAC (*S*)–3 cytotoxic activity is HSD17B11-dependent in multiple cell lines.

**Figure supplement 5—source data 1.** Source data related to *Figure 1—figure supplement 5A*.

**Figure supplement 5—source data 2.** Source data related to *Figure 1—figure supplement 5C*.

**Figure supplement 5—source data 3.** Source data related to *Figure 1—figure supplement 5E*.

**Figure supplement 5—source data 4.** Source data related to *Figure 1—figure supplement 5G*.

pathway (*Brockmann et al., 2017*). However, while *KCTD5* mRNA was expressed in all DACR clones, *HSD17B11* mRNA levels were strongly reduced in the only clone without *HSD17B11* coding mutations (#A5, *Figure 1—figure supplement 2E*). This suggested that mutations or lack of expression of *HSD17B11* were responsible for DACR clone resistance. To confirm this, we sequenced *HSD17B11* cDNAs from six other DACR clones, and detected non-synonymous *HSD17B11* mutations in five, and no *HSD17B11* cDNA in the sixth, suggesting loss of expression (*Figure 1D, E*). These data strongly supported a role for *HSD17B11* in mediating (*S*)–**3** cytotoxicity.

*HSD17B11* encodes for the estradiol 17-beta-dehydrogenase 11, a member of the SDR super-family. HSD17B11, also called SDR16C2 [*Persson et al., 2009*], PAN1B, DHRS8, or retSDR2, localizes to the endoplasmic reticulum (ER) and lipid droplets (LD) via a N-terminal targeting domain (*Figure 1E*), where it uses NAD+ to catalyze oxidation of the C17 carbinol center of androstan-3-alpha,17-beta-diol to generate androsterone, a weak androgen (*Brereton et al., 2001*; *Horiguchi et al., 2008a*) (see *Figure 2A*). The HSD17B11 protein was barely detectable in all the DACR clones (*Figure 1F*, lower band), suggesting that the mutations result in protein instability. Using the DACR#A5 clone, in which *HSD17B11* RNA was strongly down-regulated (~200 fold, *Figure 1—figure supplement 2E*), we performed complementation experiments with plasmids coding for GFP alone, or wild-type (WT) or S172L HSD17B11-GFP. This mutation was selected because the S172 residue is critical for catalysis (*Filling et al., 2002*; *Gao et al., 2021*), and the DACR#A4 clone, which carried S172L mutations, was the only one in which traces of full-length HSD17B11 could be detected (*Figure 1F*). Complemented DACR#A5 cells stably expressing WT and S172L HSD17B11-GFP at similar levels were successfully isolated (*Figure 1G*), and (*S*)–**3** was ~50 times more active against cells expressing WT HSD17B11 compared to control GFP-complemented cells or cells expressing S172L HSD17B11 (*Figure 1H*). This supports that HSD17B11 catalytic activity is critical for (*S*)-DAC cytotoxicity. Notably, the DACR#A4 clone (S172L mutation) was also resistant to six other cytotoxic AACs: the naturally occurring AAC (*S*)–**1** (*Figure 1—figure supplement 3A*), its synthetic enantiomer (*R*)–**1**, its shorter homologue (*R*)–**2**, the synthetic AAC (*S*)–**4** with an internal C≡C bond and an external C=C bond (*Figure 1—figure supplement 3B*), the allenylalkynylcarbinol (AllAC) (*R*,*S*ₐ)–**5** (*Listunov et al., 2018a*) and the more cytotoxic butadiynylalkynylcarbinol (BAC) (*S*)–**6** (*Bourkhis et al., 2018*; *Figure 1—figure supplement 3C*). Thus, HSD17B11 functionality governs the enantiospecific cytotoxicity of the natural compound (*S*)–**1** but also of all the more cytotoxic synthetic derivatives tested. In addition, HSD17B11 has been recently identified as mediating, through an unknown mechanism, the cytotoxic effect of dehydrofalcarinol, a polyacetylenic compound with a terminal butadiynylalkenylcarbinol motif isolated from several plants of the *Asteraceae* family (*Grant et al., 2020*).

We next tested the cytotoxic activity of (*S*)–**3** on a panel of 15 cancer cell lines. This revealed that the osteosarcoma U2OS cell line was the most sensitive to (*S*)–**3** while the breast cancer cell line T47D was highly resistant (*Figure 1—figure supplement 4A*). Accordingly, HSD17B11 protein was undetectable in T47D, while U2OS displayed the highest levels (*Figure 1—figure supplement 4B*), in agreement with reported mRNA levels (*The Cancer Cell Line Encyclopedia dataset* [*Barretina et al., 2012*]). In addition, (*S*)–**3** was particularly cytotoxic toward four other osteosarcoma cell lines as compared to normal cell lines or primary osteoblasts (*Figure 1—figure supplement 4C*). CRISPR/Cas9-mediated inactivation of HSD17B11 also conferred significant (*S*)–**3** resistance to U2OS cells, which was suppressed by wild-type HSD17B11-GFP but not by the S172L mutant or GFP alone (*Figure 1—figure supplement 5A, B*). In contrast, complementation with HSD17B11 carrying the L14P or V16D mutations, lying outside of the catalytic domain and identified in the DACR clones #B1 and #A1/#A2, respectively, restored (*S*)–**3** cytotoxic activity, in agreement with these mutations affecting HSD17B11 protein stability and not its catalytic activity (*Figure 1—figure supplement 5C, D*). These data also support that the C-terminal FLAG-GFP tag and/or the CMV promoter-based

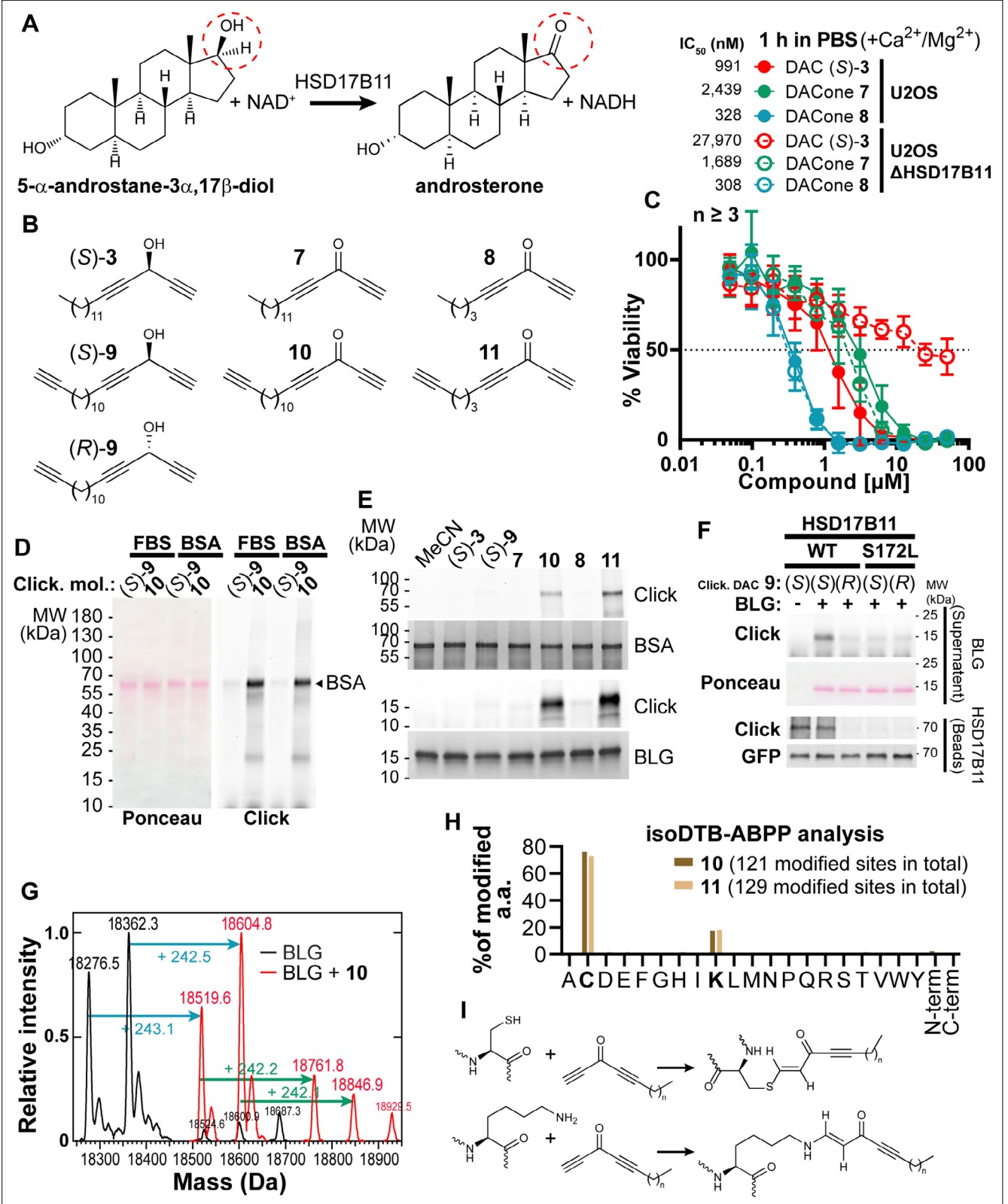

**Figure 2.** DACones are protein reactive species. (**A**) Reaction catalyzed by HSD17B11. (**B**) Clickable DACs and DACones used in the study. (**C**) Viability analysis of U2OS cells treated in PBS for 1 h with (*S*)–**3** or DACones and incubated for an additional 72 h after drug washout. (**D**) FBS or purified BSA were incubated 40 min at 30 °C with clickable DAC (*S*)–**9** or clickable DACone **10**. After reaction, CuAAC was used to ligate an azido-AlexaFluor647 to clickable molecules. Modified proteins were detected by scanning membrane fluorescence after SDS-PAGE and transfer. Ponceau S stains total proteins.

*Figure 2 continued*

(**E**) BSA or BLG were incubated with the indicated DACs or DACones, as in (**D**). After reaction, modified proteins were detected as in (**D**). Coomassie stains total proteins. (**F**) WT or S172L HSD17B11-GFP were immunoprecipitated from complemented U2OS KO HSD17B11 cells and incubated with clickable DAC **9** and BLG. After reaction, modified proteins were detected in the supernatant (BLG) or on the beads (HSD17B11-GFP) as in (**D**). GFP immunoblotting confirmed that equal amounts of WT and S172L proteins were used. (**G**) Analysis by direct-infusion mass spectrometry of purified BLG (mixture of isoform A and B) modified or not by DACone **10**. Cyan and green arrows indicate the formation of a first and second adduct, respectively. (**H**) % of each amino acid detected as modified by DACones **10** or **11** in U2OS extracts as determined using an isoDTB-ABPP-based framework. (**I**) Proposed reactions of DACones with cysteine and lysine side chains in proteins.

The online version of this article includes the following source data and figure supplement(s) for figure 2:

**Source data 1.** Source data related to *Figure 2D*.

**Source data 2.** Source data related to *Figure 2E*.

**Source data 3.** Source data related to *Figure 2F*.

**Figure supplement 1.** DACones are protein-reactive species.

**Figure supplement 1—source data 1.** Source data related to *Figure 2—figure supplement 1A*.

**Figure supplement 2.** Molecular docking of (*S*)- and (*R*)–3 on 17-βHSD.

**Figure supplement 2—source data 1.** Source data related to *Figure 1—figure supplement 2G*.

**Figure supplement 3.** Sequence context of modified cysteines and lysines.

**Figure supplement 4.** Characterization of DACone reaction products with NAC and NAK.

overexpression partly overcome the impact of these mutations on HSD17B11 expression level. The role of HSD17B11 in (*S*)–**3** cytotoxic activity was further confirmed using two different small-interfering RNAs (siRNA) to down-regulate HSD17B11 in U2OS (*Figure 1—figure supplement 5E, F*) and in the non-small cell lung carcinoma cell line A549, in which CRISPR/Cas9-mediated HSD17B11 inactivation also conferred (*S*)–**3** resistance (*Figure 1—figure supplement 5G,H*). Altogether, these data establish that HSD17B11 is critical in multiple cell lines for (*S*)–**3** cytotoxic activity, and suggest that (*S*)–**3** behaves as an HSD17B11-bioactivated prodrug. In addition, the acute toxicity of (*S*)–**3** towards osteosarcoma cell lines suggests that DACs could be developed into a targeted anticancer therapy, but this would need to be further investigated, especially *in vivo*.

## Dialkynylketones are protein-reactive species

We next investigated the downstream mechanism of cytotoxic action of the DAC (*S*)–**3.** The C17 carbinol center of androstan-3-alpha,17-beta-diol, which is naturally oxidized by HSD17B11 (*Figure 2*; *Brereton et al., 2001*), has the same spatial orientation as the (*S*)–**3** carbinol when its lipidic chain is superimposed with the C13(C18) side of the steroid skeleton (*Figure 2B*). This suggested that HSD17B11 enantiospecifically recognizes and oxidizes (*S*)–**3** into a "dialkynylketone" **7** (DACone), a diynone that could be the cytotoxic species. However, when the DACone **7** was previously synthesized and tested, no cytotoxic activity was found (*Listunov et al., 2015a*). Given the high *in vitro* electrophilic reactivity of ynones as *Michael* acceptors of thiols and amines (*Worch et al., 2021*), we considered that medium components such as serum albumin may rapidly react with and inactivate DACones. To test this, we synthesized the DACone **7**, as well as a homologue with a shorter alkyl chain **8,** and treated U2OS cells in a protein-free medium (PBS containing CaCl$_2$ and MgCl$_2$ to maintain cellular adhesion). Both the DACones **7** and **8** were indeed cytotoxic in the absence of serum, with **8** (short chain) being even more active than (*S*)–**3** (*Figure 2C*). While the cytotoxicity of (*S*)–**3** was strongly reduced by inactivation of HSD17B11, the cytotoxicity of the DACones **7** and **8** was not affected, supporting the notion that the DACones are the cytotoxic products generated from DACs by HSD17B11.

To further analyze the interaction between DACones and proteins, we synthesized 'clickable' analogues, that is bearing a terminal C≡CH tag, for each DAC enantiomer ((*S*)–**9** and (*R*)–**9**), and for long and short DACones (**10** and **11**, *Figure 2B*), and used them to monitor the formation of covalent bonds between DACones and serum proteins by copper-catalyzed azide-alkyne cycloaddition (CuAAC 'click chemistry', [*Tornøe et al., 2002*; *Rostovtsev et al., 2002*]). The clickable DACone **10**, or clickable DAC (*S*)–**9** as control, were incubated with fetal bovine serum (FBS) or purified bovine serum albumin (BSA), followed by CuAAC-mediated ligation of an AlexaFluor647-azido fluorophore to the free C≡CH tag. The proteins were separated by SDS-PAGE and scanned for fluorescence

(*Speers and Cravatt, 2004*). Covalent adducts were formed on BSA with DACone **10** but not with (*S*)–**9** (*Figure 2D*). Moreover, the DACone **10** also reacted with several other model proteins, including the bovine beta-lactoglobuline (BLG) (*Figure 2—figure supplement 1A*). Using BSA and BLG, we established that DACone adducts are produced only when using the clickable DACones **10** or **11** (*Figure 2E*), suggesting that the terminal triple bond of the DACone pharmacophore is modified or masked after reaction. Finally, we could recapitulate the activation of (*S*)–**9**, but not of (*R*)–**9**, into protein-reactive species by immunopurified WT HSD17B11, but not by the S172L mutant (*Figure 2F*), supporting an enantiospecific bioactivation of (*S*)–**9** into the BLG-reactive DACone **10** by HSD17B11. Considering that HSD17B11 known activity is the NAD+-dependent oxidation of a secondary carbinol into a ketone and that the only hydroxyl group on (*S*)–**3** is the one occurring in the dialkynylcarbinol pharmacophore, this experiment strongly supports the notion that HSD17B11 enantiospecifically oxidizes (*S*)–**3** into the DACone **7**, which immediately reacts with nearby proteins, including HSD17B11-GFP itself as observed in *Figure 2F*. This high level of reactivity unfortunately precludes isolating the HSD17B11-produced DACones. To understand the basis for this enantiospecific bioactivation, we used AlphaFold2 (*Jumper et al., 2021*; *Evans et al., 2021*; *Mirdita et al., 2021*) to generate a structural model of HSD17B11 (*Figure 2—figure supplement 2A*) and performed molecular docking of (*S*)–**3** and (*R*)–**3** into its catalytic core. Both (*S*)–**3** and (*R*)–**3** docked into the catalytic cavity (*Figure 2—figure supplement 2B,C*), but (*R*)–**3** had a lower computed affinity than (*S*)–**3** (155 nM vs 15 nM) and only (*S*)–**3** had its hydroxyl group properly positioned to engage hydrogen bonds with the S172 and Y185 catalytic amino acids (*Figure 2—figure supplement 2D, E*), which is critical for further carbinol oxidation via hydride transfer to NAD+ (*Filling et al., 2002*; *Gao et al., 2021*). Combining docking of (*S*)- and (*R*)–**3** on AlphaFold2 models of the 12 other human 17β-hydroxysteroid dehydrogenase (17β-HSDs) SDRs, with filtering to select the most stringent interactions, was used to identify other SDRs that might be able to bioactivate (*S*)–**3** (*Figure 2—figure supplement 2F*). This filtering revealed that, beyond HSD17B11, (*S*)–**3** also docked onto the catalytic domains of only two 17β-HSDs, HSD17B13 and HSD17B3, while (*R*)–**3** docked onto two different 17β-HSDs, HSD17B9 and HSD17B6. HSD17B3 (aka EDH17B3 or SDR12C2 [*Persson et al., 2009*]) is a reductive SDR involved in testosterone biosynthesis so it was not tested further. In contrast, since HSD17B13 (aka SCDR9 or SDR16C3 [*Persson et al., 2009*]), whose expression is restricted to the liver, is an oxidative SDR, the closest homologue of HSD17B11 and is also localized at the ER (*Horiguchi et al., 2008b*), we tested whether it could complement U2OS KO for HSD17B11. This revealed that HSD17B13 is also able to bioactivate (*S*)–**3** into cytotoxic compounds, albeit in a less efficient manner as compared to HSD17B11 (IC$_{50}$ of 30 nM *vs* 12 nM for HSD17B11 with similar complementation levels, *Figure 2—figure supplement 2G, H*).

## Reaction of DACones with proteins

To further decipher the reaction of DACones with proteins, we used direct-infusion mass spectrometry to analyze BLG modified with the clickable DACone **10**. Purified BLG contains two isoforms (A and B, differing by 86.0 Da) and, when incubated with DACone, both BLG isoforms were completely modified with the formation of one or two adducts of ~+242 Da (*Figure 2G*), which corresponds to the mass of the clickable DACone **10**. Monitoring the absorbance spectra of modified BLG revealed that BLG gains an absorption band at ~323 nm upon modification by DACone (*Figure 2—figure supplement 1B*). Using this, we confirmed that both BLG and BSA are modified by the DACones **7** and **8** or their clickable analogues **10** and **11** (*Figure 2—figure supplement 1B, C*). The shorter DACone **8** proved to be even more reactive, in line with its greater cytotoxicity (*Figure 2C*, *Figure 2—figure supplement 1B, C*). Next, we assessed the selectivity of DACones towards amino acid residues in the whole proteome in an unbiased fashion. For this purpose, we incubated the DACones **10** and **11** with U2OS total cell extracts in PBS. We then used residue-specific chemoproteomics with isotopically labeled desthiobiotin azide (isoDTB) tags (*Backus et al., 2016*; *Weerapana et al., 2010*; *Zanon et al., 2020*) coupled to a novel MSFragger-based FragPipe computational platform (*Zanon et al., 2021*) to detect the modified amino acids on the enriched peptides. This revealed that both DACones reacted with cysteine and lysine side chains, with the expected modification being detected ( + 729.4498/ + 723.4408 Da (Heavy/Light) for DACone **10**, + 631.3404/ + 625.3332 Da (Heavy/Light) for DACone **11**, *Figure 2H*, *Figure 2—figure supplement 1D, E, F*, *Supplementary file 1A, B*). We also detected many unmodified peptides (*Supplementary file 1D, E*), most of them with at least one missed lysine trypsin cleavage site (~92% of sequences), suggesting that these were still modified

during digest and the modification lost during the subsequent workflow, potentially during the final trifluoroacetic acid elution (0.1%, pH~2 [*Zanon et al., 2021*]). We cannot fully exclude that other amino acids were also modified by the probe to some degree and that this modification was also lost during the workflow, but this data points to the fact that lysines and cysteines are the main modification sites. We next confirmed the reactivity of DACones with cysteine and lysine side chains by monitoring the appearance of the ~323 nm absorbance band after reaction of the DACone **8** with isolated amino acids, using *N*-acetylated versions to prevent reactions with the N-terminal amino group. At neutral pH, DACones only reacted with *N*-acetyl-L-cysteine (NAC) but not with $N_\alpha$-acetyl-L-lysine (NAK) (*Figure 2—figure supplement 1G*, left spectrum), whereas at higher pH they reacted with both NAC and NAK (*Figure 2—figure supplement 1G*, right spectrum), in agreement with the nucleophilic reactivity of the non-protonated $\varepsilon-NH_2$ group of the lysine chain. No reaction was observed with *N*-acetyl glycine (NAG), supporting that the reaction involves the side chain. By monitoring the adducts absorbance, we confirmed that the DACone **8**-NAK linkage is progressively lost when incubated in 0.1% TFA (38% reduction after 1 h incubation), while the DACone **8**-NAC linkage remains unaffected in these conditions (*Figure 2—figure supplement 1H, I*). These data suggest that the reactivity with lysine side chains is underestimated by our isoDTB-ABPP experiment due to preferential loss of the lysine-DACone species. The reaction with lysine side chains is compatible with the $pK_a$ value for the lysine ε-amino group that can be as low as ~5 in hydrophobic domains in proteins (*Isom et al., 2011*). Accordingly, analysis of the sequence context of the amino acids identified as modified by the DACone **10** revealed an enrichment of hydrophobic amino acids around the modified lysines, which was not observed for the modified cysteines (*Figure 2—figure supplement 3A, B*, *Supplementary file 2*). Using nuclear magnetic resonance (NMR), we characterized the products of the reaction of the short DACone **8** with NAC (*Figure 2—figure supplement 4A, B*) or NAK (*Figure 2—figure supplement 4C, D*). This revealed that a covalent bond forms by addition of the thiol (NAC) or amino (NAK) group onto the terminal alkyne of the DACone head (*Figure 2—figure supplement 4E*). A similar reaction probably occurs with the cysteine and lysine residues of proteins, which is supported by their gain of a similar absorbance band upon treatment with DACone (*Figure 2I*). This additional absorption band can be accounted for by the donor-acceptor extension of π-electron delocalization in the enone adducts (S–CH=CH–C=O for NAC, N–CH=CH–C=O for NAK). Altogether, our data show that DACones are highly reactive with proteins *in vitro*.

## Bioactivated (*S*)-DACs lipoxidize multiple proteins in cells

Protein modification by lipidic DACs equates to their lipoxidation (a term used to designate the covalent modification of a protein by a reactive lipid [*Viedma-Poyatos et al., 2021*]) by one or several C17 hydrophobic chain(s) (*Figure 3—figure supplement 1*). Considering that protein palmitoylation (addition of a C16 lipidic chain) can trigger membrane tethering of proteins, we hypothesized that lipoxidation by DACs could affect protein localization and/or function and account for the cytotoxicity of bioactivated DACs in cells as described for other reactive lipids (*Viedma-Poyatos et al., 2021*). To challenge this hypothesis in cells, we took advantage of the clickable DAC **9** (*Figure 2B*, [*Listunov et al., 2015b*]). As observed for the DAC **3**, the cytotoxicity of the clickable DAC **9** was enantiospecific, biased toward (*S*)–**9**, and dependent on bioactivation by HSD17B11 (*Figure 3—figure supplement 2A*). Cells were treated with clickable (*S*)- or (*R*)–**9** DACs, extracts prepared and click chemistry used to detect the covalent adducts of DACs onto proteins (*Speers and Cravatt, 2004*). Multiple modified proteins were detected in extracts from (*S*)–**9**-treated cells, while no adduct with (*R*)–**9** was detected (*Figure 3A*).

To identify the proteins lipoxidized by DACs upon bioactivation, and inspired by previous studies (*Larrieu et al., 2014*; *Vila et al., 2008*), we used streptavidin pull-down to isolate (*S*)–**9**-modified proteins after click chemistry-mediated ligation of a biotin handle in extracts of treated cells and identified them using bottom-up proteomics. Forty-two proteins were significantly enriched more than 2-fold in the (*S*)–**9** condition as compared to (*R*)–**9** (*Figure 3B*, *Supplementary file 3*), with three proteins being enriched more than 60-fold: BRAT1, PLIN3, and PSMD2. In the MS/MS data, we could not detect peptides with the DACone-triazole-PEG-biotin modification, suggesting either that the biotin-carrying peptides remained associated with the beads or that the modification was lost during the LC-MS/MS workflow. We also attempted to adapt the isoDTB tag-based chemoproteomics workflow to identify the modified peptides directly from U2OS cells treated for 2 h with 2 µM (*S*)–**9**. Under

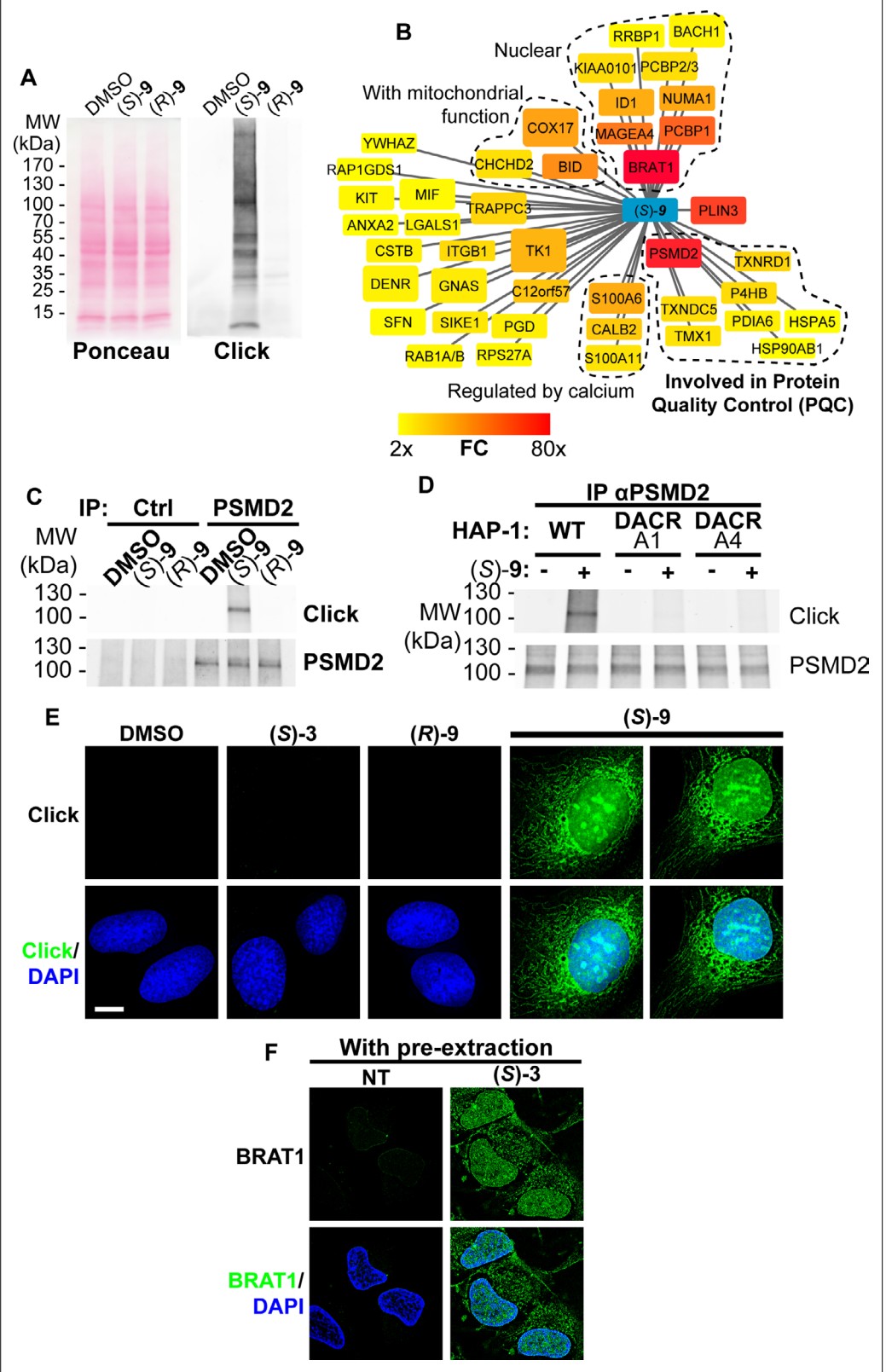

**Figure 3.** (*S*)-DACs lipoxidize multiple cellular proteins, triggering their association with cellular membranes. (**A**) U2OS cells were incubated for 2 h with 2 μM (*S*)- or (*R*)–**9**, proteins were extracted and DAC-modified proteins were detected by CuAAC-mediated ligation of azido-AlexaFluor-647 to clickable molecules, separation by SDS-PAGE, transfer to a membrane which was scanned for fluorescence. (**B**) Landscape of proteins modified in U2OS cells by

*Figure 3 continued*

clickable DAC (*S*)–**9** computed from three independent experiments. Fold enrichment (FC) as compared to the clickable (*R*)–**9** is computed and color-coded as depicted. Box size corresponds to -log(p) computed as described in the materials and methods section. (**C**) PSMD2 or control immunoprecipitations (Ctrl) were performed from extracts of U2OS cells treated 2 h with 2 µM clickable DAC (*S*)- or (*R*)–**9**. DAC-modified proteins were detected by CuAAC-mediated ligation of azido-AlexaFluor-647 to clickable molecules, separation by SDS-PAGE, transfer to a membrane, which was scanned for fluorescence. PSMD2 was subsequently visualized by immunoblotting. (**D**) PSMD2 immunoprecipitations were performed from extracts of wild-type or DAC-resistant HAP-1 cells treated or not for 2 h with 2 µM clickable DAC (*S*)–**9**. DAC-modified proteins were detected by CuAAC-mediated ligation as in (**C**). PSMD2 was subsequently visualized by immunoblotting. (**E**) U2OS cells were treated 2 h with 0.5 µM DAC, fixed, permeabilized, and clickable molecules were detected by click with AlexaFluor488 azide. (**F**) U2OS expressing GFP-BRAT1 were treated 2 h with 1 µM (*S*)–**3**, pre-extracted, fixed and processed for analysis by fluorescence microscopy.

The online version of this article includes the following source data and figure supplement(s) for figure 3:

**Source data 1.** Source data related to *Figure 3A*.

**Source data 2.** Source data related to *Figure 3C*.

**Source data 3.** Source data related to *Figure 3D*.

**Figure supplement 1.** Parallel between protein palmitoylation and protein lipoxidation by the DAC **3**.

**Figure supplement 2.** DAC (*S*)–**9** is bioactivated into protein-reactive species.

**Figure supplement 2—source data 1.** Source data related to *Figure 3—figure supplement 1B*.

**Figure supplement 2—source data 2.** Source data related to *Figure 3—figure supplement 2C*.

**Figure supplement 2—source data 3.** Source data related to *Figure 3—figure supplement 2D*.

**Figure supplement 2—source data 4.** Source data related to *Figure 3—figure supplement 2E*.

**Figure supplement 3.** Glutathione can be modified by DACones *in vitro* but does not impact on (*S*)-DAC 3 cytotoxic activity.

**Figure supplement 4.** The clickable DAC (*S*)–**9** staining colocalizes with the nucleus, ER and mitochondria.

**Figure supplement 5.** The DAC (*S*)–**3** triggers ER-swelling and mitochondrial fission.

these conditions, we detected a few peptides modified on a cysteine by the expected adduct plus the mass of a dithiothreitol group (DTT, *Supplementary file 4A*), evocative of a DTT reaction with the internal alkyne functionality (see *Supplementary file 4B* for the potential adduct structure, *Lei et al., 2021*), a type of adduct which was also detected in the isoDTB-based analysis presented in *Figure 2H*, albeit in a smaller proportion (~30% of all quantified sites with DACone **10** & **11**). This experiment supports that the reaction between DACones and N-acetyl-amino acids established *in vitro* also takes place in cells when DACones are produced by HSD17B11. The small number of modified sites identified in this experiment can be explained by the instability of the adduct on lysines during the isoDTB-ABPP workflow, the lower DAC concentration used for cell treatment (2 µM for (*S*)–**9**) as compared to the *in vitro* experiment (100 µM of **10** or **11**) and the need for bioactivation of (*S*)–**9** in contrast to the constantly active **10** or **11**.

To validate that the three most highly enriched hits are modified in cells, we overexpressed BRAT1, PSMD2 and PLIN3 individually (in addition to TK1) as GFP fusions in U2OS cells (*Figure 3—figure supplement 2B*), and used GFP pull-down to determine whether they were modified by clickable DAC **9** in cells. These proteins were found robustly modified by (*S*)–**9** but not by (*R*)–**9**, especially BRAT1 and PSMD2, while GFP alone was not modified (*Figure 3—figure supplement 2C*). The fact that only 42 proteins were found robustly and reproducibly enriched with (*S*)–**9** over (*R*)–**9** in cells indicates a certain degree of selectivity for protein modification by DACones that could result from its production and reactivity being restricted to the vicinity of HSD17B11. In agreement, while the thiol group of reduced glutathione can be modified *in vitro* by DACones at pH7 with the formation of an adduct similar to the one with NAC (*Figure 3—figure supplement 3A–C*), suggesting that it could provide some protection against the reactivity of DACones in cells, the toxicity of (*S*)–**3** was not significantly decreased by co-treatment with a cell-permeable glutathione or increased by a glutathione-S-transferase inhibitor (GSTi, *Figure 3—figure supplement 3D, E*). These data suggest that DACones rapidly and selectively react with a specific set of proteins in the vicinity of HSD17B11 as shown for another *in situ* generated electrophile (*Paxman et al., 2018*).

Among the three main hits, PSMD2 drew our attention as an essential protein in HAP-1 cells (*Blomen et al., 2015*). PSMD2, also called Rpn1, is a critical non-catalytic subunit of the 19 S regulatory particle of the 26 S proteasome, a large complex responsible for the ubiquitin-dependent degradation of cellular proteins. PSMD2 is essential for 19 S assembly and for docking of ubiquitin, ubiquitin receptors and the deubiquitinase USP14 (*Shi et al., 2016*). Immunoprecipitation of endogenous PSMD2 from U2OS cells treated with clickable DAC confirmed that PSMD2 is covalently modified after treatment with the DAC (*S*)–**9** but not with (*R*)–**9** (*Figure 3C*). Moreover, the clickable DACone **10** efficiently modified PSMD2 *in vitro* (*Figure 3—figure supplement 2D*). Using the (*S*)-DAC-resistant HAP-1 clones A1 and A4 (expressing V16D and S172L HSD17B11 mutants, respectively), we also confirmed that the modification of cellular proteins by (*S*)–**9**, including PSMD2, was dependent on HSD17B11 (*Figure 3D* and *Figure 3—figure supplement 2E*). Our proteomics approach also revealed that, in addition to PSMD2, a cluster of proteins involved in protein quality control (PQC) was also modified by the DAC (*S*)–**9** (*Figure 3B*), including several protein disulfide isomerases (P4HB/PDIA1, PDIA6 and TMX1), thioredoxin reductases (TXNDC5 and TXNRD1) and protein chaperones (the ER-resident HSP70, HSPA5/GRP78/BiP; and HSP90AB1), the alteration of which likely also contributes to the DAC cytotoxic effect. Altogether, these data show that (*S*)-DACs are bioactivated by HSD7B11 into highly reactive DACones that covalently lipoxidize nearby proteins, including essential proteins involved in PQC such as PSMD2, a critical subunit of the ubiquitin-proteasome system (UPS).

We then used click-based imaging to monitor the localization of DAC-modified proteins. (*S*)–**9** gave a strong nuclear and cytoplasmic staining, the latter being evocative of ER and mitochondrial membranes (*Figure 3E*). In agreement, we observed a co-occurrence of (*S*)–**9**-click staining with markers of ER (*Figure 3—figure supplement 3A*) and mitochondria (*Figure 3—figure supplement 3B, C*). The lack of staining in cells treated with the inactive (*R*)–**9** (*Figure 3E*) supported that the staining corresponds to DAC-modified proteins. No staining was observed with (*S*)–**3**, supporting that the -C≡CH group of the dialkynylcarbinol head is modified after bioactivation and subsequent reaction with proteins (as shown in *Figure 2I*). Since GFP-BRAT1 is the protein that was the most robustly modified by DACs in cells (*Figure 3—figure supplement 2C*), we pre-extracted soluble proteins with mild nonionic detergent to assess GFP-BRAT1 association to subcellular compartments. Under these conditions, most GFP-BRAT1 was removed by the pre-extraction in untreated cells, as expected for a soluble protein, while it was retained to subcellular compartments evocative of ER and nucleus after treatment with (*S*)–**3** (*Figure 3F*). These data suggest that protein lipoxidation by bioactivated DACs results, at least for some of them, into their relocalization to cellular membranes.

## (*S*)-DACs trigger ER-stress, inhibition of ubiquitin-proteasome system (UPS) and apoptosis.

In agreement with (*S*)-DAC impairing PQC, we observed that treatment of cells by (*S*)–**3** triggers ER swelling as shown by the appearance of a large cytoplasmic vacuole between 4 and 8 h of treatment that preceded cell death (*Figure 4A*). The use of a GFP variant targeted and retained into the ER confirmed that these vacuoles derived from this compartment (*Figure 4B*), while a mCherry protein addressed to mitochondria in the same cells showed that (*S*)-DAC treatment concomitantly triggered mitochondrial fission, a hallmark of cell stress (*Figure 3—figure supplement 4*, see also *Video 1*). ER swelling is a feature of ER-stress, which can result from the accumulation of unfolded proteins within the ER and can be triggered by various defects of PQC such as inhibition of the UPS, as seen with the UPS inhibitor MG132 (*Figure 4—figure supplement 1A*). In agreement with (*S*)–**3** cytotoxicity being mediated by the accumulation of unfolded proteins, vacuolization and cell death induced by (*S*)–**3** could be blocked by inhibition of protein synthesis by cycloheximide (CHX, *Figure 4—figure supplement 1B*). (*S*)–**3** also triggered a strong accumulation of the chaperone HSP70 and of the cell stress response protein p21, similarly to MG132 (*Figure 4C*) or PSMD2 depletion using siRNA (*Figure 4—figure supplement 1C*). It is noteworthy that ER-stress itself can trigger UPS inhibition, likely through consuming free ubiquitin (*Menéndez-Benito et al., 2005*). In agreement with this hypothesis, we observed that (*S*)–**3** induces the accumulation of poly-ubiquitinated proteins (*Figure 4D*), a hallmark of UPS inhibition, and blocked the degradation of an artificial substrate of the UPS system in a manner similar to MG132 (*Figure 4E and F*). Similarly to MG132, (*S*)–**3** also blocked the assembly of the protein 53BP1 into foci at sites of DNA double-strand breaks (*Figure 4—figure supplement 1D*), a process that depends on the local *de novo* DNA damage-induced ubiquitination of histones (*Mailand et al.,*

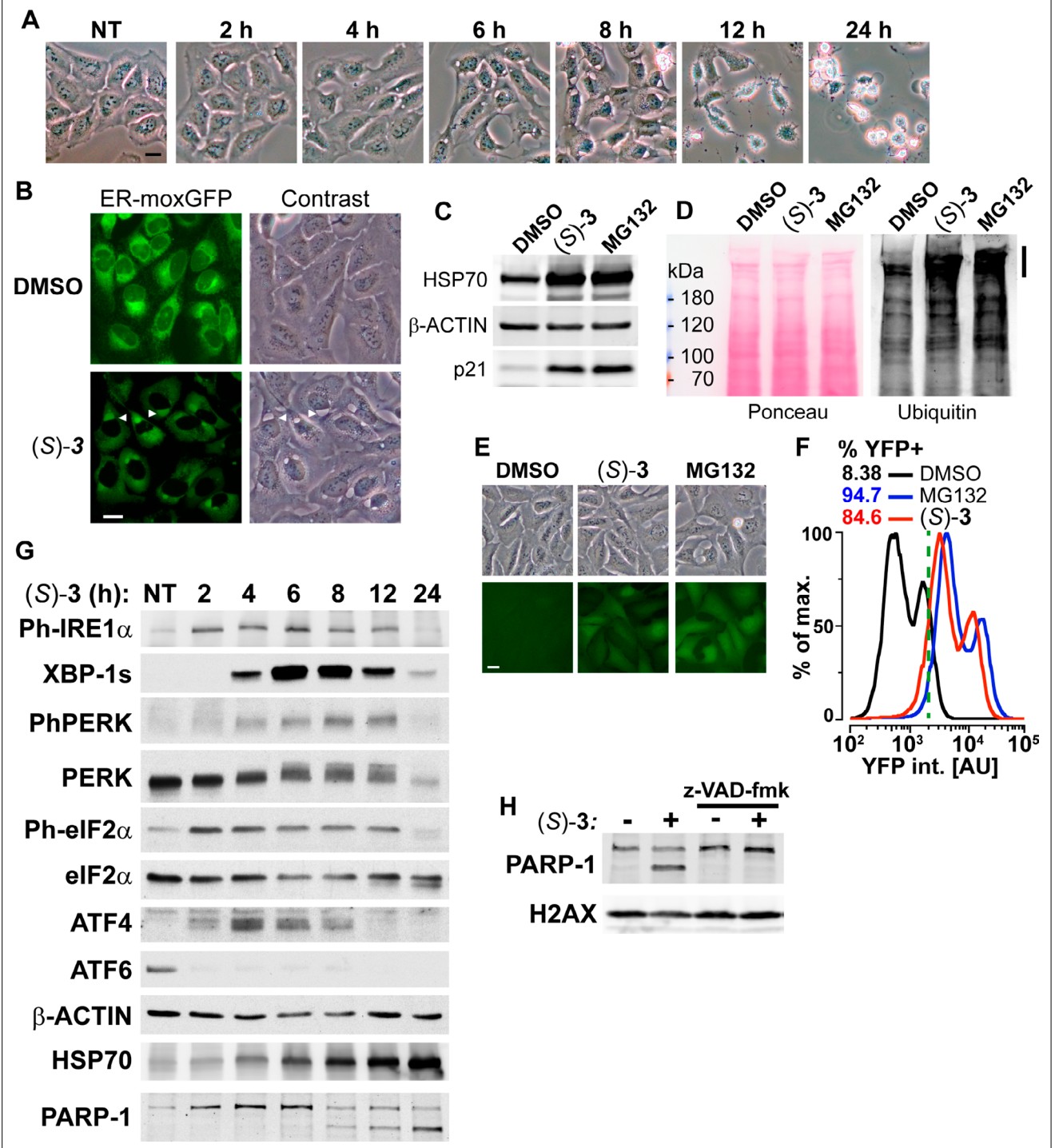

**Figure 4.** DAC (*S*)–3 triggers ER-stress, ubiquitin-proteasome system inhibition and apoptosis. (**A**) U2OS stably co-expressing a GFP variant addressed and retained in the endoplasmic reticulum were treated with 1 µM (*S*)–3 and monitored by live imaging. Representative pictures of U2OS cells, either untreated (NT) or treated with 1 µM (*S*)–3 for the indicated time. (**B**) Representative pictures of U2OS with stably GFP-labeled endoplasmic reticulum and either untreated or treated for 8 h with 1 µM (*S*)–3. (**C**) Immunoblotting of extracts from U2OS untreated or treated with 1 µM (*S*)–3 or 20 µM MG132 for 8 h. (**D**) Immunoblotting of ubiquitin in extracts from U2OS untreated or treated with 1 µM (*S*)–3 or 20 µM MG132 for 2 h. High-molecular-weight ubiquitin conjugates are indicated by a vertical bar on the right. (**E**) Representative pictures of U2OS stably expressing Ub-G76V-YFP and either untreated or treated for 4 h with 1 µM (*S*)–3 or 20 µM MG132. (**F**) Analysis of YFP fluorescence by flow cytometry of U2OS Ub-G76V-YFP treated as described in (**E**),% of cells scored as positives using the vertical green bar as a threshold are indicated. (**G**) Immunoblotting using extracts from U2OS cells treated with 1 µM (*S*)–3 for increasing times, indicated in hours. To probe for all the UPR markers, the same extracts were analyzed on different

*Figure 4 continued on next page*

*Figure 4 continued*

immunoblots, each one with its loading control (see the source data), which were grouped in logical order on the figure to facilitate the interpretation. (**H**) Immunoblotting using extracts from U2OS cells treated with 1 µM (*S*)–**3** for 12 h with or without 50 µM z-VAD-fmk.

The online version of this article includes the following source data and figure supplement(s) for figure 4:

**Source data 1.** Source data related to *Figure 4C*.

**Source data 2.** Source data related to *Figure 4D*.

**Source data 3.** Source data related to *Figure 4G*.

**Source data 4.** Source data related to *Figure 4H*.

**Figure supplement 1.** DAC (*S*)–**3** triggers ER-stress, inhibition of the Ub-proteasome system and apoptosis.

**Figure supplement 1—source data 1.** Source data related to *Figure 4—figure supplement 1C*.

*2007*), thereby confirming depletion of free ubiquitin by treatment with (*S*)– **3**. It is noteworthy that (*S*)– **3** treatment itself did not induce DNA damage nor blocked ATM activation, an important component of the DNA damage response (*Figure 4—figure supplement 1D*). Accumulation of unfolded proteins in the ER activates three ER-resident transmembrane signal transducers: the IRE1α kinase/endoribonuclease, the PERK kinase and the ATF6 transcription factor, which represent the three arms of the Unfolded Protein Response (UPR). The UPR aims at restoring ER protein homeostasis by reducing the influx of proteins into the ER and by increasing the activity of ER protein quality control mechanisms (*Preissler and Ron, 2019*; *Szegezdi et al., 2006*). IRE1α activation is marked by its autophosphorylation on S724 and the processing of the XBP-1 mRNA resulting in the production of the XBP1-s transcription factor. PERK activation is marked by its autophosphorylation on T980 and the phosphorylation of eIF2α S51, which reduces translation initiation and triggers the accumulation of the ATF4 transcription factor. ATF6 activation relies on its cleavage which releases a short-lived transcriptionally active fragment. Persistent UPR activation can promote apoptosis in a JNK-dependent manner (*Szegezdi et al., 2006*). In agreement with (*S*)–**3** triggering unfolded proteins accumulation and UPR activation, we observed that treatment with (*S*)–**3** induced the activation of the three UPR pathways with the rapid phosphorylation of IRE1α (S724), PERK (T980) and eIF2α (S51), a strong decrease of full-length ATF6, preceding the accumulation of cytoplasmic HSP70 and the transient accumulation of XBP-1s and ATF4 (*Figure 4G*). Of note, treatment with inhibitors of either IRE1α endoribonuclease, IRE1α kinase or PERK did not modify the toxicity of (*S*)–**3**, suggesting that the ER stress induced by (*S*)–**3** is too acute to be resolved in a timely manner by the UPR (*Figure 4—figure supplement 1E*). In contrast, JNK inhibition provided a small degree of resistance to (*S*)–**3** ($IC_{50}$ shift from 33 to 70 nM), suggesting a contribution of the JNK pathway to cell death (*Figure 4—figure supplement 1F*). Ultimately, (*S*)–**3** treatment resulted in apoptosis, as marked by PARP-1 cleavage (*Figure 4G*), in a caspase inhibitor (z-VAD-fmk)-sensitive manner (*Figure 4H*). Caspase inhibition blocked cell death but not the vacuolization process (*Figure 4—figure supplement 1G*). Altogether, these results show that (*S*)–**3**, once bioactivated by HSD17B11, covalently modifies multiple proteins including critical components of the PQC, resulting in ER-stress, activation of the UPR and inhibition of the UPS, ultimately leading to apoptotic cell death.

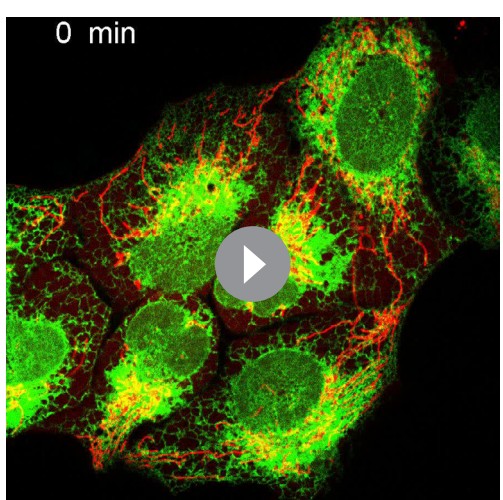

**Video 1.** This video corresponds to a time lapse of pictures of U2OS cells stably expressing ER-moxGFP and Mito-mCherry and treated with 1 µM (*S*)-. Pictures were acquired every 40 min for 1040 min.

https://elifesciences.org/articles/73913/figures#video1

## Identification of new SDR-specific prodrugs

Our discovery that an SDR, HSD17B11, bioactivates a secondary alcohol prodrug by oxidation into the corresponding ketone is particularly interesting given that the SDR superfamily is one

of the largest protein superfamilies, with over 500,000 members found in most forms of life. This means that this mechanism of prodrug activation could be exploited to develop an extensive range of new drugs and compounds to kill cells or organisms expressing specific SDRs with high selectivity. SDRs use NAD(H) or NADP(H) cofactors to perform oxidoreductase, lyase or isomerase activities on a large variety of substrates including steroids, retinoic acids, lipids, polyols, sugars, nucleotides and xenobiotics (*Kavanagh et al., 2008*). *In vitro*, SDR enzymes can frequently catalyze both oxidation and reduction, depending on the supplied co-factor. In cells, however, they show directionality, which depends on their sub-cellular localization and cofactor availability. We first determined whether other SDRs with the proper *in cellulo* directional polarity could activate other secondary alkynylcarbinol-containing compounds into cytotoxic species. We found that the AAC (*S*)–**4** (with a terminal C=CH$_2$ group, *Figure 5A*), despite its activity being strongly reduced in DAC-resistant clone A4 (~17-fold, *Figure 1—figure supplement 3B*), still retained some cytotoxicity (IC$_{50}$ ~2.6 µM) on these cells, possibly through bioactivation by a second dehydrogenase. To identify such a dehydrogenase, EMS was used to mutagenize the DACR clone #A4 once again. This mutagenized population was selected with a lethal concentration of 10 µM (*S*)–**4** and over twenty resistant clones (AACR clones) were isolated. Among those, we selected seven, which all showed similar increased resistance to (*S*)–**4** (*Figure 5A*), and submitted them to RNA-seq analysis. This allowed the identification of an average of six coding mutations in each clone (*Figure 5—figure supplement 1A*), and analysis revealed that the *RDH11* gene carried point mutations in five of these seven clones (*Figure 5—figure supplement 1A,B*). Further analysis revealed that the clone A6 contained two alleles of *RDH11*, each with a different point mutation (*Figure 5B*), and *RDH11* expression was severely downregulated in the clone B4 (*Figure 5—figure supplement 1C*). In conclusion, the *RDH11* gene was mutated (*Figure 5C*) or downregulated in all the AACR clones, implicating it as an SDR, with similar prodrug bioactivation capacity as HSD17B11, that mediates AAC (*S*)–**4** toxicity.

The human *RDH11* gene codes for retinol dehydrogenase 11 (also called SDR7C1 [*Persson et al., 2009*], PSDR1 or HSD17B15), a member of the SDR superfamily which also localizes to the ER (*Kedishvili et al., 2002*) where it uses NADP+ to catalyze the conversion of retinol (*Figure 5D*, preferentially the 11-*cis*, all-*trans* and 9-*cis* isomers) into retinal through oxidation of the C-15 carbinol center, with a pro-*R* hydrogen specificity (*Haeseleer et al., 2002*). Full-length RDH11 protein was not detectable in any of the AACR clones, suggesting that all the identified mis-sense mutations decrease RDH11 stability in a manner similar to HSD17B11 mutations (*Figure 5E*). To validate the ability of RDH11 to mediate some of the (*S*)–**4** cytotoxic effects, we compared its activity on U2OS, U2OS KO HSD17B11, and U2OS KO HSD17B11 cells in which RDH11 was inactivated using CRISPR/Cas9 (*Figure 5—figure supplement 2A*). In this panel, we confirmed that, while HSD17B11 was responsible for the nM activity of the AAC (*S*)–**4** (48-fold IC$_{50}$ increase as a result of HSD17B11 inactivation), the remaining toxicity of (*S*)–**4** in HSD17B11 KO cells was further decreased by inactivating RDH11 (sixfold IC$_{50}$ increase by RDH11 inactivation, *Figure 5—figure supplement 2B*). Comparison of the activity of (*S*)–**3** and (*S*)–**4** on U2OS KO [HSD17B11 + RDH11] complemented either by HSD17B11-GFP or RDH11 (*Figure 5—figure supplement 2C*) revealed that RDH11 bioactivates preferentially (*S*)–**4** (IC$_{50}$~0.28 µM, with a terminal C=CH$_2$ group) over (*S*)–**3** (IC$_{50}$~1.92 µM, with a terminal C≡CH group, *Figure 5—figure supplement 2D*), but also established that HSD17B11 was much more efficient than RDH11 for the bioactivation of either molecule (IC$_{50}$ < 40 nM).

As a proof-of-principle of the potential value of these discoveries for designing novel prodrugs with controlled cytotoxic activity, we attempted to design a new prodrug whose cytotoxic effects would depend more on the RDH11 SDR. We used WT U2OS or U2OS inactivated for HSD17B11, RDH11 or both (*Figure 5F*) to test several AAC analogues. Introduction of a second secondary carbinol function in the AAC structure, generating the AlkenylAlkynylDiCarbinol **12** (AADC), reduced its bioactivation by HSD17B11, while leaving intact its RDH11-dependent activation and thereby generating a new prodrug equally bioactivated by HSD17B11 and RDH11 (*Figure 5G*). As mixtures of diastereoisomers were used for this experiment, lower IC$_{50}$ can be expected with selected AADC enantiomers, especially with the external carbinol center in the RDH11-preferred (*S*) configuration.

Another SDR, HPGD, also called 15-PGDH, PGDH1 or SDR36C1 (*Persson et al., 2009*), is responsible for the inactivation of prostaglandins through oxidation of the (*S*)-C-15 carbinol center in a NAD+-dependent manner (*Figure 5H*; *Tai et al., 2006*). HPGD expression is tissue-restricted (prostate and bladder) and null in U2OS cells (*Broad Institute Cancer Cell Line Encyclopedia* RNA-seq dataset).

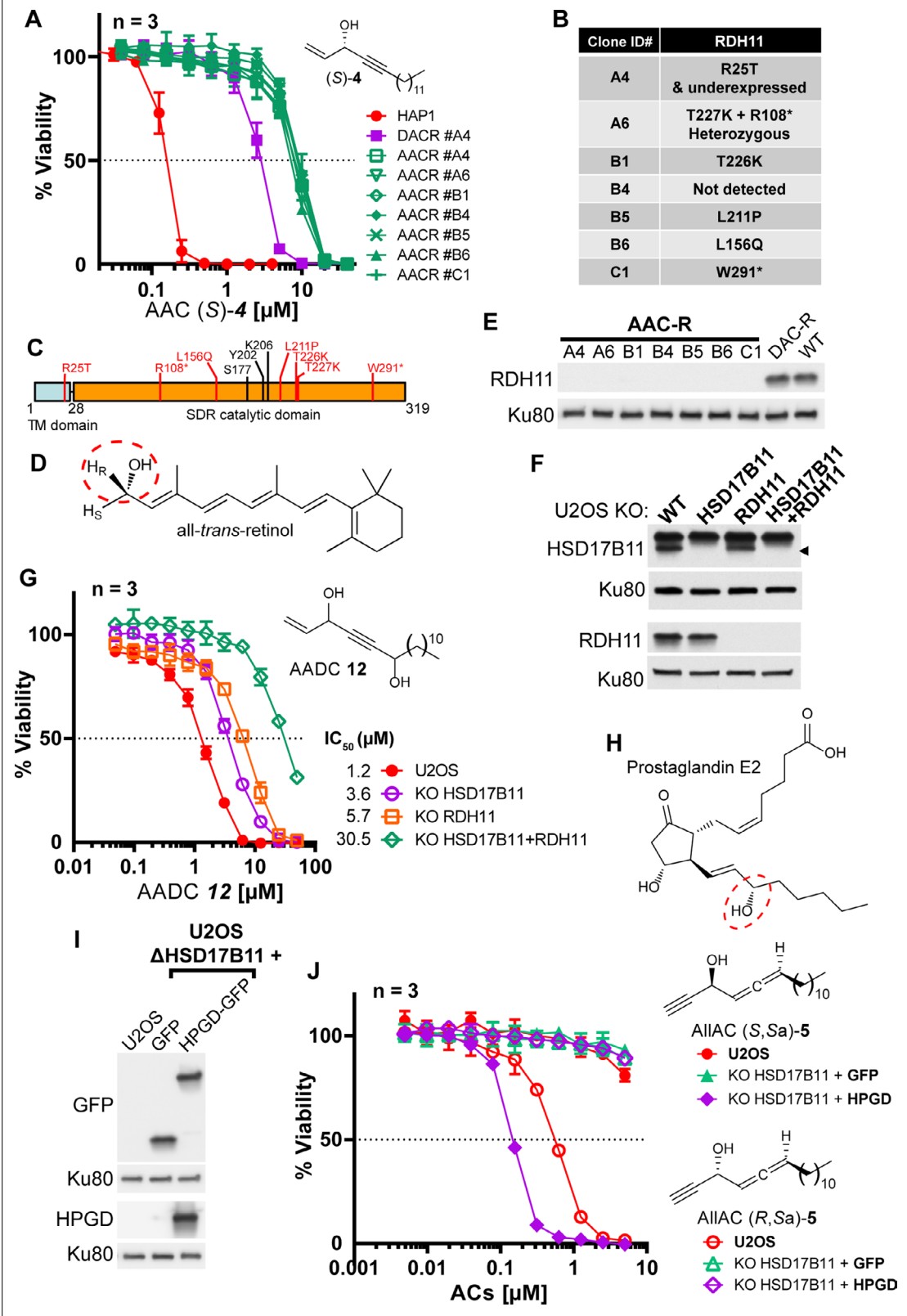

**Figure 5.** Bioactivation of other lipidic alkynylcarbinols by specific SDRs. (**A**) Cell viability analysis of wild-type HAP-1, DACR clone A4 and AACR clones treated with AAC (*S*)–**4**. (**B**) List of mutations identified on RDH11 by RNA-seq of individual AACR clones. (**C**) Schematic representation of RDH11 with, in red, the positions of the mutations identified and, in black, the three amino acids critical for catalysis. TM = single-pass transmembrane domain. (**D**) Structure of all-*trans*-retinol, a substrate for RDH11. (**E**) Analysis by immunoblotting of RDH11 levels in wild-type HAP-1, in DACR clone A4 and in the

*Figure 5 continued on next page*

*Figure 5 continued*

different AACR clones. (F) Analysis by immunoblotting of RDH11 and HSD17B11 levels in wild-type U2OS or clones inactivated for either HSD17B11, RDH11 or both. (G) Cell viability analysis of wild-type U2OS or U2OS clones inactivated for HSD17B11, RDH11, or both and treated with AADC **12**. (H) Structure of prostaglandin E2, a substrate of HPGD. (I) Analysis by immunoblotting of GFP and HPGD levels in WT U2OS or U2OS KO HSD17B11 stably complemented with GFP or HPGD-GFP. (J) Cell viability analysis of U2OS or U2OS inactivated for HSD17B11, stably complemented with either HSD17B11-GFP or HPGD-GFP and treated for 72 h with AllAC (*S*,*S*$_a$)- or (*R*,*S*$_a$)–**5**.

The online version of this article includes the following source data and figure supplement(s) for figure 5:

**Source data 1.** Source data related to *Figure 5E*.

**Source data 2.** Source data related to *Figure 5F*.

**Source data 3.** Source data related to *Figure 5I*.

**Figure supplement 1.** RDH11 is mutated or underexpressed in all AACR clones.

**Figure supplement 2.** RDH11 bioactivates preferentially *(S)*-AAC over *(S)*-DAC.

**Figure supplement 2—source data 1.** Source data related to *Figure 5—figure supplement 2A*.

**Figure supplement 2—source data 2.** Source data related to *Figure 5—figure supplement 2C*.

To determine whether this SDR could also bioactivate potential prodrugs, we used U2OS inactivated for HSD17B11 and overexpressing GFP alone or HPGD-GFP (*Figure 5I*), and screened a small collection of lipidic alkynylcarbinols. We found that the AllAC (*S*,*S*$_a$)–**5** was selectively bioactivated by HPGD, resulting in an IC$_{50}$ of ~147 nM in HPGD-overexpressing cells *vs* normal U2OS (*Figure 5J*). Thus, this mechanism of action is likely a general property of SDRs that could be exploited to develop a wide range of tailored prodrugs to cause selective cytotoxicity.

## Discussion

This study highlights the strengths of the genomic approach implemented here, relying on chemical mutagenesis of human haploid cells, selection with a toxic concentration and the use of RNA-seq to identify simultaneously changes in expression levels and mutations that could confer resistance to cytotoxic drugs. RNA-seq analysis of only four DAC (*S*)–**3**-resistant clones was sufficient to identify HSD17B11 as the DAC (*S*)–**3**-activating enzyme. Three clones carried non-sense mutations, including one on a key catalytic residue (S172, [*Gao et al., 2021*]), while the other ones showed lack of HSD17B11 expression. The strength of the approach is also reminiscent of the use of chemical mutagenesis. Indeed, while selection of drug-resistant clones based on the appearance of spontaneous mutations is frequently used in bacteria to identify the mechanism of action of small molecules, exploiting their high rate of mutations, human cells are less prone to spontaneous mutations. In human cells, screens based on spontaneous mutations most frequently use DNA repair (mismatch)-deficient cells, such as HCT-116, to increase the mutation rate (see for example *Girdler et al., 2008* and *Wacker et al., 2012*). However, spontaneous mutations can be diverse in nature and in that regard, the use of chemical mutagenesis with ethyl methanesulfonate (EMS), in addition to increasing the rate of resistant clones formation, simplifies the identification of the mutations of interest. Indeed, EMS does not induce deletions or insertion, but principally single nucleotide changes, most frequently transitions (*Forment et al., 2017*) which can be selected during the analysis, restricting the number of false positives. Another strength of this approach is the possibility to readily perform multiple rounds of screening to decipher primary and secondary mechanisms of action for a drug. Here, the DAC (*S*)–**3**-resistant clones could be mutagenized and selected a second time to identify a second SDR, RDH11, as the AAC (*S*)–**4**-bioactivating enzyme. It is noteworthy that the AAC (*S*)–**4** still shows some remaining toxicity on the DAC + AAC-resistant clones (HSD17B11 mutation +RDH11 mutation), which indicates that a third round of mutagenesis, selection and RNA-seq would most likely lead to identify another dehydrogenase, probably in the SDR superfamily, as responsible for the remaining bioactivation. This also illustrates how alkynylcarbinols could constitute a rich reservoir of SDR-specific prodrugs, whose selectivity can be improved through parallel structure-activity relationship studies. Finally, this approach is remarkably easier and cheaper to implement than loss of function CRIPSR/Cas9 screens and can lead to directly identify essential genes as mediators of drug cytotoxic effect. We recently exemplified this by using this approach to directly identify the DNA topoisomerase II alpha (coded by the essential *TOP2A* gene) as the main driver of the cytotoxic effects of CX-5461, a G-quadruplex

ligand and inhibitor of RNA polymerase I (*Bossaert et al., 2021*). The identified non-sense mutations on *TOP2A* dissociate its essential function from its function in generating DNA double-strand breaks upon G-quadruplex stabilization by CX-5461.

Through this powerful framework, our study reveals the original mode of action of a large family of natural and synthetic cytotoxic lipids characterized by a chiral terminal functional alkynylcarbinol pharmacophore. We show that these molecules are oxidized in an enantiospecific manner by a specific SDR, HSD17B11, converting them into an alkynylketone species (ynones). Oxidation of DACs produces dialkynylketones (DACones) that proved to be highly protein-reactive electrophiles, forming *Michael* adducts with cysteines and lysines. Consequently, bioactivated DACs modify several proteins in cells, resulting in their lipoxidation by a C17 lipidic chain (*Figure 3—figure supplement 1*). Lipoxidation of a protein can modify its solubility, folding, interactions, activity and/or localization (*Viedma-Poyatos et al., 2021*). In agreement, DAC treatment triggers the association of BRAT1, one of the most DAC-modified proteins, to ER and nuclear membranes (*Figure 3F*). Covalent modification of BRAT1 by the natural diterpene curcusone D was recently reported and resulted in BRAT1 degradation and reduced DNA damage response (*Cui et al., 2021*). Regarding (*S*)-DAC, in addition to BRAT1, several proteins involved in mechanisms of PQC are also lipoxidized. Considering that lipoxidation of a single protein can inhibit the UPS (*Shringarpure et al., 2000*), the simultaneous modification of several critical actors of PQC is likely to result in an acute proteotoxic stress. In agreement with the

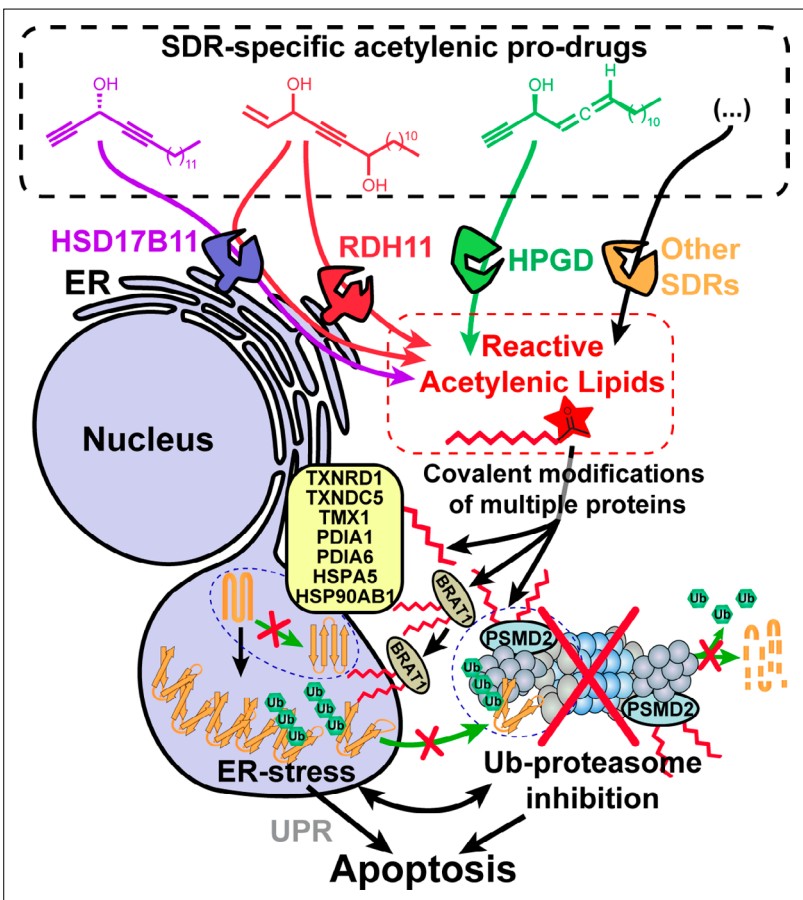

**Figure 6.** Model depicting the stereospecific bioactivation of alkynylcarbinol-containing compounds by specific SDRs into cytotoxic protein-reactive species. The protein reactive species generated upon bioactivation modify several proteins including the essential 26 S proteasome subunit PSMD2, thereby triggering Ub-proteasome system (UPS) inhibition, ER-stress, activation of the Unfolded Protein Response (UPR) and cell death mediated by apoptosis. See *Figure 6—figure supplement 1* for a graphical abstract depicting the main findings and techniques used.

The online version of this article includes the following figure supplement(s) for figure 6:

**Figure supplement 1.** Graphical abstract.

general effects of protein lipoxidation, DAC treatment triggers ER-stress and inhibition of the UPS (each one fueling the other, [*Menéndez-Benito et al., 2005*]). This leads to the early activation of the three UPR pathways (*Figure 4G*). In addition to unfolded proteins accumulation, UPR activation could also be the result of HSPA5/GRP78/BiP lipoxidation and/or tethering of multiple proteins to ER membranes. Finally, (*S*)-DACs also quickly induce mitochondrial fission and caspase-dependent apoptosis (*Figure 6*, see *Figure 6—figure supplement 1* for a graphical abstract). Hundreds of cytotoxic natural compounds have one or several alkynylcarbinol motifs, therefore the mechanism of action identified here could be shared in its principles by these molecules. This is supported by the fact that HSD17B11 was recently identified as mediating the toxicity of the natural compound dehydrofalcarindiol (*Grant et al., 2020*), produced in several plants. It is also noteworthy that two other related lipidic natural products, falcarinol and callyspongynic acid, isolated respectively from several *Apiaceae* plant species and the marine sponge *Callyspongia truncata*, have been identified as covalently binding to cellular proteins (*Heydenreuter et al., 2015*; *Nickel et al., 2015*), suggesting that they could also be bioactivated into protein-reactive species by a yet unidentified mechanism. Consequently, we propose that alkynylcarbinol-containing natural molecules provide a defense mechanism through bioactivation by specific SDRs in the body of predators, pathogens or parasites. In agreement, fulvindione, the AAC-oxidized form of the cytotoxic *Haliclona fulva*-produced fulvinol, was found in the body of the dorid nudibranch *Peltodoris atromaculata* feeding on *Haliclona* (*Ciavatta et al., 2014*). The presence of multiple alkynylcarbinol motifs in some of these natural compounds could provide modular pro-cytotoxic agents bioactivable by SDRs in different organisms. The fact that HSD17B11 mediates the toxicity of several natural compounds of unrelated origins questions the reasons for it being the common bioactivating enzyme of these prodrugs. The literature on HSD17B11 is currently limited. It was shown *in vitro* to promote androgen inactivation by converting the potent androstan-3-alpha,17-beta-diol into a weaker androgen (*Brereton et al., 2001*). In human, the HSD17B11 variant (rs9991501; Arg283Gln) was found associated with lean body mass (*Zillikens et al., 2017*), supporting that its activity controls muscle physiology, a process regulated by androgens. However, which HSD17B11-generated metabolites are involved and how the Arg283Gln variant impacts on HSD17B11 activity is currently unknown. HSD17B11 is nearly ubiquitously expressed (*Brereton et al., 2001*; *Chai et al., 2003*), but is also overexpressed in some human malignancies, including advanced prostate cancer and non-small cell lung cancer cell lines, as compared to normal tissues or cell lines (*Grundner-Culemann et al., 2016*; *Nakamura et al., 2009*). However, the potential role for HSD17B11 in cancer progression remains to be established. The HSD17B11-bioactivated clickable prodrugs described here could represent valuable tools to investigate HSD17B11 enzymatic activity in tissues through imaging and to decipher its physiological and therapeutic relevance.

Our work also provides a proof-of-concept that new pro-cytotoxic agents can be designed to be bioactivated through an enantiospecific oxidation catalyzed by selected SDRs. We exemplified this with three different human dehydrogenases, HSD17B11, RDH11 and HPGD (*Figure 6*). We also show here that multiple human cell lines derived from osteosarcoma, a rare pediatric cancer, were particularly sensitive to (*S*)-DACs (*Figure 1—figure supplement 4C*), suggesting that HSD17B11- and more generally SDR-bioactivable prodrugs could find anticancer applications, especially considering that 71 different SDRs are found in humans (*Bray et al., 2009*). For example, HSD3B1-specific pro-drugs could prove useful to treat castration-resistant prostate cancers, since the stabilized N367T variant of this SDR, found in ~30% of the population (rs1047303), has been associated to resistance to androgen deprivation therapies by allowing self-sufficient production of pro-proliferative androgens by prostate cancer cells (*Chang et al., 2013*; *Hearn et al., 2020*; *Thomas and Sharifi, 2020*). Exploring these potential applications *in vivo* will be the focus of future studies.

Of note, another family of diyne molecules, the enediynes, are already used in anticancer treatments in particular its best known representative, calicheamicin-gamma1 (cali), extracted from the soil bacteria *Micromonospora echinospora* (*Lee et al., 2002*). Cali shows a cytotoxic activity in the picomolar range that is exploited by two anticancer treatments, gemtuzumab ozogamicine (Mylotarg) and inotuzumab ozogamicin (Besponsa), consisting respectively of a CD33-cali or CD22-cali antibody-drug conjugates used for the treatment of acute myeloid leukemia (*Williams et al., 2019*). In the enediyne motif, the equivalent position of the DAC carbinol unit >CHOH is replaced by an ethylene unit -CH=CH-. Cali is also a prodrug which is activated by the intracellular reducing environment triggering the reduction, mainly carried by cellular glutathione, of a trisulfide unit to a thiol group,

initiating a cascade process by an intramolecular Michael addition leading to the DNA-damaging biradical species generated by Bergman cyclization. This, together with the binding of cali to the minor groove of DNA through its aryloligosaccharide moiety, allows the reaction of cali with both DNA strands by promoting hydrogen abstraction on the deoxyribose part of DNA, leading to cytotoxic DNA double-strand breaks (*Zein et al., 1988*).

The SDR superfamily is one of the largest, with representatives in all known life forms, except viruses: 507 673 SDRs are currently identified in the Uniprot database (in 7546 species, Pfam ID: PF00106). A bioinformatic classification of SDRs has revealed that among the 314 SDR subfamilies identified (*Kallberg et al., 2010*), half are specific to bacteria. This offers the prospect of designing antibiotics with novel mechanisms of action. In addition, other oxidoreductases, especially among the Medium-Chain dehydrogenase/Reductase (MDR) superfamily, could in principle carry out similar oxidation of secondary alcohol-containing substrates (*Persson et al., 2008*), which could widen the field of application of our findings. Conversely, some carbonyl reductases, especially among the Aldo-Keto Reductase (AKR) superfamily (*Oppermann, 2007*; *Penning et al., 2021*), could theoretically antagonize DACones cytotoxicity by reverting the reactive ketone into the corresponding alcohol, which could be further conjugated and eliminated. However, the fact that modulating glutathione levels or GST activity in cells only had a very small impact on (*S*)-DAC cytotoxicity (*Figure 3—figure supplement 3D,E*) indicates that, once produced, DACones immediately react with nearby proteins which limits the activity of general detoxification mechanisms.

Finally, our findings extend the toolbox of protein-reactive warheads with the identification of the DACone motif as a potent reactive group toward the thiol and amino groups of the side chains of cysteine and lysine, respectively. This reaction is quantitative, without co-product and operating in aqueous buffer at room temperature, making it well suited for many applications. In our study, multiple proteins could be functionalized by a terminal alkyne motif using clickable DACone **10** or **11** and the resulting linkage was found to be highly stable at neutral pH, while a selective cleavage of the DACone-amine linkage was observed at acidic pH (*Figure 2—figure supplement 1H,I*), a property that could be exploited. The reactive species could also be readily produced *in situ* by using the appropriate SDR with its co-factor (see *Figure 2F*). Our study also provides the first insights into the proteome-wide reactivity of these novel electrophiles using the isoDTB-ABPP approach (*Zanon et al., 2021*), which was also applied to demonstrate the formation of the DACone-protein adducts in cells (*Supplementary file 4A*). Reactive lysines and cysteines have been mapped on functional domains of human proteins (*Abbasov et al., 2021*; *Hacker et al., 2017*; *Weerapana et al., 2010*) and could be targeted through exploiting this new reactive entities supplied exogenously or generated *in situ*, thereby offering new avenues for protein functionalization or covalent inhibitors design.

## Materials and methods

### Plasmids

Detailed information regarding the plasmids generated in this study are provided in *Supplementary file 5A*. Plasmids generated in this study have been produced (i) by ligating annealed primers into BbsI digested plasmids (for cloning of sgRNAs) or (ii) by ligating digested PCR-amplified cDNAs into dephosphorylated plasmids. Phusion DNA polymerase, FastDigest restrictions enzymes and FastAP phosphatase were used (Thermo Fisher Scientific). For all plasmids, DNA sequencing (Mix2Seq, Eurofins Genomics) was used with the specified primers, to confirm that the desired sequence was inserted. All new plasmids have been deposited on Addgene. The plasmids ERmox-GFP (Addgene #68072, *Costantini et al., 2015*), Ub-G76V-YFP (Addgene #11949, *Menéndez-Benito et al., 2005*), mCherry-Mito-7 (Addgene #55102, *Olenych et al., 2007*), pEGFP-N1-ATG-FLAGC (Addgene #60360, *Britton et al., 2014*), pEGFP-C1-FLAGN (Addgene #46956, *Britton et al., 2013*), pICE-EGFP-FLAG-Ku70siR-WT (Addgene #46961, *Britton et al., 2013*), and pCAG-eSpCas9-2A-GFP (Addgene #79145) were provided by Addgene thanks to Addgene contributors.

### Oligonucleotides

DNA oligonucleotides used in the study are described in the *Supplementary file 5B* and were ordered from Eurofins Genomics.

## Cell lines and treatments

U2OS (ATCC), SAOS-2 (ATCC), 143B (Sigma-Aldrich), HOS (ECACC/Sigma-Aldrich), HS5 (ATCC), G292 clone 141B1 (ECACC/Sigma-Aldrich), HCT-116 (Horizon Discovery), A549 (ATCC), HT-1080 (ATCC), MDA-MB-436 (ATCC), SK-MEL-28 (ATCC), DLD-1 (ATCC), HEK293T (ATCC), MRC5-SV (ECACC/Sigma-Aldrich), HeLa (ATCC) and PC3 (ATCC) cells were grown in DMEM 10% FBS; CAPAN1 (ATCC) in IMDM 20% FBS; T47D (ATCC) and CAPAN2 (ATCC) cells in RPMI1640 10% FBS with Glutamax-I; HAP-1 (*Carette et al., 2011*) (Horizon Discovery) in IMDM 10% FBS and BJ-hTERT (gift from R. Weinberg, Whitehead Institute, Cambridge, USA) in DMEM 15% FBS 16% M199; human adult primary osteoblast (Cell Applications Inc, Sigma-Aldrich) in human osteoblast growth medium (Cell Applications Inc, Sigma-Aldrich). All cells media, except human osteoblast growth medium, contained penicillin and streptomycin (pen./strep.; Thermo Fisher Scientific) and cells were grown at 37 °C in 5% $CO_2$ humidified incubator. Cells were used at low passage and routinely confirmed free of Mycoplasma. Cells were treated in complete growth medium except when stated otherwise. Cycloheximide (C7698), ethacrynic acid (E1800000, GSTi), reduced glutathione monoethyl ester (G1404, GSHe), GSK-2606414 (516535, IRE1i), KIRA6 (5322810001, IRE1i), MG-132 (474790), SP600125 (420119, JNKi), STF-083010 (412510, PERKi) (all from Sigma-Aldrich) were added directly to cell media as indicated.

## Plasmid transfection and stable cell generation

Cells were transfected at 90% confluency in 60 mm dishes and using 5 µg DNA and lipofectamine 2000 (Thermo Fisher Scientific) following manufacturer's instructions. The day after transfection, cells were seeded at limiting dilution in 140 mm dishes. Selection of stable transfectants was performed either with 0.4 mg/mL G418 or 0.2 µg/mL puromycin. Individual clones were isolated. Homogeneous transgene expression was confirmed by monitoring cell fluorescence (for GFP tagged constructs) or by immunofluorescence.

## CRISPR/Cas9-mediated gene inactivation

Cells were transfected with pCAG-eSpCas9-2A-GFP plasmids coding for the *S. pyogenes* Cas9 K848A K1003A R1060A variant, which displays reduced off-target editing (*Slaymaker et al., 2016*), and co-expressing a guide against HSD17B11 or RDH11. One week after transfection, cells could be plated at limiting dilution to perform selection of individual clones, which were analysed for proper target inactivation. Rescue experiments were performed (with HSD17B11 or RDH11) to rule out off-target-related effects.

## Small-interfering RNA (siRNA)-mediated depletion

siRNA with a dTdT 3'extension were ordered from Eurofins Genomic against the following sequences: Control (Ctrl, target Firefly Luciferase) CGUACGCGGAAUACUUCGA, HSD17B11 #3 (ORF) CACA AGATCCTCAGATTGAAA, HSD17B11 #5 (3'-UTR) AACCGTTTATTTAACATATAT, PSMD2 #5 (ORF) TGGGTGTGTTCCGAAAGTTTA, PSMD2 #9 (3'-UTR) AAGGTTGTTCAATAAAGACTT. A total of 250,000 U2OS cells were seeded in sixwell plate the day before the first transfection. The day after, cells were transfected with 50 nM of each siRNA (Control, PSMD2 or HSD17B11) for 4–5 hr according to manufacturer's instruction before the medium being replaced by DMEM 10% FBS without antibiotics. A second transfection was performed the day after. Cells were seeded in 60 mm dishes the day before being used for experiments.

## Cell viability assays

Cell viability was analyzed using SulfoRhodamine B assays (SRB). Cells were seeded in 96-well plates 24 hr before being treated continuously for 72 hr with the indicated concentration of each molecule. For the experiments described in *Figure 2C*, cells were gently washed twice with PBS $Ca^{2+}/Mg^{2+}$ just before treatment (to remove residual media), treated with drugs for 1 hr in PBS containing $CaCl_2$ and $MgCl_2$ (to maintain cellular adhesion), then rinsed twice with complete medium (to remove residual drug), followed by a 72 hr post-incubation in complete culture medium. For analysis, cells were fixed for 1 hr at 4 °C by addition of cold trichloroacetic acid at a 3.33% final concentration. After being washed four times with water and dried, cells were stained by a 30 min incubation in a solution of 0.057% (wt:vol) SRB in 1% acetic acid. The wells were washed four times with 1% acetic acid, dried and the dye was resuspended by a 2 hr incubation in a 10 mM Tris-Base solution. Absorbance at 490 nm

of each well was measured (μQuant plate reader, Bio-tek) and used as a readout of cell number. For calculation, background absorbance was subtracted to each value and the data were normalized to the value measured in untreated wells. Each point was measured in duplicate and the graphs correspond to at least three independent experiments. $IC_{50}$ were computed with the GraphPad Prism software using a non-linear regression to a four-parameter logistic curve (log[inhibitor] *vs* response; variable slope). The error bars represent standard deviation (SD).

## Analysis of RNA expression levels in published dataset

The data visualization tool Ordino (*Streit et al., 2019*) was used to compare the RNA expression levels of selected genes in The Cancer Cell Line Encyclopedia RNA-seq dataset (http://www.broadinstitute.org/ccle; *Barretina et al., 2012*).

## Mutagenesis and selection with (*S*)-3 and (*S*)-4

$100.10^6$ haploid HAP1 cells at 60% confluency were treated for 72 hr with 0.3 mg/mL ethyl methanesulfonate (EMS, Sigma-Aldrich) directly added to the cell medium. After recovery, two 140 mm dishes at $10^6$ cells/dish were seeded from this mutagenized population and selected by treatment with 0.25 μM DAC (*S*)–**3** for 72 h. After treatment, the medium was refreshed and, after 2–3 weeks, individual clones were isolated (DACR clones). To isolate AACR clones, the DACR clone #A4 at early passage was mutagenized again (0.3 mg/ml EMS for 72 h). After recovery, the mutagenized DACR population was seeded into 140 mm dishes ($10^6$ cells/dish) and selected by treatment with 10 μM (*S*)–**4** for 72 h. After treatment, the medium was refreshed and, after 2–3 weeks, individual clones were isolated (AACR clones).

## RNA-seq

RNA-seq was performed at the GeT-PlaGe core facility, INRA Toulouse, from total RNA prepared with the RNeasy Plus Mini Kit (Qiagen) according to the manufacturer's instructions. RNA-seq libraries were prepared according to Illumina's protocols using the Illumina TruSeq Stranded mRNA sample prep kit. Briefly, mRNAs were selected using poly-dT beads. Then, RNAs were fragmented and adaptors ligated. Eleven cycles of PCR were applied for library amplification. Library quality was assessed using a Fragment Analyzer System (Agilent) and libraries were quantified by Q-PCR using the Kapa Library Quantification Kit (Roche). RNA-seq experiments were performed on an Illumina HiSeq3000 using a paired-end read length of 2 × 150 pb. RNA-seq data have been deposited on SRA (Bioproject IDs PRJNA668246 & PRJNA668322).

## RNA-Seq alignment and SNP prediction and filtering

Read quality was confirmed within the ng6 environment (*Mariette et al., 2012*) using fastQC (http://www.bioinformatics.babraham.ac.uk/projects/fastqc/) and Burrows-Wheeler Aligner BWA (*Li and Durbin, 2009*) to search for contamination. The reads were cleaned with cutadapt v1.8.3 and aligned against hg38 reference human genome with STAR v2.5.2b (*Dobin et al., 2013*). Expression levels were computed with featureCounts (*Liao et al., 2014*) using Ensembl annotation. Alignments were deduplicated with samtools rmdup and reads not uniquely mapped removed. Then GATK v3.5 base quality score recalibration was applied (*McKenna et al., 2010*). Indel realignment, SNP and INDEL discovery were performed with HaplotypeCaller using standard hard filtering parameters according to GATK best practices recommendations for RNAseq. Finally variants were annotated using snpEff v4.3t (*Cingolani et al., 2012*). A python script was used to select protein coding variants specific to resistant clones as compared to the parental HAP-1 (wild-type for DACR clones, and DACR#A4 for AACR clones) with a minimal allele frequency of 0.9 and a depth greater than 10 reads. Among these variants, were selected the ones that resulted in frameshifts, mis- and non-sense mutations as compared to the reference human genome hg38. Cytoscape v3.2.0 (*Shannon et al., 2003*) was used to identify genes found mutated in several clones and to generate a graphical overview.

## Targeted sequencing of HSD17B11 cDNA from HAP-1 clones

Total RNAs were extracted from wild-type or DACR HAP-1 with the RNeasy Plus Mini Kit (Qiagen) according to the manufacturer's instructions. HSD17B11 cDNA was produced from these RNAs with the Superscript III First-Strand kit (Thermo Fisher Scientific) according to the manufacturer's

instructions and using the HSD17B11-RNA-Rv primer. The resulting HSD17B11 cDNAs was amplified using the primer pair HSD17B11-RNA-Fw – HSD17B11-RNA-Rv and sequenced using the primers HSD17B11-SEQ-F1 and HSD17B11-SEQ-R1 (Eurofins Genomics).

## Antibodies

For immunoblotting, horse-radish peroxidase-conjugated goat anti-mouse or anti-rabbit secondary antibodies (Jackson Immunoresearch Laboratories), or IRDye800CW-conjugated donkey anti-mouse or anti-rabbit secondary antibodies (LI-COR Biosciences) were used, diluted at 1/10,000 in PBS 0.1% Tween-20. For immunofluorescence, AlexaFluor488- or AlexaFluor594-conjugated goat anti-mouse or anti-rabbit antibodies (Thermo Fisher Scientific) were used diluted at 1/1000 in blocking buffer. A list of primary antibodies used in this study, together with related information is provided in *Supplementary file 5C*.

## Live imaging

Pictures of living cells were acquired using an Olympus IX73 fluorescence microscope fitted with a 40 X objective (0.75NA UPlanFLN, Olympus) or a 20 × 0.40 NA objective (LCAChN 0.4NA, Olympus), a X-Cite Series 120Q lamp (Lumen dynamics), a DP26 camera (Olympus) and using the adequate filters set. For time series, cells were seeded in glass-bottom dishes (from MatTek or ibidi μSlide) in phenol red-free Leibovitz's L-15 medium containing 10% FBS and pen./strep. For each time point, z-stacks were acquired using a Andor/Olympus Yokogawa CSU-X1 confocal spinning disk fitted with 60 X (UPLSAPO NA 1.35, Olympus) or 100 X (UPLSAPO NA 1.4, Olympus) objectives, a Andor iXon Life 888 EM-CCD camera and with temperature and humidity control. The white scale bars on representative pictures represent 10 μm.

## Immunofluorescence

Cells were seeded on glass coverslips (#1.5 thickness; ~ 170 μm, VWR). At the end of the treatment, the cells were washed twice with PBS, fixed by a 15 min incubation with 2% paraformaldehyde (PFA) in PBS and washed three times. The cells were then permeabilized 5 min with 0.2% Triton X-100 in PBS and washed three times with PBS. The coverslips were incubated 10 min in blocking buffer consisting in PBS 0.1% Tween-20 (PBS-T) containing 5% bovine serum albumin (BSA). The coverslips were incubated for 75 min with the primary antibodies diluted in blocking buffer (mouse anti-γH2AX antibody at 1/1000 and rabbit anti-53BP1 at 1/800), washed four times in PBS-T and then incubated 45 min with the secondary antibodies diluted in blocking buffer, washed four times in PBS-T and twice in PBS, incubated 15 min with 2 μg/mL DAPI (4',6-diamidino-2-phenylindole) in PBS, washed twice with PBS, dipped in double-distilled water and mounted in VectaShield on a glass slide. Pictures were acquired using an Olympus IX73 microscope fitted with a 40 x UPlanFLN objective (Olympus), a X-Cite Series 120Q lamp (Lumen dynamics), a DP26 camera (Olympus) and using the adequate filters set. The white scale bars on each picture represent 10 μm.

## Immunoblotting

For whole-cell extracts (WCE), cells were washed with cold PBS and scrapped in 75 μL SDS-lysis buffer (120 mM Tris-HCl pH 6.8, 20% glycerol, 4% SDS), incubated 5 min at 95 °C and passed 10 times through a 25 G needle. Measuring the absorbance at 280 nm with a Nanododrop spectrometer (Thermo Fisher Scientific) was used to evaluate protein concentration and, after adjustment with SDS-Lysis buffer, extracts were diluted by addition of equal volume of SDS-Loading Buffer (5 mM Tris pH 6.8, 0.01% bromophenol blue, 0.2 M dithiothreitol). Immunoblotting was performed with 25–50 μg of WCE. Proteins were separated on gradient gels (BioRad 4–12% TGX pre-cast gels) and transferred onto Protran 0.45 μm nitrocellulose membranes (GE Healthcare). After transfer, membranes could be scanned with an infrared imager (Odyssey, LI-Cor Biosciences) to acquire the signal from AlexaFluor647 modified proteins. Homogeneous loading and transfer were checked by Ponceau S staining. When necessary, membranes were cut into horizontal strips to simultaneously probe for multiple proteins. For immunoblotting, membranes were blocked with PBS containing 5% non-fat dry cow milk, washed and incubated for 1–2 hr at room temperature or 16 hr at 4 °C with primary antibodies diluted in PBS-T containing 1% bovine serum albumin (immunoglobulin- and lipid-free fraction V BSA, Sigma-Aldrich). After extensive washes, membranes were probed 1 hr at room temperature with adequate secondary

antibodies coupled with horse-radish peroxidase (HRP) or with IRDye800CW. For HRP-conjugated secondary antibodies, signal acquisition was performed with a CCD camera (Chemidoc, BioRad) or using autoradiographic films (Blue Devil, Genesee Scientific) after incubation with peroxidase chemi-luminescent substrates (BioRad Clarity ECL for CCD acquisition; Advansta WesternBright ECL for autoradiographic film exposure). For IRDye800CW-coupled secondary antibodies, membranes were scanned using an infrared imager (Odyssey, LI-Cor Biosciences). For SDS-PAGE, PageRuler Prestained Protein Ladder or PageRuler Plus Prestained Protein Ladder (Invitrogen) were used.

## Analysis by SDS-PAGE of proteins modified by DACs in cells

Sub-confluent 140 mm dishes, seeded two days before with U2OS or HAP-1 cells, were treated for 2 hr with 2 µM of DAC. At the end of treatment, cells were collected by trypsination and centrifugation (900 rotations per minute (RPM), 4 °C, 5 min). The cell pellet was washed with cold PBS before being lysed by sonication on ice (Vibracell, Bioblock Scientific, ten 2s-pulses, of amplitude 30) in 400 µL of IPL buffer (20 mM Tris-HCl pH 7.8, 1 mM EDTA, 150 mM NaCl, 0.5% IGEPAL CA-630, HALT proteases and phosphatases inhibitor cocktail (Thermo Fisher Scientific)). A centrifugation (15,000 RPM, 4 °C, 4 min) was used to remove insoluble material and the supernatant was used for click reactions. The click reaction was performed by incubation at 20 °C for 30 min of a mix containing 240 µg of proteins (diluted in 10 µL IPL buffer), 4 mM $CuSO_4$, 2 µM azido-AlexaFluor647 and 10 mM sodium ascorbate in IPD buffer (20 mM Tris-HCl pH 7.8, 1 mM EDTA, 150 mM NaCl, 0.05% IGEPAL CA-630, HALT inhibitors cocktail) to reach 80 µL. 20 µL of 5XLoading Buffer (300 mM Tris-HCl pH 6.8, 5% SDS, 0.025% bromophenol blue, 15% glycerol, 250 mM dithiothreitol) was added at the end of the reaction, followed by incubation at 95 °C for 5 min. Twenty-four µg of proteins were separated on SDS-PAGE gels (BioRad 4–15% TGX pre-cast gels), followed by transfer onto Protran 0.45 µm nitro-cellulose membranes (GE Healthcare) which were scanned on an infrared imager (Odyssey, LI-COR Biosciences). A ponceau S staining was used to control for homogeneous loading.

## Immunoprecipitation (IP)

For IP, U2OS and HAP-1 parental or overexpressing GFP-tagged protein cell lines were seeded in 140 mm dishes 2 days before treatment. Sub-confluent cells were treated (or not) with the indicated molecules for 2 hr. At the end of treatment, cells were collected by trypsination and centrifugation (900 RPM, 5 min, 4 °C). The cell pellet was washed with cold PBS before being lysed by sonication on ice (Vibracell, Bioblock Scientific, ten 2s-pulses, of amplitude 30) in 400 µL of IPL buffer (20 mM Tris-HCl pH 7.8, 1 mM EDTA, 150 mM NaCl, 0.5% IGEPAL CA-630, HALT proteases and phosphatases inhibitor cocktail (Thermo Fisher Scientific)). A centrifugation (15,000 RPM, 4 °C, 4 min) was used to remove insoluble material and the supernatant was used for IP. IP were performed by incubation of lysates 4 hr at 4 °C on a rotating wheel with either 50 µL of DynaBeads M-280 protein A magnetic beads (Thermo Fisher Scientific), pre-loaded with 8 µg of rabbit control (Dako) or anti-PSMD2 (Bethyl Laboratories) antibodies, or with 50 µL of anti-GFP magnetic beads (GFP-Trap, Chromotek). Each IP was done on 240 µg of proteins diluted in 220 µL of IPL buffer to which 480 µL of IPD buffer was added (to dilute IGEPAL CA-630 to ~0.15%). On-bead protein modification by DACones could be performed at that stage (see below). Then the beads were washed 3 times with high-salt IPW buffer (20 mM Tris-HCl pH 7.8, 1 mM EDTA, 500 mM NaCl, 0.05% IGEPAL CA-630, HALT inhibitors cocktail) and once with IPD buffer (with 500 µL for each wash).

## On-bead protein modification by DACones

After IP, beads were washed twice in MoD buffer (10 mM phosphate pH 7.4, 2.7 mM KCl, 137 mM NaCl, 0.05% IGEPAL CA-630, HALT proteases and phosphatases inhibitor cocktail). Reactions with DACones were performed by incubating the beads 30 min at 30 °C with vigorous intermittent shaking in 400 µL of MoD buffer containing 1 µM DAC (*S*)–**9** or DACone **10**. Beads were then washed with IPW buffer as described above and clicked as described below.

## On-bead click with azido-AlexaFluor647

Immunopurified proteins were clicked on beads by a 30-min incubation at 20 °C with vigorous inter-mittent shaking in 200 µL of IPD buffer containing 4 mM $CuSO_4$, 5 µM azido-AlexaFluor647 and 10 mM sodium ascorbate. The beads were washed with IPD buffer and resuspended in 20 µL SDS-Lysis Buffer

to which 20 µL of SDS Loading buffer was added. Beads were incubated 5 min at 95 °C in this solution and 20 µL of supernatant was analyzed on gradient gels (BioRad 4–12% TGX pre-cast gels). The modified proteins were detected with an infrared imager (Odyssey, LI-COR Bioscience) after transfer onto a nitrocellulose membrane (Protran, 45 µm pores, GE Healthcare) and total proteins on the membrane were stained with Ponceau S.

### *In vitro* protein modification by DACones for analysis by SDS-PAGE

FBS (Euromedex), bovine serum albumin (BSA, A-7030, Sigma-Aldrich), bovine carbonic anhydrase (CANH, C-3934, Sigma-Aldrich), Jack bean concanavalin A (ConcA, C-2010, Sigma-Aldrich), bovine beta-lactoglobulin (BLG, L-3908, Sigma-Aldrich) were resuspended at 2 mg/mL in MoD buffer. Reactions were performed for 40 min at 30 °C in 50 µL of MoD buffer containing 20 µg of protein and 2 µM of DAC or DACone. The reactions were then diluted to 75 µL by sequential addition of IPD buffer, CuSO₄, azido-AlexaFluor647 and sodium ascorbate to a final concentration of 4 mM, 4 µM and 10 mM, respectively and incubated 30 min at 20 °C with vigorous intermittent shaking. At that stage, unclicked azido-AlexaFluor647 could be removed using BioRad MicroBioSpin P-6 columns equilibrated with SDS-Lysis buffer and following manufacturer's instructions. Then, 8 µL of the click reaction medium were supplemented with 12 µL of SDS-Lysis buffer and 20 µL of SDS Loading buffer, incubated 5 min at 95 °C and separated on gradient gels (BioRad 4–12% TGX pre-cast gels). The modified proteins could be detected with an infrared imager (Odyssey, LI-COR Biosciences) directly in the gel or after transfer onto a nitrocellulose membrane. Total proteins in the gel or on the membrane were visualized using Coomassie (Instant*Blue*, Sigma-Aldrich, scanned with the Odyssey or BioRad Chemidoc imagers) or Ponceau S staining, respectively.

### *In vitro* DAC bioactivation assays

HSD17B11 was immunopurified from UO2S KO HSD17B11 complemented either with HSD17B11-GFP wild-type or the S172L mutant. After extensive washes with a buffer containing 500 mM NaCl, the magnetic beads were used as a source of HSD17B11 enzyme. HSD17B11 is a membrane-anchored protein (*Horiguchi et al., 2008a*) and its activity required maintaining a minimal 0.2% IGEPAL CA-630 concentration in all the buffers. In details, after IP, 30 µL of beads were washed twice with MoD buffer and incubated 30 min at 30 °C with vigorous intermittent shaking in 50 µL of MoD buffer containing 0.2% IGEPAL CA-630, 1 mM β-NAD+, 2 µM (*S*)- or (*R*)–**9** and 40 µg of beta-lactoglobulin (BLG, L-3908, Sigma-Aldrich). The reactions were then diluted to 75 µL by sequential addition of MoD buffer, CuSO₄, azido-AlexaFluor647 and sodium ascorbate to final concentrations of 4 mM, 3 µM and 10 mM, respectively. The click reaction was performed by incubation 30 min at 20 °C with vigorous intermittent shaking. The supernatants, containing the BLG, and the beads, carrying HSD17B11-GFP, were analyzed separately by SDS-PAGE and transferred onto nitrocellulose membranes (Protran, 45 µm pores, GE Healthcare). DACone adducts onto BLG and HSD17B11 were visualized by scanning the membranes with an infrared imager (Odyssey, LI-COR Biosciences), total protein stained with Ponceau S and HSD17B11-GFP levels analyzed by anti-GFP immunoblotting.

### Molecular docking

A model of the human HSD17B11 dimer was generated using CollabFold to run AlphaFold2-multimer with MMseqs2 and HHsearch (*Evans et al., 2021*; *Jumper et al., 2021*; *Mirdita et al., 2021*). The NAD+ co-factor was added to the model by alignment with the crystallographic structure of rat HSD17B10 in complex with NAD (PDB:1E6W, *Powell et al., 2000*). The HSD17B11-NAD+ complex was then minimized with Chimera ("minimize structure" tools with the AMBER parameter and 100 cycles of steepest descent and 10 cycles of conjugate gradient). The docking study was carried out using SeeSAR (version 11.2, Bio SolveI T GmbH) for characterization of the binding pocket and subsequent docking of (*S*)–**3** and (*R*)–**3**. To analyze the selectivity of the binding towards HSD17B11 as compared to the other 17β-HSDs, the DAC (*S*)–**3** and (*R*)–**3** were docked as described above on the deposited AlphaFold2 models for each of the other human 17-betaHSD SDRs (IDs: AF-P14061-F1, AF-P37059-F1, AF-P37058-F1, AF-P51659-F1, AF-O14756-F1, AF-P56937-F1, AF-Q92506-F1, AF-Q92781-F1, AF-Q99714-F1, AF-Q8NBQ5-F1, AF-Q53GQ0-F1, AF-Q7Z5P4-F1, AF-Q9BPX1-F1). The number of conformers filtered reaching favorable lipophilic ligand efficiency, ligand efficiency and affinity was scored and used to assess the selectivity (*Figure 2—figure supplement 2F*).

## MS analysis of entire β-lactoglobulin modified by DACone

Commercially available purified BLG (mixture of isoforms A and B) was incubated for 40 min at 30 °C in 40 µL of MoD buffer containing 30 µM BLG and 100 µM DACone **10** (or the same volume of acetonitrile as control, 10% final concentration). Prior to MS analysis, unmodified and modified BLG samples were desalted in 200 mM ammonium acetate, pH 7 using BioRad Micro Bio-Spin 6 devices and diluted to ~4 µM in 50% acetonitrile plus 0.2% formic acid final. Samples were analyzed on a SYNAPT G2-Si mass spectrometer (Waters, Manchester, UK) running in positive ion mode and coupled to an automated chip-based nano-electrospray source (Triversa Nanomate, Advion Biosciences, Ithaca, NY, USA). The voltage applied to the chip and the cone voltage were set to 1.6 kV and 150 V, respectively. The instrument was calibrated with a 2 mg/mL cesium iodide solution in 50% isopropanol. Raw data were acquired with MassLynx 4.1 (Waters, Manchester, UK) and deconvoluted with UniDec using the following parameters: m/z range: 1300–2800 Th; subtract curved: 1; Gaussian smoothing: 10; bin every 1 Th; charge range: 5–15; mass range: 18,000–19,000 Da; sample mass: every 1 Da; peak FWHM: 1 Th; peak detection range: 50 Da, and peak detection threshold: 0.1. The mass spectrometry proteomics data have been deposited to the ProteomeXchange Consortium via the PRIDE (*Perez-Riverol et al., 2019*) partner repository with the data set identifier PXD033059.

## isoDTB-ABBP-based framework for assessing the proteome-wide selectivity of DACones

Total cell extracts from U2OS cells were prepared by sonicating in 5 mL of PBS 0.1% IGEPAL CA-630 containing protease and phosphatase inhibitors (HALT cocktail, Pierce) a pellet of 1.5 mL of U2OS cells, collected by scrapping in cold PBS, followed by centrifugation 30 min at 15,000 RPM at 4 °C. Protein concentration in the supernatant was evaluated using a Nanodrop device (Thermo Fisher Scientific) and the extracts diluted at 1 mg/mL with PBS containing protease and phosphatase inhibitors. 1 mL of extracts was incubated for 40 min at 30 °C with 100 µM of DACone **10** or **11** or acetonitrile (1% final, solvent control). Proteins were then precipitated overnight at –20 °C by addition of 4 volumes of cold acetone followed by centrifugation at 16,000 g for 15 min at 4 °C. The precipitates were resuspended in 1 mL cold MeOH by sonication and centrifuged (10 min, 21100 × g, 4 °C). The supernatant was removed and the washing step with MeOH was repeated once. The pellets were dissolved in 1 mL 0.8% SDS in PBS by sonication. Duplicates were performed for each condition, one being clicked with the heavy isoDTB tag, the other one with the light tag. The click reaction, all subsequent experimental and analytical steps were performed as described (*Zanon et al., 2021*). The data presented correspond to two independent experiments (Exp.1 and 2 in *Supplementary files 1 and 2*). PSSM sequence logos were generated by analyzing the 10 amino acids surrounding the modified site using Seq2Logo (*Thomsen and Nielsen, 2012*). The mass spectrometry proteomics data have been deposited to the ProteomeXchange Consortium via the PRIDE (*Perez-Riverol et al., 2019*) partner repository with the data set identifier PXD033059.

## Reaction of DACone with purified proteins for analysis of the absorbance spectrum

Reactions were performed by incubation at 30 °C for 40 min in 30 µL of MoD buffer containing 115 µg of protein (equivalent to 0.1 mM BLG and 0.035 mM BSA) and 0.3 mM of DAC or DACone. 2 µL of each reaction were used to acquire a UV-visible (190–840 nm) absorbance spectrum on a Nanodrop device (Thermo Fisher Scientific).

## Reaction of DACones with NAG, NAC, and NAK for analysis of the absorbance spectrum

Reactions were performed by incubation at 30 °C for 40 min in 40 µL of a mixture containing 1 mM NAG, NAC, or NAK and 1 mM DACone **8** in 20 mM phosphate buffer (pH 7.4) or 10 mM KOH (pH 10). A UV-visible (190–840 nm) absorbance spectrum for each reaction was acquired using a Nanodrop spectrometer (Thermo Fisher Scientific).

## NMR characterization of DACone reaction products with $N_\alpha$-acetyl Lysine, *N*-acetyl Cysteine and reduced glutathione

Reactions were performed by incubating at 30 °C for 40 min a 300 µL mixture containing 20 mM DACone **8**, 20 mM NAC, 20 mM NAK and 20 mM reduced glutathione and 10 mM KOH (pH 10) or 20 mM phosphate buffer (pH 7.2). Each reaction was characterized by high-resolution mass spectrometry. After lyophilization, the samples were dissolved at a concentration of 10 mM in PBS buffer pH 7.4 with 10% $D_2O$ containing DSS at 1 µM. All compounds were fully characterized by [1]H and [13]C NMR spectroscopy. NMR spectra were recorded at 298 K on Avance III HD 700 spectrometer ([1]H: 700.13 MHz, [13]C: 176.04 MHz). [1]H NMR spectra were recorded with water suppression. The correlation [1]H spectroscopy was acquired with the Bruker pulse sequence dipsiesgp using an excitation sculpting sequence for water suppression. The recycle delay was set to 1.5 s. [1]H-[13]C HSQC experiment (Bruker pulse sequence hsqcphpr) was recorded with carrier frequencies set to 4.7 ppm ([1]H) and 85 ppm ([13]C) and the spectral widths were set to 12 ppm ([1]H) and 180 ppm ([13]C). For this experiment, the recycle delay was set to 1 s. Spectral data are provided in *Figure 2—figure supplement 4*, *Figure 3—figure supplement 3* and bellow.

## Spectral data corresponding to the NMR characterization of DACone reaction products with $N_\alpha$-Acetyl Lysine, N-Acetyl Cysteine and reduced glutathione

**$N_\alpha$-Acetyl-L-Lysine:** [1]H-NMR (700 MHz, $H_2O$): $\delta$ [ppm] 4.13 (dd, 1 H, Hα, $J_{H\alpha,H\beta}$ = 8.5 Hz); 2.79 (t, 1 H, Hε); 2.02 (s, 3 H, $CH_3$); 1.78 (m, 1 H, Hβ); 1.67 (m, 1 H, Hβ); 1.55 (m, 2H, Hδ); 1.37 (m, 2H, Hγ). [13]C NMR (700 MHz, $H_2O$): $\delta$ [ppm] 57.94 Cα; 42.83 Cε; 24.70 $C_{CH3}$; 34.05 Cβ; 34.02 Cβ; 31.85 Cδ; 25.11 Cγ.

**$N_\alpha$-Acetyl L-Lysine modified by DACone 8:**[1]H-NMR (700 MHz, $H_2O$): $\delta$ [ppm] 8.03 (d, 1 H, *J* = 13.52 Hz); 7.93 (d, 1 H, *J* = 12.81 Hz); 5.46 (d, 1 H, *J* = 13.54 Hz); 4.13 (m, 1 H, Hα); 3.35 (t, 1 H, Hε); 3.22 (t, 1 H, Hε); 2.42 (t, 2 H, H4); 2.00 H (s, 1 H, $CH_3$); 1.80 (m, 1 H, Hβ); 1.66 (m, 1 H, Hβ); 1.61 (m, 2 H, Hδ); 1.56 (m, 2 H, H3); 1.42 (m, 2 H, H2); 1.38 (m, 2 H, Hγ); 0.90 (t, 3 H, H1). [13]C NMR (700 MHz, $H_2O$): $\delta$ [ppm] 162.79 $C_9$; 166.44 $C_9$; 102.38 $C_8$; 57.71 Cα; 51.45 Cε; 45.95 Cε; 32.08 $C_4$; 24.5 $C_{CH3}$; 33.93 Cβ; 33.90 Cβ; 29.70 Cδ; 32.08 $C_3$; 24.28 $C_2$; 25.04 Cγ; 15.42 $C_1$.

**N-Acetyl-L-Cysteine:** [1]H-NMR (700 MHz, $H_2O$): $\delta$ [ppm] 4.37 (t, 1 H, Hα); 2.91 (d, 1 H, Hβ); 2.90 (d, 1 H, Hβ); 2.05 (s, 3 H, $CH_3$). [13]C NMR (700 MHz, $H_2O$): δ [ppm] 57.94 Cα; 42.83 Cε; 24.70 $C_{CH3}$; 34.05 Cβ; 34.02 Cβ; 31.85 C δ; 25.11 Cγ.

**N-Acetyl-L-Cysteine modified by DACone 8:** [1]H-NMR (700 MHz, $H_2O$): $\delta$ [ppm] 8.23 (d, 1 H (0.25), $J_{H,H}$ = 15.20 Hz, H8); 7.59 (d, 1 H (0.75), $J_{H,H}$ = 9.98 Hz, H8); 6.51 (d, 1 H (0.75), $J_{H,H}$ = 9.98 Hz,H9); 6.34 (d, 1 H (0.25), $J_{H,H}$ = 15.20 Hz, H9); 4.53 (dd, 1 H (0.25), Hα); 4.46 (dd, 1 H (0.75), Hα); 3.45 (dd, 1 H (0.25), Hβ); 3.37 (dd, 1 H (0.75), Hβ); 3.29 (dd, 1 H (0.25), Hβ); 3.19 (dd, 1 H (0.75), Hβ); 2.49 (t, 2 H (0.25), H4); 2.45 (t, 2 H (0.75), H4); 2.02 (s, 3 H, $CH_3$); 1.59 (q, 2 H (0.25), H3); 1.56 (q, 2 H (0.75), H3); 1.43 (sx, 2 H (0.25), H2); 1.42 (sx, 2 H (0.75), H2); 0.90 (t, 3 H (0.25), H1); 0.89 (t, 3 H (0.75), H1). [13]C NMR (700 MHz, $H_2O$): $\delta$ [ppm] 161.24 C1; 157.68 C8;125.06 C9; 127.65 C9; 56.79 Cα; 57.57 Cα; 38.00 Cβ; 41.81 Cβ; 38.02 Cβ; 41.79 Cβ; 20.74 C4; 20.77 C4; 24.61 $C_{CH3}$; 31.78 C3; 24.12 C2; 15.36 C1.

**Reduced glutathione:** [1]H-NMR (700 MHz, $H_2O$) and [13]C-NMR: $\delta$ [ppm] 8.22 H (t, 1 H, $H_N$, $J_{H,H}$ = 6.02 Hz); 3.77 H (m, 2 H, Hα); 46.13 Cα; 8.45 H (d, 1 H, $J_{H,H}$ = 7.51 Hz, $H_N$); 4.56 H (q, 1 H, Hα); 2.95 H (dd, 2 H, Hβ); 58.36 Cα; 28.36 Cβ; 3.77 (dd, 1 H, Hα); 2.15 H (m, 2 H, Hβ); 2.54 H (m, 2 H, Hγ); 57.02 Cα; 28.87 Cβ; 34.11 Cγ.

**Reduced glutathione modified by DACone 8:** [1]H-NMR (700 MHz, $H_2O$) and [13]C-NMR: δ [ppm] 8.26 H (t, 1 H, $H_N$, $J_{H,H}$ = 6.02 Hz); 3.76 H (m, 2 H, Hα); 46.13 Cα; 8.65 H (d, 1 H (0.40), $H_N$, $J_{H,H}$ = 7.51 Hz); 8.60 H (d, 1 H (0.60), $H_N$, $J_{H,H}$ = 7.51 Hz); 4.76 H (q, 1 H (0.40), Hα); 4.71 H (q, 1 H (0.60), Hα); 3.48 H (dd, 1 H (0.40), Hβ A); 3.42 H (dd, 1 H (0.60), Hβ A); 3.31 H (dd, 1 H (0.40), Hβ B); 3.18 H (dd, 1 H (0.60), Hβ B); 56.20 Cα; 40.58 Cβ;36.47 Cβ; 3.73 H (dd, 1 H, Hα); 2.12 H (m, 2 H, Hβ); 2.51 H (m, 2 H, Hγ); 57.02 Cα; 28.97 Cβ; 34.11 Cγ; 8.19 H8 (d, 1 H (0.40), $J_{H,H}$ = 15.44 Hz); 7.59 H8 (d, 1 H (0.60), $J_{H,H}$ = 10.13 Hz); 6.54 H9 (d, 1 H (0.60), $J_{H,H}$ = 10.13 Hz); 6.35 H9 (d, 1 H (0.40), $J_{H,H}$ = 15.44 Hz); 1.56 H3 (m, 2 H, H3); 1.42 H2 (m, 2 H, H2); 0.89 H (t, 3 H, H1). [13]C NMR (700 MHz, $H_2O$): $\delta$ [ppm] 159.55 C8; 156.68 C8; 125.34 C9; 127.91 C9; 20.77 C4; 31.75 C3; 24.12 C2; 15.51 C1.

## Identification of proteins modified by DAC (*S*)-9 in cells

Two 140 mm dishes, seeded with $2.5 \times 10^6$ U2OS cells 2 days before, were treated for 2 h with 2 µM of DAC (*S*)–**9** or (*R*)–**9**. At the end of treatment, cells were collected by trypsination and centrifugation (900 RPM, 4 °C, 5 min). The cell pellet was washed with cold PBS before being lysed by sonication on ice (Vibracell, Bioblock Scientific, ten 2 s-pulses, of amplitude 30) in 240 µL of IPL buffer (20 mM Tris-HCl pH 7.8, 1 mM EDTA, 150 mM NaCl, 0.5% IGEPAL CA-630, HALT proteases and phosphatases inhibitor cocktail (Thermo Fisher Scientific)). A centrifugation (15,000 RPM, 4 °C, 4 min) was used to remove insoluble material and the supernatant was used for click pull-down. A volume of lysate corresponding to 300 µg of proteins was diluted to 500 µL with IPD buffer (20 mM Tris-HCl pH 7.8, 1 mM EDTA, 150 mM NaCl, 0.05% IGEPAL CA-630, HALT inhibitors cocktail) bringing IGEPAL CA-630 concentration to 0.2–0.25%. The lysate was pre-cleared by incubation with 60 µL streptavidin-coupled magnetic beads (Dynabeads M-280, Thermo Fisher Scientific) 1 h at 4 °C on a rotating wheel. Then CuSO$_4$, biotin-PEG-azido (PEG4 carboxamide-6-azidohexanyl biotin, B10184, Thermo Fisher Scientific) and sodium ascorbate were added to the pre-cleared lysate at 4 mM, 5 µM and 10 mM final concentration, respectively. The click-reaction was performed by incubation for 30 min at 20 °C in the dark with vigorous intermittent shaking and the free biotin was removed from the clicked extracts using PD MiniTrap G-25 column (GE Healthcare) equilibrated with IPD buffer. The extracts were then complemented to 600 µL with IPD buffer and incubated for 2 h at 4 °C on a rotating wheel with 60 µL of streptavidin-coupled magnetic beads (Dynabeads M-280, Thermo Fisher Scientific). The beads, corresponding to the proteins associated to (*S*)- or (*R*)–**9**, were then washed extensively with high-salt IPW buffer (20 mM Tris-HCl pH 7.8, 1 mM EDTA, 500 mM NaCl, 0.05% IGEPAL CA-630, HALT inhibitors cocktail), then with IPD buffer. For on-bead trypsic digestion, beads were washed twice with 50 mM ammonium bicarbonate buffer, and then suspended in 7 M urea and 25 mM DTT (Sigma-Aldrich). After 60 min under agitation (850 rpm) at room temperature, the samples were alkylated by incubation in 90 mM iodoacetamide (Sigma-Aldrich) during 30 min in the dark. Samples were then washed twice as described above and submitted to overnight proteolysis in ammonium bicarbonate buffer containing 1 µg of trypsin (Promega) per sample at 37 °C. The supernatants were collected, dried in speed-vac and resuspended with 2% acetonitrile and 0.05% trifluoroacetic acid (Sigma-Aldrich), for mass spectrometry analysis. The resulting peptides were analyzed with a NanoLC (Ultimate 3000 RSLCnano system Thermo Scientific) coupled to a LTQ Orbitrap Velos mass spectrometer (Thermo Fisher Scientific, Bremen, Germany). Raw MS files were processed with MaxQuant v1.5.2.8 software for database search with the Andromeda search engine and quantitative analysis. Data were searched against human entries in the Swissprot protein database. To perform relative quantification between proteins identified, we used the LFQ from the MaxQuant 'protein group.txt' output. The experiment was repeated three times (exp. #1, #2, and #3 in *Supplementary file 3*). The mass spectrometry proteomics data have been deposited to the ProteomeXchange Consortium via the PRIDE (*Perez-Riverol et al., 2019*) partner repository with the data set identifier PXD033059. For each protein identified, the ratio (*S*)–**9** *vs* (*R*)–**9** conditions was computed (fold change, FC) and a Student's t-test was used to select proteins reproductively enriched (FC >2 and p < 0.05) in the (*S*)–**9** *vs* (*R*)–**9** condition. Cytoscape (*Shannon et al., 2003*) was used to generate *Figure 3B* visual representation, in which the FC is color-coded, while the -log(p) was used to define protein box size (*id* large box means highly significant enrichment).

## Identification of peptides modified by DAC (*S*)-9 in cells using isoDTB tags

Total cell extracts were prepared by sonicating, in 2 mL of PBS 0.5% IGEPAL CA-630 containing protease and phosphatase inhibitors (HALT cocktail, Pierce), a pellet of ~300 µL of U2OS cells treated for 2 hr with 2 µM DAC (*S*)–**9**, collected by scrapping in cold PBS, followed by centrifugation 30 min at 15,000 RPM at 4 °C. Protein concentration in the supernatant was evaluated using a Nanodrop device (Thermo Fisher Scientific) and the extracts diluted at 1 mg/mL with PBS containing protease and phosphatase inhibitors. One mg of proteins were then precipitated overnight at –20 °C by addition of 4 volumes of cold acetone followed by centrifugation at 16,000 g for 15 min at 4 °C. The precipitates were further processed as described for the detection of the DACone-amino acid selectivity. The data presented correspond to two independent experiments (Exp.1 and 2 in *Supplementary file 4*). The

mass spectrometry proteomics data have been deposited to the ProteomeXchange Consortium via the PRIDE (*Perez-Riverol et al., 2019*) partner repository with the data set identifier PXD033059.

## Click-based imaging, GFP fluorescence and immunofluorescence imaging, pre-extraction

Cells were seeded on #1.5 glass coverslips (VWR) the day before the experiment. For labelling mitochondria with MitoTracker, MitoTracker Red CMXRos (Thermo Fisher Scientific) was added at 0.2 µM in complete medium 30 min before the end of the treatments. To monitor the association of GFP-BRAT1 to insoluble compartments, cells were pre-extracted at the end of the treatment by a 2 min incubation on ice in cold PBS 0.1% Triton X-100 containing HALT proteases/phosphatase inhibitors cocktail. At the end of treatments or after pre-extraction, cells were washed twice with PBS, fixed 15 min with 2% PFA in PBS and washed three times with PBS. Cells were permeabilized by incubation 8 min in PBS 0.2% Triton X-100 before being washed three times with PBS. When co-staining with antibody was performed, cells were incubated 10 min in IF blocking buffer (PBS-T 5% BSA), before being incubated 75 min with AlexaFluor594-coupled anti-COXIV antibody diluted at 1:50 in IF blocking buffer. For co-staining of ER membranes with concanavalin A, fixed cells were incubated 30 min AlexaFluor488-coupled concanavalin A diluted at 100 µg/mL in blocking buffer. At the end of antibody/concanavalin staining, cells were washed four times, fixed for 15 min, washed three times with PBS and incubated in blocking buffer. Click with AlexaFluor488-azido or AlexaFluor594-azido was performed as described (*Rozié et al., 2018*). At the end of the procedure, cells were washed four times with PBS-T, twice with PBS and incubated 15 min in PBS containing 2 µg/mL DAPI (Sigma-Aldrich). The coverslips were washed twice with PBS and mounted with VectaShield (Vector laboratories) on glass slides. Images were acquired on a Deltavision PersonalDV microscope (Applied Precision, 1024 × 1024 CoolSNAP HQ, z-stack of 0.2 µm interval) equipped with a 100 x UPlanSApo/1.40 oil objective (Olympus) or with a Zeiss Elyra 7 3D Lattice SIM super-resolution microscope fitted with a 63 X objective (PLANAPO NA 1.4, Zeiss) and dual sCMOS cameras (pco.edge). Deconvolutions were then performed with SoftWoRx (Applied Precision) in conservative mode while 3D-SIM reconstruction were performed with Zen Black (Zeiss). The white scale bars on representative pictures represent 10 µm.

## Flow cytometry

U2OS stably expressing the UPS fluorescent reporter Ub-G76V-YFP (*Menéndez-Benito et al., 2005*) were treated for 4 hr with 20 µM MG132 or 1 µM (*S*)–**3**. At the end of the treatment, cells were collected by trypsination, washed in PBS 1% BSA and fixed by incubation at room temperature in 500 µL of 2% PFA in PBS. Cells were washed with PBS 1% BSA and stored in the same buffer. A minimum of 30,000 cells were analyzed on BD LSR II flow cytometer (Becton Dickinson). Data were analyzed and formatted using FlowJo v8.8.7. Untreated cells were used to define a gate to identify the YFP-positive cells in the treated conditions.

## Synthesis

Synthesis and characterization of (*S*)–**1** (*El Arfaoui et al., 2013*), (*R*)–**1** (*El Arfaoui et al., 2013*), (*R*)–**2** (*El Arfaoui et al., 2013*), (*S*)–**3** (*El Arfaoui et al., 2013*), (*R*)–**3** (*El Arfaoui et al., 2013*), (*S*)–**4** (*El Arfaoui et al., 2013*),(*R,Sa*)–**5** (*Listunov et al., 2018a*), (*S, Sa*)–**5** (*Listunov et al., 2018a*), (*S*)–**6** (*Bourkhis et al., 2018*), (*S*)–**9** (*Listunov et al., 2015b*), (*R*)–**9** (*Listunov et al., 2015b*) have been described previously. Enantiomeric (ee) and diasteromeric (de) excesses of aforementioned compounds are the following: (*S*)–**1** (90% ee), (*R*)–**1** (80% ee), (*R*)–**2** (80% ee), (*S*)–**3** (91% ee), (*R*)–**3** (93% ee), (*S*)–**4** (91% ee), (*R,Sa*)–**5** (98% ee, 91% de), (*S,Sa*)–**5** (89% ee, 74% de), (*S*)–**6** (89% ee), (*S*)–**9** (97% ee), (*R*)–**9** (73% ee).

## Synthesis of novel compounds

All reagents were obtained from commercial suppliers and used without any further purification. Reactions were run under nitrogen or argon atmosphere in oven-dried glassware. Standard inert atmosphere techniques were used in handling air and moisture sensitive reagents. Dichloromethane (CH$_2$Cl$_2$) and tetrahydrofuran (THF) were obtained by filtration through a drying column on a filtration system. Thin-layer chromatography analyses were performed on precoated, aluminum-backed silica gel (Merck 60 F254). Visualization of the developed chromatogram was performed by UV light (254 nm) and using aqueous potassium permanganate (KMnO$_4$) stain. Flash column chromatography

was performed using flash silica gel (SDS 35–70 mm or 60 Å, C.C 70–200 μm). Nuclear magnetic resonance spectra were recorded on Bruker Advance 300 or 400 MHz spectrometers. Chemical shifts for $^1$H NMR spectra are quoted in parts per million relative to residual solvent peak. Data are reported as follows: chemical shift, multiplicity (s = singlet, d = doublet, t = triplet, q = quartet, qn = quintet, m = multiplet), coupling constant in Hz and integration. Chemical shifts for $^{13}$C NMR spectra are quoted in parts per million relative to residual solvent peak. All $^{13}$C NMR spectra were obtained with complete proton decoupling. Infrared analyses were run on a Perkin-Elmer Spectrum 100 FT-IR spectrometer or a Thermo-Nicolet Diamond ATR (4 cm$^{-1}$ of resolution, 16 scans) equipped with a DTGS detector and are reported in reciprocal centimeters (cm$^{-1}$). High-resolution mass spectrometry (HRMS) was performed on a Thermo-Finnigan MAT 95 XL instrument.

## Synthesis of heptadeca-1,4-diyn-3-one (7)

To a solution of the previously described racemic dialkynylcarbinol **3** (42.5 mg, 0.17 mmol) in CH$_2$Cl$_2$ (4.0 mL) was added Dess-Martin periodinane (109 mg, 0.26 mmol, 1.5 eq) at RT. The reaction mixture was stirred until completion (TLC monitoring). After 1.5 hr, water (2 mL) was slowly added to the solution. Layers were separated and the aqueous layer was extracted three times with CH$_2$Cl$_2$. The combined organic layers were dried over MgSO$_4$ and concentrated under reduced pressure at room temperature. Flash chromatography on silica (gradient elution up to 10% Et$_2$O in pentane) of the crude mixture afforded DACone **7** (42 mg, 0.17 mmol, quant. yield) as a slightly yellow oil. **$^1$H NMR** (300 MHz, CD$_3$CN) δ 3.79 (s, 1 H), 2.45 (t, $J$ = 7.0 Hz, 2 H), 1.65–1.50 (m, 2 H), 1.45–1.20 (m, 18 H), 0.88 (t, $J$ = 6.7 Hz, 3 H). **$^{13}$C NMR** (75 MHz, CD$_3$CN) δ: 161.2, 98.2, 82.8, 82.4, 80.3, 32.6, 30.3, 30.2, 30.1, 30.0, 29.6, 29.4, 28.1, 23.4, 19.4, 14.4. **HRMS-DCI** (CH$_4$): $m/z$ calcd for C$_{17}$H$_{27}$O [M]$^+$: 247.2062, found: 247.2060. **FTIR** (cm$^{-1}$) (neat): $\nu$ 2925, 2855, 2223, 2098, 1636, 1,211. DACone **7** was dissolved in CH$_3$CN at 1 mM. See **Supplementary file 6A** for **7** full spectra.

## Synthesis of 1-[tris(propan-2-yl)silyl]-nona-1,4-diyn-3-ol (CBD513)

To a solution of 1-hexyne (0.273 mL, 2.97 mmol, 1 eq.) in THF (5 mL) under stirring at 0 °C was added $n$-butyllithium (2.5 M solution in hexane, 1.19 mL, 2.97 mmol, 1 eq.). The solution was stirred for 10 min at 0 °C and then 30 min at RT before treatment with 3-[tris(propan-2-yl)silyl]-prop-2-ynal (625 mg, 2.97 mmol, 1 eq.) at 0 °C. The mixture was stirred 10 min at 0 °C and then overnight at RT. After treatment with a saturated aqueous NH$_4$Cl solution, the aqueous layer was extracted three times with Et$_2$O. The combined organic layers were washed with brine, dried over MgSO$_4$ and concentrated under reduced pressure. The residue was purified by silica gel column chromatography (cyclohexane/Et$_2$O 9:1) to give **CBD513** (480 mg, 1.64 mmol, 55%) as a colorless oil. **$^1$H NMR** (300 MHz, CDCl$_3$) δ 5.11 (s, 1 H), 2.25 (td, $J$ = 6.8 Hz, 2.1 Hz, 2 H), 1.40–1.54 (m, 4 H), 1.1 (s, 21 H), 0.93 (t, $J$ = 7.2 Hz, 3 H). **$^{13}$C NMR** (75 MHz, CDCl$_3$): δ 104.8, 85.3, 85.1, 77.7, 52.8, 30.3, 21.8, 18.5, 18.3, 13.5, 11.1. **HRMS-DCI** (CH$_4$): $m/z$ calcd for C$_{18}$H$_{33}$OSi [M + H]$^+$: 293.2290, found: 293.2301. See **Supplementary file 6B** for **CBD513** full spectra.

## Synthesis of nona-1,4-diyn-3-ol (CBD515)

To a solution of alcohol **CBD513** (480 mg, 1.64 mmol) in THF (50 mL) under stirring at 0 °C was added dropwise tetra-$n$-butylammonium fluoride (1 M in THF, 4.92 mL, 4.92 mmol). Then, the mixture was stirred for 90 min at room temperature (RT) before addition of a saturated aqueous NH$_4$Cl solution. The aqueous layer was extracted with Et$_2$O. The combined organic layers were washed with brine, dried over MgSO$_4$ and concentrated under reduced pressure. The residue was purified by silica gel column chromatography (pentane/Et$_2$O 95:5) to give **CBD515** (180 mg, 80%) as a colorless oil. **$^1$H NMR** (300 MHz, CDCl$_3$) δ 5.12 (q, $J$ = 2.1 Hz, 1 H), 2.56 (d, $J$ = 2.3 Hz, 1 H), 2.25 (td, $J$ = 7 Hz, 2.1 Hz, 2 H) 1.34–1.64 (m, 4 H), 0.9 (t, $J$ = 7.2 Hz, 3 H). **$^{13}$C NMR** (100 MHz, CDCl$_3$): δ 96.0, 81.5, 77.0, 72.1, 52.1, 21.9, 18.3, 17.7, 13.5. See **Supplementary file 6C** for **CBD515** full spectra.

## Synthesis of nona-1,4-diyn-3-one (8)

To a solution of diynol **CBD515** (163 mg, 1.2 mmol) in CH$_2$Cl$_2$ (30 mL) was added Dess-Martin periodinane (0.3 M in CH$_2$Cl$_2$, 8 mL, 2.4 mmol) at RT and the resulting mixture was stirred for 4 hr. The reaction was quenched by addition of a saturated aqueous NH$_4$Cl solution. The aqueous layer was extracted with CH$_2$Cl$_2$. The combined organic layers were washed with brine, dried over MgSO$_4$ and

concentrated under reduced pressure. The residue was purified by silica gel column chromatography (pentane/ Et$_2$O 9:1) to give **8** (75 mg, 46%) as a colorless oil. **$^1$H NMR** (300 MHz, CDCl$_3$) $\delta$ 3.28 (s, 1 H), 2.44 (t, $J$ = 7 Hz, 2 H), 1.55–1.72 (m, 2 H), 1.39–1.55 (m, 2 H), 0.95 (t, $J$ = 7.2 Hz 3 H). **$^{13}$C NMR** (75 MHz, CDCl$_3$) $\delta$ 160.4, 96.7, 82.2, 81.9, 77.9, 29.4, 21.9, 18.8, 13.4. **HRMS-DCI** (CH$_4$) m/z calcd for C$_9$H$_{11}$O, [M + H]$^+$: 135.0810, found: 135.0806. See *Supplementary file 6D* for **8** full spectra.

## Synthesis of heptadeca-1,4,16-triyn-3-one (10)

To a solution of the previously described racemic clickable dialkynylcarbinol **9** (30.0 mg, 0.12 mmol) in CH$_2$Cl$_2$ (2.9 mL) was added Dess-Martin periodinane (78 mg, 0.18 mmol, 1.5 eq) at RT. The reaction mixture was stirred until completion (TLC monitoring). After 80 min, water (1.5 mL) was slowly added to the solution. Layers were separated and the aqueous layer was extracted three times with CH$_2$Cl$_2$. The combined organic layers were dried over MgSO$_4$ and concentrated under reduced pressure at RT. Flash chromatography on silica (gradient elution up to 15% Et$_2$O in pentane) of the crude mixture afforded clickable DACone **10 (**26.5 mg, 0.12 mmol, 89% yield) as a colorless oil. **$^1$H NMR** (300 MHz, CD$_3$CN) $\delta$ 3.80 (s, 1 H), 2.45 (t, $J$ = 7.0 Hz, 2 H), 2.20–2.10 (m, 3 H), 1.65–1.52 (m, 2 H), 1.52–1.25 (m, 14 H). **$^{13}$C NMR** (75 MHz, CD$_3$CN) $\delta$ 161.2, 98.2, 85.5, 82.8, 82.4, 80.3, 69.5, 30.1, 30.0, 29.7, 29.6, 29.4, 29.4, 29.3, 28.1, 19.4, 18.7. **HRMS-DCI** (CH$_4$): m/z calcd for C$_{17}$H$_{23}$O [M + H]$^+$: 243.1749, found: 243.1761. **FTIR** (cm$^{-1}$) (neat): $\nu$ 3288, 2923, 2851, 2214, 2100, 1629, 1,207. Clickable DACone **10** was dissolved in CH$_3$CN at 1 and 20 mM. See *Supplementary file 6E* for 1**0** full spectra.

## Synthesis of 1-[tris(propan-2-yl)silyl]-deca-1,4,9-triyn-3-ol (CBD514)

To a solution of hepta-1,6-diyne (0.342 mL, 3 mmol) in THF (5 mL) under stirring at –78 °C was added dropwise n-butyllithium (2.5 M in hexane, 1.2 mL, 3 mmol). The solution was stirred for 1 hour at –78 °C and 30 min at RT before treatment with 3-[tris(propan-2-yl)silyl]-prop-2-ynal (631 mg, 3 mmol). The mixture was allowed to warm slowly up to RT and stirred overnight. After treatment with a saturated aqueous NH$_4$Cl solution, the aqueous layer was extracted three times with Et$_2$O. The combined organic layers were washed with brine, dried over MgSO$_4$ and concentrated under reduced pressure. The residue was purified by silica gel column chromatography (cyclohexane/Et$_2$O 99:1) to give **CBD514** (490 mg, 54%) as a colorless oil. **$^1$H NMR** (400 MHz, CDCl$_3$) d 5.11 (s, 1 H), 2.39 (td, $J$ = 7.0, 2.1 Hz, 2 H,), 2.33 (td, $J$ = 7.0, 2.7 Hz, 2 H), 1.97 (t, J = 2.7 Hz, 1 H), 1.76 (qn, $J$ = 7.0 Hz, 2 H), 1.10 (s, 21 H). **$^{13}$C NMR** (75 MHz, CDCl$_3$) $\delta$ 104.6, 85.5, 84.0, 83.3, 78.5, 68.9, 52.7, 27.2, 18.5, 17.7, 17.4, 11.1. See *Supplementary file 6F* for **CBD514** full spectra.

## Synthesis of deca-1,4,9-triyn-3-ol (CBD516)

To a solution of **CBD514** (490 mg, 1.60 mmol) in THF (50 mL) under stirring at 0 °C was added dropwise tetra-n-butylammonium fluoride (1 M in THF, 4.8 mL, 4.8 mmol). Then, the mixture was stirred for 2 h at RT before addition of a saturated aqueous NH$_4$Cl solution. The aqueous layer was extracted with Et$_2$O. The combined organic layers were washed with brine, dried with MgSO$_4$ and concentrated under reduced pressure. The residue was purified by silica gel column chromatography (pentane/Et$_2$O 9:1) to give **CBD516** (130 mg, 47%) as a colorless oil. **$^1$H NMR** (300 MHz, CDCl$_3$) $\delta$ 5.12 (q, $J$ = 2.1 Hz, 1 H), 2.57 (d, $J$ = 2.3 Hz, 1 H), 2.40 (td, $J$ = 7.1, 2.1 Hz, 2 H), 2.33 (td, $J$ = 7.0, 2.7 Hz, 2 H), 1.99 (t, $J$ = 2.7 Hz, 1 H), 1.68–1.85 (m, 2 H). **$^{13}$C NMR** (75 MHz, CDCl$_3$) $\delta$ 84.7, 83.3, 81.3, 77.7, 72.3, 69.0, 52.1, 27.1, 17.7, 17.5. **HRMS-DCI** (CH$_4$) m/z calcd for C$_{10}$H$_8$ [M-H$_2$O]$^+$: 128.0626, found: 128.0623. See *Supplementary file 6G* for **CBD516** full spectra.

## Synthesis of deca-1,4,9-triyn-3-one (11)

To a solution of **CBD516** (130 mg, 0.74 mmol) in CH$_2$Cl$_2$ (30 mL) was added Dess-Martin periodinane (0.3 M in CH$_2$Cl$_2$, 3.7 mL, 1.12 mmol) at RT and the resulting mixture was stirred for 4 hr. The reaction was quenched by addition of a saturated aqueous NH$_4$Cl solution. The aqueous layer was extracted with CH$_2$Cl$_2$. The combined organic layers were washed with brine, dried with MgSO$_4$ and concentrated under reduced pressure. The residue was purified by silica gel column chromatography (pentane/Et$_2$O 9:1) to give **11** (65 mg, 50%) as a colorless oil. **$^1$H NMR** (300 MHz, CDCl$_3$) $\delta$ 3.3 (s, 1 H), 2.60 (t, $J$ = 7.1 Hz, 2 H), 2.37 (td, $J$ = 6.9, 2.7 Hz, 2 H), 2.02 (t, $J$ = 2.6 Hz, 1 H), 1.86 (qn, $J$ = 7.1 Hz, 2 H). **$^{13}$C NMR** (75 MHz, CDCl$_3$) $\delta$ 160.1, 95.2, 82.5, 82.2, 82.1, 78.2, 69.6, 26.3, 18.1, 17.6. **HRMS-DCI**

$(CH_4)$ $m/z$ calcd for $C_{10}H_9O$, $[M + H]^+$: 145.0653, found: 145.0649. See *Supplementary file 6H* for **11** full spectra.

## Synthesis of *tert*-butyldimethyl(tetradec-1-yn-3-yloxy)silane (MVB39)

Imidazole (310 mg, 4.56 mmol), DMAP (19.9 mg, 0.163 mmol,) and TBSCl (737 mg 4.89 mmol) were added to a solution of **MVB36** (*Shiina et al., 2012*) (3.26 mmol, 686 mg) in anhydrous $CH_2Cl_2$ (10 mL) at RT. After stirring overnight, the reaction mixture was quenched with $H_2O$ (10 mL) and the aqueous layer was extracted with $CH_2Cl_2$ (3 × 10 mL). The combined organic layers were dried over anhydrous $MgSO_4$ and concentrated under reduced pressure. The residue was purified by silica gel column chromatography (petroleum ether) to give *tert*-butyldimethyl(tetradec-1-yn-3-yloxy)silane (655 mg, 62%) as a colourless oil. **¹H NMR** (300 MHz, $CDCl_3$) $\delta$ 4.33 (td, $J$ = 6.5, 2.1 Hz, 1 H), 2.37 (d, $J$ = 2.1 Hz, 1 H), 1.70–1.63 (m, 2 H), 1.46–1.36 (m, 2 H), 1.34–1.20 (s, 16 H), 0.91 (s, 9 H), 0.88 (t, $J$ = 6.9 Hz, 3 H), 0.13 and 0.11 (2 s, 2 × 3 H). **¹³C NMR** (75 MHz, $CDCl_3$) $\delta$ 86.0, 72.0, 62.9, 38.7, 32.1, 29.8, 29.8, 29.7, 29.7, 29.5, 29.4, 25.9 (3 CH₃), 25.3, 22.8, 18.4, 14.3,–4.4, –4.9. **HRMS-DCI** ($CH_4$): $m/z$ $[M + H]^+$: calcd for $C_{20}H_{41}OSi$: 325.2927 found: 325.2918. **FTIR** (cm⁻¹) (neat): $v$ 3312, 2924, 2854, 1463, 1250, 1094, 835, 776, 652, 626, 559. See *Supplementary file 6I* for **MVB39** full spectra.

## Synthesis of 6-((*tert*-butyldimethylsilyl)oxy)heptadec-1-en-4-yn-3-ol (MVBL05)

To a solution of **MVB39** (117 mg, 0.361 mmol,) in THF (8 mL) at –78 °C, *n*-butyllithium solution (2.5 M in hexane, 159 µL, 0.40 mmol) was added dropwise. After 30 min, a solution of acrolein (80.7 mg, 1.44 mmol) in THF (0.5 mL) was added dropwise. The reaction was warmed up to RT and maintained under stirring overnight. It was quenched by the addition of saturated $NH_4Cl$ aqueous solution (5 mL) and extracted with EtOAc (3 × 8 mL). The combined organic layers were dried over anhydrous $MgSO_4$, and concentrated under reduced pressure. The residue was purified by silica gel column chromatography (pentane/EtOAc 25:1) to give 6-((*tert*-butyldimethylsilyl)oxy)heptadec-1-en-4-yn-3-ol (38 mg, 27%). Mixture of diastereoisomers **¹H NMR** (400 MHz, $CDCl_3$): $\delta$ 5.96 (ddd, $J$ = 17.0, 10.2, 5.3 Hz, 1 H), 5.45 (dd, $J$ = 17.0, 1.3 Hz, 1 H), 5.21 (d, $J$ = 10.2 Hz, 1 H), 4.95–4.85 (m, 1 H) 4.39 (td, $J$ = 6.5, 1.6 Hz, 1 H), 1.75–1.60 (m, 2 H), 1.47–1.35 (m, 2 H), 1.35–1.25 (m, 16 H), 0.90 (s, 9 H), 0.85 (t, $J$ = 6.9 Hz, 3 H), 0.12 and 0.10 (2 s, 2 × 3 H). **¹³C NMR** (100 MHz, $CDCl_3$): $\delta$ 137.15/137.13, 116.48/116.46, 88.59/88.56, 82.64/82.60, 63.4, 63.1, 38.7, 32.1, 29.9, 29.8, 29.8, 29.7, 29.7, 29.5, 29.4, 26.0, 25.4, 22.8, 18.4, 14.3,–4.3, –4.8. **HRMS-DCI** ($CH_4$): $m/z$ calcd for $C_{23}H_{45}O_2Si$ $[M + H]^+$: 381.3189, found: 381.3186. **FTIR** (cm⁻¹) (neat): $v$ 3343, 2924, 2854, 1463, 1251, 1092, 984, 927, 835, 776. See *Supplementary file 6J* for **MVBL05** full spectra.

## Synthesis of heptadec-1-en-4-yne-3,6-diol (12)

To a stirring solution of **MVBL05** (0.10 mmol, 38 mg) in THF (5 mL) at 0 °C, was added a solution of tetrabutylammonium fluoride (1 M in THF, 198 µL, 0.198 mmol). The mixture was stirred for 4 hr at RT. The reaction was quenched by addition of saturated aqueous solution of $NH_4Cl$, followed by extraction with EtOAc (3 × 5 mL). The combined organic fractions were dried over anhydrous $MgSO_4$, filtrated and concentrated under reduced pressure. The residue was purified by silica gel column chromatography (pentane/EtOAc 10:3) to give heptadec-1-en-4-yne-3,6-diol(**12**) as a colourless oil (17 mg, 64%). Mixture of diastereomers **¹H NMR** (400 MHz, $CDCl_3$): $\delta$ 5.97 (ddd, $J$ = 17.0, 10.1, 5.3 Hz, 1 H), 5.45 (dt, $J$ = 17.0, 1.5 Hz, 1 H), 5.23 (dt, $J$ = 10.2, 1.3 Hz, 1 H), 4.95–4.89 (m, 1 H), 4.42 (td, $J$ = 8.0 Hz, 1.5 Hz, 1 H), 1.75–1.66 (m, 2 H), 1.49–1.39 (m, 2 H), 1.35–1.20 (m, 16 H), 0.87 (t, $J$ = 4.0 Hz, 3 H). **¹³C NMR** (100 MHz, $CDCl_3$): $\delta$ 136.9, 116.7, 87.7, 83.7, 63.3, 62.6, 37.8, 32.1, 29.8, 29.8, 29.7, 29.7, 29.5, 29.4, 25.3, 22.8, 14.3. **HRMS-DCI** ($CH_4$): $m/z$ calcd for $C_{17}H_{31}O_2$ $[M + H]^+$: 267.2324 found: 267.2330. **FTIR** (cm⁻¹) (neat): $v$ 3277, 2920, 2852, 1466, 1268, 1146, 1015, 985, 927, 697. See *Supplementary file 6K* for **12** full spectra.

## Statistical analysis

Error bars on graphs represent standard deviations (SD). To assess statistically significant differences, Student's t-tests were used, and the resulting p-value provided. A difference was considered significant when < 0.05.

## Acknowledgements

We are grateful to the Genotoul bioinformatics platform Toulouse Occitanie and TRI-IPBS Imaging Core Facility, member of TRI-Genotoul, for providing help, computing and storage resources. The NMR spectra were recorded on spectrometers of the Integrated Screening Platform of Toulouse (PICT, IBISA). We thank Laurence Nieto (team C Muller, IPBS, Toulouse, France), Raphaël Rodriguez (Institut Curie, Paris, France), Frédérique Fallone (Team C Muller, IPBS, Toulouse, France), Frédéric Deschaseaux (STROMALab, Toulouse, France), Pierre Cordelier (CRCT, Toulouse, France), Robert A Weinberg (Whitehead Institute, Boston, USA), Erik Snapp (HHMI's Janelia Research Campus, Ashburn, USA), Nico Dantuma (Karolinska Institutet, Stockholm, Sweden) for the generous gift of reagents; Antonio Peixoto & Emmanuelle Näser (IPBS, Toulouse, France) for technical assistance; Andreas Merdes (CBI, Toulouse) for providing access to his microscope. We gratefully acknowledge Stephan A Sieber and his group for their generous support. This study was funded by the grants IDEX Transversalité "Fishing Sponge" (2015 program) from Université Paul Sabatier; N°PJA 20171206477 from "Fondation ARC", ANR-17-CE18-0002-01 from "Agence Nationale de la Recherche" and CAPES-COFECUB Ph-C n° 883/17. The GeT and proteomics facilities received funding from "Investissements d'avenir" program as part of the "Genomic French Infrastructure" (grant ANR-10-INBS-09) and the "Proteomics French Infrastructure" (grant ANR-10-INBS-08 to OB-S), respectively. The Proteomics facility received financial support from the "Fonds Européens de Développement Régional Toulouse Métropole and the Région Midi-Pyrénées" (to OB-S). SMH and PRAZ acknowledge funding by the Fonds der Chemischen Industrie through a Liebig Fellowship and a Ph.D. fellowship and by the TUM Junior Fellow Fund.

## Additional information

### Competing interests

Dymytrii Listunov: Dymytrii Listunov is one of the inventors on a patent deposited on related molecules. Pauline Rullière: Pauline Rullière is one of the inventors on a patent deposited on related molecules. Stéphanie Ballereau: Stéphanie Ballereau is one of the inventors on a patent deposited on related molecules. Valérie Maraval: Valérie Maraval is one of the inventors on a patent deposited on related molecules. Patrick Calsou: Patrick Calsou is one of the inventors on a patent deposited on related molecules. Yves Génisson: Yves Génisson is one of the inventors on a patent deposited on related molecules. Remi Chauvin: Remi Chauvin is one of the inventors on a patent deposited on related molecules. Sébastien Britton: Sébastien Britton is one of the inventors on a patent deposited on related molecules. The other authors declare that no competing interests exist.

### Funding

| Funder | Grant reference number | Author |
| --- | --- | --- |
| University of Toulouse | IDEX "Transversalite" | Yves Génisson<br>Remi Chauvin<br>Sébastien Britton |
| Agence Nationale de la Recherche | ANR-17-CE18-0002-01 | Alexandrine Rozié<br>Patrick Calsou<br>Sébastien Britton |
| CAPES-COFECUB Ph-C | 883/17 | Maria Vieira de Brito<br>Maria Conceição Ferreira Oliveira<br>Vania Bernardes-Génisson |
| Agence Nationale de la Recherche | ANR-10-INBS-09 | Remy-Felix Serre<br>Olivier Bouchez |
| Agence Nationale de la Recherche | ANR-10-INBS-08 | Julien Marcoux<br>Marlène Marcellin<br>Odile Burlet-Schiltz |
| FEDER | | Odile Burlet-Schiltz |

| Funder | Grant reference number | Author |
| --- | --- | --- |
| Fonds der Chemischen Industrie | Liebig and PhD Fellowship | Patrick RA Zanon Stephan M Hacker |
| Technical University of Munich | Junior Fellow Fund | Patrick RA Zanon Stephan M Hacker |
| Ligue Contre le Cancer | Programme Equipe Labellisée | Alexandrine Rozié Karen Pradines Romain Hee Patrick Calsou Sébastien Britton |
| Toulouse Metropole | | Odile Burlet-Schiltz |
| Region Midi-Pyrénées | | Odile Burlet-Schiltz |
| Fondation ARC pour la Recherche sur le Cancer | PJA 20171206477 | Pauline Rullière Stéphanie Ballereau Valérie Maraval Patrick Calsou Yves Génisson Remi Chauvin Sébastien Britton |

The funders had no role in study design, data collection and interpretation, or the decision to submit the work for publication.

## Author contributions

Pascal Demange, Data curation, Formal analysis, Investigation, Methodology, Supervision, Visualization, Writing – review and editing; Etienne Joly, Pauline Rullière, Formal analysis, Investigation, Writing – review and editing; Julien Marcoux, Data curation, Formal analysis, Investigation, Methodology, Validation, Visualization, Writing – review and editing; Patrick RA Zanon, Data curation, Formal analysis, Investigation, Methodology, Visualization, Writing – review and editing; Dymytrii Listunov, Cécile Barthes, Investigation, Writing – review and editing; Céline Noirot, Data curation, Formal analysis, Methodology, Writing – review and editing; Jean-Baptiste Izquierdo, Maria Vieira de Brito, Formal analysis, Investigation; Alexandrine Rozié, Karen Pradines, Romain Hee, Remy-Felix Serre, Olivier Bouchez, Investigation; Marlène Marcellin, Data curation, Formal analysis, Investigation; Odile Burlet-Schiltz, Funding acquisition, Methodology, Supervision; Maria Conceição Ferreira Oliveira, Investigation, Supervision; Stéphanie Ballereau, Vania Bernardes-Génisson, Valérie Maraval, Formal analysis, Investigation, Supervision, Writing – review and editing; Patrick Calsou, Conceptualization, Formal analysis, Funding acquisition, Validation, Writing – review and editing; Stephan M Hacker, Data curation, Formal analysis, Investigation, Supervision, Visualization, Writing – review and editing; Yves Génisson, Conceptualization, Formal analysis, Funding acquisition, Supervision, Writing – review and editing; Remi Chauvin, Conceptualization, Formal analysis, Funding acquisition, Writing – review and editing; Sébastien Britton, Conceptualization, Data curation, Formal analysis, Funding acquisition, Investigation, Methodology, Project administration, Supervision, Validation, Visualization, Writing – original draft, Writing – review and editing

## Author ORCIDs

Patrick RA Zanon http://orcid.org/0000-0002-8883-8275
Stéphanie Ballereau http://orcid.org/0000-0002-7250-6188
Yves Génisson http://orcid.org/0000-0002-3647-4617
Remi Chauvin http://orcid.org/0000-0002-4491-6390
Sébastien Britton http://orcid.org/0000-0002-7008-5316

## Decision letter and Author response

Decision letter https://doi.org/10.7554/eLife.73913.sa1
Author response https://doi.org/10.7554/eLife.73913.sa2

## Additional files

### Supplementary files

• Transparent reporting form

• Supplementary file 1. Amino acid selectivity of the reaction between DACones and proteins in U2OS extracts.

• Supplementary file 2. Positions on proteins of the lysines and cysteines modified by DACones in U2OS extracts.

• Supplementary file 3. List of proteins identified as modified by the (S)-DAC 9 in U2OS cells.

• Supplementary file 4. Characterization and positions of DAC adducts on proteins in (S)-DAC 9-treated U2OS cells.

• Supplementary file 5. Information regarding the plasmids generated and the oligonucleotides and antibodies used in this study.

• Supplementary file 6. NMR spectra of the novel compounds generated in this study.

### Data availability

RNA-seq data have been deposited on SRA (Bioproject IDs PRJNA668246 and PRJNA668322). Mass spectrometry proteomics data have been deposited to the ProteomeXchange Consortium via the PRIDE partner repository with the data set identifier PXD033059. Plasmids are deposited on the Addgene plasmid repository. Other data generated or analysed during this study are included in the manuscript and supporting files.

The following datasets were generated:

| Author(s) | Year | Dataset title | Dataset URL | Database and Identifier |
|---|---|---|---|---|
| Britton S, Noirot C | 2020 | RNA-seq sequencing data from individual (S)-DAC-resistant HAP-1 clones | https://www.ncbi.nlm.nih.gov/bioproject/?term=PRJNA668246 | NCBI BioProject, PRJNA668246 |
| Britton S, Noirot C | 2020 | RNA-seq sequencing data from individual (S)-AAC-resistant HAP-1 clones | https://www.ncbi.nlm.nih.gov/bioproject/?term=PRJNA668322 | NCBI BioProject, PRJNA668322 |
| Britton S, Marcoux J, Hacker SM, Zanon P | 2021 | Lipidic alkynylcarbinols as pro-cytotoxic agents: enantiospecific bioactivation by SDR enzymes | http://proteomecentral.proteomexchange.org/cgi/GetDataset?ID=PXD033059 | ProteomeXchange, PXD033059 |

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
