## [Editor Report]

This manuscript describes an elegant chemical-genetic strategy to discover that human oxidoreductase HSD17B11 is a major contributor to the bioactivation of the dialkinylcarbinol class of cytotoxic natural products. Mechanistic work further revealed that the reactive metabolites generated by HSD17B11 modify lysine and cysteine side-chains on proteins, leading to an unfolded protein response and apoptotic cell death. This study thus provides a plausible mechanism to explain how dialkynylcarbinol compounds exert their cytotoxic properties and identifies enzyme targets for controlling this process in human cells.

---

## [Decision Letter]

**Decision letter after peer review:**

Thank you for resubmitting your work entitled "SDR enzymes oxidize specific lipidic alkynylcarbinols into cytotoxic protein-reactive species" for further consideration by *eLife*. Your revised article has been evaluated by David Ron (Senior Editor) and a Reviewing Editor.

The manuscript has been improved but there are some remaining issues that need to be addressed, as outlined below:

Summarizing the reviews and discussion, consensus was reached that your manuscript reported interesting findings that were, in large part, supported by the data. However, we feel that determining the chemical structure of the covalent adduct formed with nucleophilic amino acid side chains on representative protein targets is an important objective that should be experimentally addressed as part of the revision process. We also ask that you minimize the speculation on the therapeutic implications of the work, as this direction raises several additional questions for which data are lacking. This latter point can likely be addressed through appropriate changes to the text of the revised manuscript to better emphasize the intriguing fundamental research findings while toning down the speculation on drug development.

*Reviewer #1:*

Demange et al., identify through a series of elegant experiments human oxidoreductase HSD17B11 as the major contributor to the bioactivation of dialkinylcarbinols (DACs) to a reactive carbonyl species. The identification of HSD17B11 as main target of the activation to the major reactive metabolite was achieved by using an EMS approach in the pseudo-haploid cell line HAP1 to mutate in a genome-wide manner and select clones that are resistant to treatment with the starting DAC. The wild-type line requires bioactivation to a cytotoxic carbonyl species and is sensitive to the cytotoxic agent. These bioactivated reactive, electronegative compounds modify lysine and cysteine side-chains of several proteins by a Michael addition reaction mechanism. This can induce an unfolded protein response (UPR), which in turn can lead to apoptotic cell death. HSD17B11 as well as the related enzymes hydroxyprostaglandin dehydrogenase and retinol dehydrogenase 11, which also appear to be involved in this bioactivation step (although to a lesser degree), are members of the short-chain dehydrogenase/reductase (SDR) superfamily, an evolutionarily conserved protein family found in all forms of life with about 80 members identified in the human genome. The study provides a plausible mechanism how a class of compounds containing dialkynylcarbinol motifs, most often found in natural products, exert their cytotoxic properties and accordingly this work provides significant information, of utility in the toxicology field.

Strengths: the undisputed strength (and beauty) of the manuscript is the chemical-genomic approach using a combination of biochemical and molecular resources, such as use of the pseudo-haploid cell line, the EMS mutational approach to identify the underlying oxidoreductase(s), the mass spectrometry work and chemical synthesis.

Weaknesses: Several open questions remain. Several parts of the paper appear overemphasized. In part, the underlying hypothesis (for example the explanation of 17beta HSDs as DAC dehydrogenase through the 17beta-hydroxy stereochemistry of the steroid D ring) appears farfetched: all secondary alcohol substrates should have a similar configuration and hence any oxidoreductase involved could be a candidate for the observed DAC oxidation. Moreover, HPGD and RDH11 do not have steroid DH activity. Conversely, other HSD17beta DHs could possibly carry out the reaction (which they apparently do not). Moreover, and this speaks in favour of the importance of the paper, other classes of oxidoreductases such as aldo-keto reductases (AKR) or especially medium-chain dehydrogenases/reductases (MDR) could in principle carry out the observed reaction, potentially widening significantly the target field for a series of related DAC activating enzymes.

More importantly, a part of the postulated importance in developing DACs as possible cancer therapeutics has been postulated based on osteosarcoma cell line data. HSD17B11 is certainly very widely expressed including liver and blood cells. For sure, the DAC activation would take place in these tissues by HSD17B11 (and HPGD, RDH11). It would be interesting to learn more about in vivo toxicology of the DACs, I would expect a high degree of hepatotoxicity and suppression of white blood cell formation. A major concern for me is the apparent lack of GSH conjugation or other defense mechanisms (such as MDR enzymes) against the reactive carbonyl species. The reactive alkynyl-ketones should have electronegative centers that would be prime targets for glutathione-S- conjugation or even non-enzymatic reactions with GSH. Has anything like that been observed for the reactive carbonyls? The authors show it for N-acetyl cycteine for example, so it could work with GSH as well. Is there a possibility for enzymatic reduction of the yne-triple bonds?

Finally, it would be of interest to dissect the UPR further (beyond IRE1a phosphorylation). It would be of high interest to do so and look for the other elements as well- such as XBP splicing, PERK activation, eiF2a phosphorylation, ATF expression etc. Excellent tool compounds are available to probe this important phenotypic response further, including PERK and IRE1a inhibitors (targeting the nucleotidase and the kinase domains).

The most significant open questions/points have been already highlighted in the Public review section. Some of the "easy to do" experiments such as GSH adducts or mining the proteomic data for additional adducts would be nice to see (or discuss properly). It would be good to include what is known regards HSD17B11 in humans, for example the genomic data in lean mass/obesity etc. In addition, I think it would also be appropriate to highlight and discuss some of the concepts identified in the reviews in the carbonyl reductase/dehydrogenase/toxicology field such as –

Oppermann U. Carbonyl reductases: the complex relationships of mammalian carbonyl- and quinone-reducing enzymes and their role in physiology. Annu Rev Pharmacol Toxicol. 2007;47:293-322. doi:10.1146/annurev.pharmtox.47.120505.105316.

Penning TM, Jonnalagadda S, Trippier PC, Rižner TL. Aldo-Keto Reductases and Cancer Drug Resistance. Pharmacol Rev. 2021 Jul;73(3):1150-1171. doi: 10.1124/pharmrev.120.000122.

Meier M, Möller G, Adamski J. Perspectives in understanding the role of human 17beta-hydroxysteroid dehydrogenases in health and disease. Ann N Y Acad Sci. 2009 Feb;1155:15-24. doi: 10.1111/j.1749-6632.2009.03702.x.

Hoffmann F, Maser E. Carbonyl reductases and pluripotent hydroxysteroid dehydrogenases of the short-chain dehydrogenase/reductase superfamily. Drug Metab Rev. 2007;39(1):87-144. doi: 10.1080/03602530600969440. PMID: 17364882.

The authors mention correctly the official nomenclature symbols for the human SDRs – please cite the underlying reference –

Persson B, et al., The SDR (short-chain dehydrogenase/reductase and related enzymes) nomenclature initiative. Chem Biol Interact. 2009 Mar 16;178(1-3):94-8. doi: 10.1016/j.cbi.2008.10.040.

*Reviewer #2:*

Britton and colleagues reported in this manuscript their efforts to identify a class of short-chain dehydrogenase/reductase(SDR) that are responsible for converting terminal alkynylcarbinols into the corresponding ketone species, which in turn covalently modify functional cysteines and lysines in protein targets and induce strong cytotoxicity. They first performed a genetic screen to isolate cell clones that are resistant to toxic alkynylcarbinols and found that resistant mutations frequently occur on a member of SDR named HSD17B11. They then demonstrated that the catalytic activity of HSD17B11 is critical in mediating the toxicity of alkynylcarbinols and cells that are lacking the expression of this SDR are more resistant. They further synthesized different bioorthogonal analogues of alkynylcarbinols and their corresponding control compounds to show the resulting ketone species could covalently label proteins on their nucleophilic cysteines both in vitro and in living cells. By performing chemical proteomics experiments, they identified a list of proteins that are modified by the activated alkynylcarbinols, one of which is involved in the ubiquitin-proteosome pathway. Another interesting aspect of this work is that they found the proteins that are modified by these lipidic alkynylcarbinols derivatives tend to show different subcellular localizations. They finally demonstrated in proof-of-concept that the alkynylcarbinols can serve as potential prodrugs which are activated by SDR members and kill cancer cells.

Strength:

The work is super rich in content and integrates an impressive list of different techniques to confirm each hypothesis that the authors made along the way. It mechanistically interprets the pathway how the alkynylcarbinols get activated in cells and how the resulting electrophilic species react covalently with proteins and changed their physical/chemical properties. The work proposes a new prodrug strategy to selectively kill cancers.

Weakness:

With clear indication of covalent modifications of proteins by these activated alkynylcarbinols, the authors did not identify the key targets that are responsible for the observed cytotoxicity. The site-specific chemical proteomics were only implemented to survey the amino acid preference of the activated alkynylcarbinols, however, they were not used to pinpoint specific residue anchors when the compounds were added into cells. Despite that they showed cells underwent apoptosis with sign of ER-stress and inhibition of UPS, the targets or pathways that mediate the cytotoxicity of the electrophilic species are still not clear from the current study.

1. With so many techniques implemented and so much data acquired, I would suggest authors to trim certain data to improve the overall readability of the manuscript. For example, the last part of RDH11 and HPGD seemed to not absolutely necessary to me and distracted the key finding of HSD17B11. In addition, too many technical details are put into the results session which make the manuscript a bit hard to read and follow.

2. Not very clear to me why the authors used the isotopically labeled isoDTB for evaluating the site-specific reaction preferences, but when it came to identify the cellular targets, they put the isoDTB aside and instead just used label-free quantification at the protein level. Ideally, isoDTB would work perfectly with their alkynyl probes to identify and quantify critical sites of modifications and provide a better picture of target landscape of these activated alkynylcarbinols.

*Reviewer #3:*

In the manuscript "SDR enzymes oxidize specific lipidic alkynylcarbinols into cytotoxic 2 protein-reactive species," the authors use functional genomics and chemoproteomics to elucidate mode of action of cytotoxic dialkynylcarbinol (DAC) containing lipids. They discovered that the DAC lipids are bioactivated in an enantiospecific manner by members of the short-chain dehydrogenase/reductase (SDR) family of enzymes. Using this information, the authors design new SDR-bioactivated DAC lipids as a more general prodrug strategy. The studies are interesting and an impressive array of supporting data are provided. However, direct evidence for the bioactivation mechanism in cells are lacking and this gap should be addressed.

A major critique is the lack of direct evidence for oxidation of alkynylcarbinols to the bioactivated alkynylketone lipids by HSD17B11 or other SDR enzymes in cells. This data is key for supporting the prodrug mode of action of the cytotoxic alkynylcarbinol lipids proposed by the authors. The chemoproteomic evaluation of (S)-9 DAC treated cells only provided protein-level identifications. The inclusion of site of binding chemoproteomic data from cellular studies, which should reveal peptide-lipid adducts matching structures proposed by authors in Figure 2i, is important for supporting SDR-mediated production of reactive acetylenic lipids in cells as proposed by the authors. The authors could also consider the use of metabolomics to identify the alkynylketone lipid products produced in (S)-9 DAC treated cells as direct evidence for the bioactivation.

Additional comments to consider:

1. The authors use the HSD17B11 WT and S172L mutant comparison as support for catalytic function of this enzyme in (S)-DAC cytotoxicity. While these results are clear, the author do not provide direct evidence for disruption of HSD17B11catalytic activity. For example, can the author demonstrate that HSD17B11 S172L lysates are catalytically deficient compared with WT counterparts? What are the effects of this mutation on androsterone levels in this overexpression system? Finally, the inclusion of an additional plasmid coding for a mutation outside of the catalytic domain in the complementation assay is a recommended control given the protein instability issues of from mutations.

2. To support the major critique above, the authors should show by LC-MS or NMR that purified HSD17B11 incubated with (S)-9 under the appropriate biochemical assay conditions produces the dialkynylketone (DACone) 10 product. The authors should compare with the (R)-9 substrate to demonstrate enantio-selectivity. These data should be included to support the claims associated with Figure 2F.

3. Authors should show western blots for HSD17B11 detection in Figure 2F since they report issues with production of the full length S172L protein due to protein instability.

---

## [Author Response]

Summarizing the reviews and discussion, consensus was reached that your manuscript reported interesting findings that were, in large part, supported by the data. However, we feel that determining the chemical structure of the covalent adduct formed with nucleophilic amino acid side chains on representative protein targets is an important objective that should be experimentally addressed as part of the revision process. We also ask that you minimize the speculation on the therapeutic implications of the work, as this direction raises several additional questions for which data are lacking. This latter point can likely be addressed through appropriate changes to the text of the revised manuscript to better emphasize the intriguing fundamental research findings while toning down the speculation on drug development.

Regarding the therapeutic implications: In the revised version, we now restrict to the minimum the speculations about the therapeutic implications of our work. We have deleted and tuned down several sentences, for example:

“In addition, the acute toxicity of (S)-3 towards osteosarcoma cell lines indicates that DACs could be developed into a targeted anti-cancer therapy.” was rephrased to “In addition, the acute toxicity of (S)-3 towards osteosarcoma cell lines suggests that DACs could be developed into a targeted anti-cancer therapy, but this would need to be further investigated, especially in vivo.”

Regarding the identification of the sites of modification by DACs in cells: While we think that it is the modification of multiple proteins by DACs which is the cause of cell death, we were also very keen, much like the reviewers, to identify the site of modifications by DACs in cells, mainly to be able later to evolve DACs into more selective inhibitors bioactivated by HSD17B11.

To achieve this, once we had identified in vitro the mass of the adducts and the structure of the linkage by NMR (Figure 2G, Figure 2—figure supplement 4), we analyzed again our MS/MS data from our streptavidin-based pull-down of DAC-modified proteins. However, no modified peptides were identified through this approach, which could indicate (1) that they remained associated on the magnetic streptavidin beads during the on-beads digestion, or (2) that the modification was lost during the framework, including the final step in which the peptides are resuspended in an acidic solution containing 0.05% trifluoroacetic acid (TFA) or (3) that the adduct is further modified in cells giving MWs different from the expected one.

Therefore, we applied the isoDTB-based workflow developed by Stephan Hacker’s lab to identify the adducts from DAC-treated cells. In this approach, the masses of the different adducts are determined in an unbiased manner by a Fragpipe MSFragger Open search and then quantified. For this experiment, we collected large amounts of extracts from U2OS cells treated for 2 h with 2 µM clickable DAC (*S*)-9 and performed the analysis.

Using this approach, some peptides were reproducibly identified as modified by (*S*)-DAC in cells (Supplementary File 4A). The observed adducts had a higher mass than expected (+154.0122), the excess in mass corresponding to the addition of a dithiothreitol (DTT) molecule (see Supplementary File 4B for the proposed adduct structure). Upon additional inspection, such adducts were also detected in the in vitro DACone treatment experiments (30% of the quantified sites with the DACones 10 and 11). The higher proportion of modification by DTT might result from the higher DTT:peptide ratio in this experiment resulting from the smaller number of peptides being retained in this experiment. The small number of quantified peptides likely results from the lower concentrations of DAC (S)-9 that could be used for the experiments on cells (2 µM of (S)-9) as compared to the in vitro experiments (100 µM of DACone 10 or 11) and from the final acidic elution (0.1% TFA pH~2.1) which promotes the cleavage of the linkage formed between DACone and amine groups (see Figure 2—figure supplement 1H,I). Nevertheless, we were in this way able to further corroborate that the activation of (*S*)-9 in cells leads to the expected reactivity caused by oxidation to the DACone.

Our revised manuscript now incorporates these novel data supporting that DACones are produced from (*S*)-DAC in cells, resulting in the lipoxidation of a set of proteins.

Reviewer #1:Demange et al., identify through a series of elegant experiments human oxidoreductase HSD17B11 as the major contributor to the bioactivation of dialkinylcarbinols (DACs) to a reactive carbonyl species. The identification of HSD17B11 as main target of the activation to the major reactive metabolite was achieved by using an EMS approach in the pseudo-haploid cell line HAP1 to mutate in a genome-wide manner and select clones that are resistant to treatment with the starting DAC. The wild-type line requires bioactivation to a cytotoxic carbonyl species and is sensitive to the cytotoxic agent. These bioactivated reactive, electronegative compounds modify lysine and cysteine side-chains of several proteins by a Michael addition reaction mechanism. This can induce an unfolded protein response (UPR), which in turn can lead to apoptotic cell death. HSD17B11 as well as the related enzymes hydroxyprostaglandin dehydrogenase and retinol dehydrogenase 11, which also appear to be involved in this bioactivation step (although to a lesser degree), are members of the short-chain dehydrogenase/reductase (SDR) superfamily, an evolutionarily conserved protein family found in all forms of life with about 80 members identified in the human genome. The study provides a plausible mechanism how a class of compounds containing dialkynylcarbinol motifs, most often found in natural products, exert their cytotoxic properties and accordingly this work provides significant information, of utility in the toxicology field.Strengths: the undisputed strength (and beauty) of the manuscript is the chemical-genomic approach using a combination of biochemical and molecular resources, such as use of the pseudo-haploid cell line, the EMS mutational approach to identify the underlying oxidoreductase(s), the mass spectrometry work and chemical synthesis.Weaknesses: Several open questions remain. Several parts of the paper appear overemphasized. In part, the underlying hypothesis (for example the explanation of 17beta HSDs as DAC dehydrogenase through the 17beta-hydroxy stereochemistry of the steroid D ring) appears farfetched: all secondary alcohol substrates should have a similar configuration and hence any oxidoreductase involved could be a candidate for the observed DAC oxidation.

We thank Reviewer 1 for pointing this out. Like the Reviewer 1, we were also intrigued by the enantiospecific bioactivation, which we thought was the result of HSD17B11 being adapted to oxidize an hydroxyl group in a specific configuration on its natural substrate. We suspected that both DAC enantiomers could fit in the catalytic domain but that only the (*S*)-DAC would have its hydroxyl group properly oriented to interact with the key catalytic residues (S172 and Y185), necessary to promote the hydride transfer to NAD+. To support this view, we took advantage of AlphaFold2-generated HSD17B11 structural model (see the novel Figure 2—figure supplement 2A) to perform molecular docking with the DACs (*S*)-3 and (*R*)-3. Both (*S*)-3 and (*R*)-3 docked into the catalytic cavity (Figure 2—figure supplement 2B,C), but (*R*)-3 had a lower computed affinity (155 nM *vs* 15 nM for (*S*)-3) and only (*S*)-3 had its hydroxyl group properly positioned to engage into hydrogen bonds with the S172 and Y185 catalytic amino acids (Figure 2—figure supplement 2D,E), which is critical for further NAD+-dependent oxidation (Filling et al., J Biol Chem 2002; Gao et al., Crystals 2021).

Moreover, HPGD and RDH11 do not have steroid DH activity. Conversely, other HSD17beta DHs could possibly carry out the reaction (which they apparently do not).

There are 13 human 17βHSDs SDRs in human and we were also intrigued by the selective bioactivation of (*S*)-DACs by HSD17B11 over these enzymes or other SDRs. Possible explanations for this are provided by the facts that (1) most of the other 17betaHSD SDRs are not expressed in the cell lines we used in our study (U2OS and HAP1 for most experiments, see the table in Figure 2—figure supplement 2F), (2) that 4 of the other 17betaHSD SDRs have the reverse activity (reductive SDRs) and therefore could not promote the oxidative bioactivation, and (3) that HSD17B11 has some specificities such as its preference for the androgen 5α-androstane-3α,17βdiol over prototypical 17βHSD SDR substrates (dihydrotestosterone and testosterone ; Brereton et al., Mol Cell Endocrinol 2001). This could suggest that, among the 17betaHSD SDRs, HSD17B11 has a catalytic domain which is the most adapted to oxidize DACs and related molecules. To challenge this view, we took advantage of AlphaFold2-generated models for all the 17betaHSD SDRs (available on Uniprot) to perform molecular docking with the DAC (*S*)-3 and (*R*)-3 onto their catalytic domain. Filtering for the most stringent poses revealed that, in addition to HSD17B11, (*S*)-3 also docked into the catalytic domains of only 2 SDRs, HSD17B13 and HSD17B3, while (*R*)-3 docked into 2 different SDRs, HSD17B9 and HSD17B6. HSD17B3 (aka SDR12C2) is a reductive SDR involved in testosterone biosynthesis, so it was not tested further. In contrast, since HSD17B13 (aka SDR16C3), whose expression is restricted to the liver, is an oxidative SDR and the closest homolog of HSD17B11 (~63% sequence identity), we tested whether it could complement U2OS KO for HSD17B11. This revealed that HSD17B13 is also able to bioactivate (*S*)-3 into cytotoxic compounds, however in a less efficient manner as compared to HSD17B11 (IC_50_ of 30 nM vs 12 nM for HSD17B11 with similar complementation levels, see the novel Figure 2—figure supplement 2G,H). Altogether these findings support that HSD17B11 has, among the other 17betaHSDs, the best affinity for DACs and DAC-like molecules. These data also suggest that (*R*)-DACs could be bioactivated selectively by other SDRs, such as HSD17B9 and HSD17B6, which opens new directions of research. Note that we have deposited all the expression plasmids of the tested SDRs on Addgene, which should greatly stimulate and benefit this field of research.

Altogether these data support that HSD1711 or other SDRs will display an enantiospecificity for bioactivating DACs and related molecules as the result of having evolved to recognize a natural substrate which is itself in a specific configuration. This also implies that, despite being simple in term of structures, (*S*)-DAC insert in a specific manner in the catalytic domain of HSD17B11, as supported by our molecular docking analyses.

Moreover, and this speaks in favour of the importance of the paper, other classes of oxidoreductases such as aldo-keto reductases (AKR) or especially medium-chain dehydrogenases/reductases (MDR) could in principle carry out the observed reaction, potentially widening significantly the target field for a series of related DAC activating enzymes.

We agree with Reviewer 1 that other classes of enzymes able to oxidize secondary carbinol groups could in theory bioactivate specifically designed prodrugs carrying a DAC-based warhead. This is one of the key points we wanted to make through the data presented in Figure 5: it is possible to design novel substrates for bioactivation by a specific enzyme. Some selectivity can be achieved by modulating the pharmacophore (e.g. AllAC for HPGD or AAC for RDH11) and/or the lipidic backbone (e.g. addition of a second -OH group which blocks/reduces the activation by HSD17B11). We now further discuss this point taking advantage of some of the references suggested by Reviewer 1:

"In addition, other oxidoreductases, especially among the Medium-Chain dehydrogenase/Reductase (MDR) superfamily, could in principle carry out similar oxidation of secondary alcohol-containing substrates (72), which could widen the field of application of our findings. Conversely, some carbonyl reductases, especially among the Aldo-Keto Reductase (AKR) superfamily (73,74), could theoretically antagonize DACones cytotoxicity by reverting the reactive ketone into the corresponding alcohol, which could be further conjugated and eliminated."

More importantly, a part of the postulated importance in developing DACs as possible cancer therapeutics has been postulated based on osteosarcoma cell line data. HSD17B11 is certainly very widely expressed including liver and blood cells. For sure, the DAC activation would take place in these tissues by HSD17B11 (and HPGD, RDH11). It would be interesting to learn more about in vivo toxicology of the DACs, I would expect a high degree of hepatotoxicity and suppression of white blood cell formation.

We did not intend to oversell our data with the osteosarcoma cell lines. They simply show that some osteosarcoma cancer cell lines have ~10 times higher sensitivity to DAC as compared to primary osteoblast cells. Multiple anticancer agents (e.g. etoposide) have similarly low selectivity indices, associated with known side effects, which did not prevent them from being used to treat and cure some cancers. We completely agree with Reviewer 1 that (1) our data obtained with cancer cell lines are not sufficient to validate DACs as anticancer agents and (2) that the anticancer efficacy should be evaluated in vivo to take into account potential toxicity effects and unfavorable pharmacokinetics properties. We just wanted to point out that (*S*)-DACs display differential cytotoxic activity against various cell lines despite HSD17B11 being almost ubiquitously expressed. This differential cytotoxicity might be exploited to develop DACs and analogues to target specific HSD17B11-overexpressing cancers such as some advanced human prostate cancers which overexpress HSD17B11 as compared to the normal tissue (Nakamura et al., 2009). In response to Reviewer 1 comments, we tuned down the potential clinical implication of our work (see above) and added the reference related to the overexpression of HSD17B11 in human prostate cancers to highlight some potential applications.

A major concern for me is the apparent lack of GSH conjugation or other defense mechanisms (such as MDR enzymes) against the reactive carbonyl species. The reactive alkynyl-ketones should have electronegative centers that would be prime targets for glutathione-S- conjugation or even non-enzymatic reactions with GSH. Has anything like that been observed for the reactive carbonyls? The authors show it for N-acetyl cycteine for example, so it could work with GSH as well.

Like Reviewer 1, and based on our in vitro data demonstrating that DACone are highly reactive with thiol groups at pH7, we anticipated that the toxicity of DAC would be modulated by glutathione (GSH) and glutathione S-transferases (GSTs). We thought that this could even explain why some cell lines, such as A549, are not as sensitive to DAC as U2OS are, despite almost similar HSD17B11 expression levels. To investigate this, we first tested in vitro the reaction of DACone 8 with reduced GSH at pH7. This revealed that DACone 8 is indeed able to react with GSH, with the appearance of an absorbance band at ~323 nm (Figure 3—figure supplement 3A,B). NMR analyses confirmed the formation of an adduct on the GSH thiol group with a structure similar to the one observed for DACone and NAC (Figure 3—figure supplement 3C). Then we tested in cells whether GSH supplementation (using the cell permeable GSH monoethyl ester = GSHe) could reduce (*S*)-DAC sensitivity in U2OS. However, we observed only a small and non-significant impact of GSHe on (*S*)-DAC cytotoxicity (Figure 3—figure supplement 3D). We also attempted to sensitize A549 cells to (*S*)-DAC by using the GSTs inhibitor ethacrynic acid (GSTi). However, again there was only a small dose-dependent effect of the GSTi on (*S*)-3 cytotoxicity (Figure 3—figure supplement 3E). Altogether, these data show that GST and GSH only play a minor role in modulating DAC cytotoxic activity. It is possible that the local generation of DACones protects them from a general detoxification mechanism. This is in line with our MS/MS data in which only a small set of proteins was found modified by DAC (*S*)-**9**. This set of proteins might correspond to the one in the direct proximity of HSD17B11, as shown for another *in situ* generated electrophile (Paxman et al., *eLife* 2018). Thanks to Reviewer 1, these data are now added to the manuscript and discussed (novel Figure 3—figure supplement 3). They highlight that, while HSD17B11 expression is critical for (*S*)-DAC cytotoxicity, the degree of sensitivity to DACs has other determinants, such as, possibly, the level of the NAD+ co-factor.

Is there a possibility for enzymatic reduction of the yne-triple bonds?

CaeEnR1, the first enzyme being able to perform enzymatic reduction of alkyne triple bonds, was recently identified in the fungus *Cyclocybe aegerita* (Karrer et al., ChemCatChem 2021). Its closest relatives in human are prostaglandin reductases but these enzymes have not been shown to reduce alkyne groups and we feel that they are therefore unlikely to modulate DAC cytotoxicity.

Finally, it would be of interest to dissect the UPR further (beyond IRE1a phosphorylation). It would be of high interest to do so and look for the other elements as well- such as XBP splicing, PERK activation, eiF2a phosphorylation, ATF expression etc. Excellent tool compounds are available to probe this important phenotypic response further, including PERK and IRE1a inhibitors (targeting the nucleotidase and the kinase domains).

As suggested by Reviewer 1, we tested multiple markers of UPR activation. In agreement with (*S*)-3 triggering unfolded proteins accumulation and UPR activation, we observed that treatment with (*S*)-3 induced the activation of the three UPR pathways with the rapid phosphorylation of IRE1α (S724), PERK (T980) and eIF2α (S51), a strong decrease of full-length ATF6, marking its cleavage into a short-lived transcriptionally active fragment and preceding the accumulation of cytoplasmic HSP70 and the transient accumulation of XBP-1s and ATF4 (see modified Figure 4G). These data support that (*S*)-DAC 3 induces a fast (< 2 h) and strong activation of all three UPR sub-pathways.

As suggested by Reviewer 1, we also tested whether known UPR sub-pathway inhibitors could sensitize cells to (*S*)-DAC. The individual inhibition of IRE1α kinase, IRE1α endonuclease or PERK kinase activity did not show a significant impact on (*S*)-DAC cytotoxicity suggesting that the ER stress induced by (*S*)-**3** is too acute to be resolved in a timely manner by UPR activation (see novel Figure 4—figure supplement 1E) or that the combined inhibition of all three pathways may be needed to modulate cell sensitivity. In contrast, inhibition of *JNK*, which is known to contribute to the apoptosis induced by prolonged UPR activation (for review see Szegezdi et al., EMBO Reports 2006), provided a small degree of resistance to (*S*)-**3** (IC_50_ shift from 33 to 70 nM), suggesting a contribution of the *JNK* pathway to cell death (see novel Figure 4—figure supplement 1F).

The most significant open questions/points have been already highlighted in the Public review section. Some of the "easy to do" experiments such as GSH adducts or mining the proteomic data for additional adducts would be nice to see (or discuss properly).

As presented above, in our revised manuscript we now include additional experiments to investigate the role of GSH and GSTs in the modulation of (*S*)-DAC cytotoxic activity. We also used direct-infusion mass spectrometry to test whether GSH could react with DACone-modified BLG.

We found that GSH did not form an adduct with the DACone adduct itself, nor displaced it. These data have not been included since the experiments previously presented already supported that GSH and GST only play a minor role in modulating (*S*)-DAC cytotoxicity.

It would be good to include what is known regards HSD17B11 in humans, for example the genomic data in lean mass/obesity etc.

We thank the referee for this helpful suggestion and our revised version includes additional information about HSD17B11 ubiquitous expression, overexpression in some cancers and the identification of a HSD17B11 polymorphism linked to lean body mass as follows:

"The literature on HSD17B11 is currently limited. It was shown in vitro to promote androgen inactivation by converting the potent androstan-3-α,17-β-diol into a weaker androgen (17). In human, the HSD17B11 variant (rs9991501 ; Arg283Gln) was found associated with lean body mass (60), supporting that its activity controls muscle physiology, a process regulated by androgens. However, which HSD17B11-generated metabolites are involved and how the Arg283Gln variant impacts on HSD17B11 activity is currently unknown. HSD17B11 is nearly ubiquitously expressed (17,61), but is also overexpressed in some human malignancies, including advanced prostate cancer and non-small cell lung cancer cell lines, as compared to normal tissues or cell lines (62,63). However, the potential role for HSD17B11 in cancer progression remains to be established. The HSD17B11-bioactivated clickable prodrugs described here could represent valuable tools to investigate HSD17B11 enzymatic activity in tissues through imaging and to decipher its physiological and therapeutic relevance."

In addition, I think it would also be appropriate to highlight and discuss some of the concepts identified in the reviews in the carbonyl reductase/dehydrogenase/toxicology field such as –Oppermann U. Carbonyl reductases: the complex relationships of mammalian carbonyl- and quinone-reducing enzymes and their role in physiology. Annu Rev Pharmacol Toxicol. 2007;47:293-322. doi:10.1146/annurev.pharmtox.47.120505.105316.Penning TM, Jonnalagadda S, Trippier PC, Rižner TL. Aldo-Keto Reductases and Cancer Drug Resistance. Pharmacol Rev. 2021 Jul;73(3):1150-1171. doi: 10.1124/pharmrev.120.000122.Meier M, Möller G, Adamski J. Perspectives in understanding the role of human 17beta-hydroxysteroid dehydrogenases in health and disease. Ann N Y Acad Sci. 2009 Feb;1155:15-24. doi: 10.1111/j.1749-6632.2009.03702.x.Hoffmann F, Maser E. Carbonyl reductases and pluripotent hydroxysteroid dehydrogenases of the short-chain dehydrogenase/reductase superfamily. Drug Metab Rev. 2007;39(1):87-144. doi: 10.1080/03602530600969440. PMID: 17364882.

We now discuss the potential role of carbonyl reductases in limiting DAC cytotoxicity, which is something that could be explored in future studies:

“In addition, other oxidoreductases, especially among the Medium-Chain dehydrogenase/Reductase (MDR) superfamily, could in principle carry out similar oxidation of secondary alcohol-containing substrates (72), which could widen the field of application of our findings. Conversely, some carbonyl reductases, especially among the Aldo-Keto Reductase (AKR) superfamily (73,74), could theoretically antagonize DACones cytotoxicity by reverting the reactive ketone into the corresponding alcohol, which could be further conjugated and eliminated. However, the fact that modulating glutathione levels or GST activity in cells only had a very small impact on (*S*)-DAC cytotoxicity (Figure 3—figure supplement 3D,E) indicates that, once produced, DACones immediately react with nearby proteins which limits the activity of general detoxification mechanisms.”

The authors mention correctly the official nomenclature symbols for the human SDRs – please cite the underlying reference –Persson B, et al., The SDR (short-chain dehydrogenase/reductase and related enzymes) nomenclature initiative. Chem Biol Interact. 2009 Mar 16;178(1-3):94-8. doi: 10.1016/j.cbi.2008.10.040.

This reference has been added to the manuscript.

Reviewer #2:Britton and colleagues reported in this manuscript their efforts to identify a class of short-chain dehydrogenase/reductase(SDR) that are responsible for converting terminal alkynylcarbinols into the corresponding ketone species, which in turn covalently modify functional cysteines and lysines in protein targets and induce strong cytotoxicity. They first performed a genetic screen to isolate cell clones that are resistant to toxic alkynylcarbinols and found that resistant mutations frequently occur on a member of SDR named HSD17B11. They then demonstrated that the catalytic activity of HSD17B11 is critical in mediating the toxicity of alkynylcarbinols and cells that are lacking the expression of this SDR are more resistant. They further synthesized different bioorthogonal analogues of alkynylcarbinols and their corresponding control compounds to show the resulting ketone species could covalently label proteins on their nucleophilic cysteines both in vitro and in living cells. By performing chemical proteomics experiments, they identified a list of proteins that are modified by the activated alkynylcarbinols, one of which is involved in the ubiquitin-proteosome pathway. Another interesting aspect of this work is that they found the proteins that are modified by these lipidic alkynylcarbinols derivatives tend to show different subcellular localizations. They finally demonstrated in proof-of-concept that the alkynylcarbinols can serve as potential prodrugs which are activated by SDR members and kill cancer cells.Strength:The work is super rich in content and integrates an impressive list of different techniques to confirm each hypothesis that the authors made along the way. It mechanistically interprets the pathway how the alkynylcarbinols get activated in cells and how the resulting electrophilic species react covalently with proteins and changed their physical/chemical properties. The work proposes a new prodrug strategy to selectively kill cancers.Weakness:With clear indication of covalent modifications of proteins by these activated alkynylcarbinols, the authors did not identify the key targets that are responsible for the observed cytotoxicity. The site-specific chemical proteomics were only implemented to survey the amino acid preference of the activated alkynylcarbinols, however, they were not used to pinpoint specific residue anchors when the compounds were added into cells. Despite that they showed cells underwent apoptosis with sign of ER-stress and inhibition of UPS, the targets or pathways that mediate the cytotoxicity of the electrophilic species are still not clear from the current study.1. With so many techniques implemented and so much data acquired, I would suggest authors to trim certain data to improve the overall readability of the manuscript. For example, the last part of RDH11 and HPGD seemed to not absolutely necessary to me and distracted the key finding of HSD17B11. In addition, too many technical details are put into the results session which make the manuscript a bit hard to read and follow.

We have trimmed some of the technical details from the result section, but we decided to keep the part about RDH11- and HPGD-mediated oxidation since they provide a proof-of-concept that other prodrugs can be generated for bioactivation by other SDRs, different from HSD17B11, which is an important message of our study.

2. Not very clear to me why the authors used the isotopically labeled isoDTB for evaluating the site-specific reaction preferences, but when it came to identify the cellular targets, they put the isoDTB aside and instead just used label-free quantification at the protein level. Ideally, isoDTB would work perfectly with their alkynyl probes to identify and quantify critical sites of modifications and provide a better picture of target landscape of these activated alkynylcarbinols.

The reason for not using in our initial submission the more complex isoDTB-based workflow is mainly historical: the chemoproteomics analyses depicted in Figure 3B were performed in 2016, while the isoDTB-based analyses of the DACone amino acids selectivity were performed later in 2021. Despite being simpler, we are confident with the data generated by the label free approach, since we could validate the main hits as being modified in cells through high stringency immunoprecipitations (Figure 3C and Figure 3—figure supplement 2C). However, like Reviewer 2, we were also interested by identifying the protein modification sites by the bioactivated (*S*)-DAC, mainly to be able later to evolve DACs into more specific inhibitors selectively bioactivated by HSD17B11. We therefore attempted, as suggested by the Reviewers, to apply the isoDTB-ABPP workflow to the identification of the proteins modified by (*S*)-DAC directly from cell extracts.

As presented in the general response, this led us to validate that upon (*S*)-DAC treatment some peptides are modified in cells by the expected DACone adducts (Supplementary File 4A). Of note the adducts themselves were found modified by DTT during the workflow, probably on the internal alkyne (see Supplementary File 4B). Such DTT-modified adducts were also previously detected in the in vitro DACone treatment (30% of the quantified sites for the DACone 10- and 11-protein adducts).

However, it is noteworthy that the isoDTB-based analysis only identified a small number of peptides, all modified on a cysteine. This led us to investigate whether the modifications on the amine group of lysines could be partly cleaved during the isoDTB-ABPP workflow, especially during the final elution, which consists in a 1 h incubation in 0.1% TFA. As now shown in the novel Figure 2—figure supplement 1H,I, this revealed that incubation in 0.1% TFA at room-temperature progressively and selectively promotes the cleavage of the DACone-amine bonds over the DACone-thiol ones. This, combined with the reduced concentration of (*S*)-DAC which could be used for cell treatments as compared to the in vitro analysis (2 µM in cells vs 100 µM in vitro), likely accounts for the reduced number of peptides identified through this approach. Therefore, our data support that the label-free and isoDTB-based approaches are complementary with the simpler label-free one being best suited for the identification of the proteins modified by (*S*)-DAC in cells, while the isoDTB-framework being most appropriated for the unbiased identification of the adducts mass in cells and for mapping the modification sites. Using different approaches to elute or cleave the isotopically labelled tags and the associated proteins represents an attractive avenue to improve the isoDTB-ABPP workflow and to avoid the biases induced by the acidic elution.

To conclude, we are grateful for Reviewer 2’s suggestion which led us to validate the formation of the adducts in cells, to identify some modification sites which could be further explored and to reveal the differential stability of the lysine- and cysteine-DACone linkages at acidic pH. These data are now integrated and discussed in the manuscript.

Reviewer #3:In the manuscript "SDR enzymes oxidize specific lipidic alkynylcarbinols into cytotoxic 2 protein-reactive species," the authors use functional genomics and chemoproteomics to elucidate mode of action of cytotoxic dialkynylcarbinol (DAC) containing lipids. They discovered that the DAC lipids are bioactivated in an enantiospecific manner by members of the short-chain dehydrogenase/reductase (SDR) family of enzymes. Using this information, the authors design new SDR-bioactivated DAC lipids as a more general prodrug strategy. The studies are interesting and an impressive array of supporting data are provided. However, direct evidence for the bioactivation mechanism in cells are lacking and this gap should be addressed.A major critique is the lack of direct evidence for oxidation of alkynylcarbinols to the bioactivated alkynylketone lipids by HSD17B11 or other SDR enzymes in cells. This data is key for supporting the prodrug mode of action of the cytotoxic alkynylcarbinol lipids proposed by the authors. The chemoproteomic evaluation of (S)-9 DAC treated cells only provided protein-level identifications. The inclusion of site of binding chemoproteomic data from cellular studies, which should reveal peptide-lipid adducts matching structures proposed by authors in Figure 2i, is important for supporting SDR-mediated production of reactive acetylenic lipids in cells as proposed by the authors. The authors could also consider the use of metabolomics to identify the alkynylketone lipid products produced in (S)-9 DAC treated cells as direct evidence for the bioactivation.

We agree with Reviewer 3 that we did not directly show that DACones are produced in cells and we were careful not to oversell our finding regarding this aspect. However, considering the high reactivity of DACones in vitro, our view is that the reactive intermediate cannot be trapped in a simple manner when enzymatically produced. This is demonstrated thanks to one of the additional experiment requested by Reviewer 3 (see point 3, below), namely a blot showing that, in our in vitro bioactivation experiment, we had similar amounts of WT and S172L HSD17B11 mutant. On that blot, we also checked the modification of HSD17B11-GFP by bioactivated clickable DAC and we observed that HSD17B11-GFP itself was covalently modified by the HSD17B11-generated reactive species. The modification of HSD7B11-GFP was decreased by adding betalactoglobuline (BLG) to the reaction, indicating that the BLG competed with HSD17B11 for being modified by the reactive species (see the novel Figure 2F). Altogether these data support that DACones will react with any proteins in their vicinity, which precludes the possibility of isolating enzymatically-produced DACones.

Even though we could not isolate these reactive intermediates, we think that, since HSD17B11, and the related 17beta-HSD oxidative SDRs, are only known to oxidize hydroxyl groups into ketones, and since there is only one hydroxyl group on DAC, the simplest and most likely bioactivation products are the DACones. Thanks to the Reviewer 3 suggestion we now properly discuss this point in the result section:

"Finally, we could recapitulate the activation of (*S*)-**9**, but not of (*R*)-**9**, into protein-reactive species by immunopurified WT HSD17B11, but not by the S172L mutant (Figure 2F), supporting an enantiospecific bioactivation of (*S*)-**9** into the BLG-reactive DACone **10** by HSD17B11. Considering that HSD17B11 known activity is the NAD+-dependent oxidation of a secondary carbinol into a ketone and that the only hydroxyl group on (*S*)-**3** is the one occurring in the dialkynylcarbinol pharmacophore, this experiment strongly supports the notion that HSD17B11 enantiospecifically oxidizes (*S*)-**3** into the DACone **7**, which immediately reacts with nearby proteins, including HSD17B11-GFP itself as observed in Figure 2F. This high level of reactivity unfortunately precludes isolating the HSD17B11-produced DACones."

In addition, we also performed residue-specific experiments (see above) with the isoDTB-ABPP technology in cells and were able to identify some modified peptides that corroborate the production of the DACones in cells upon treatment with (*S*)-9.

Additional comments to consider:1. The authors use the HSD17B11 WT and S172L mutant comparison as support for catalytic function of this enzyme in (S)-DAC cytotoxicity. While these results are clear, the author do not provide direct evidence for disruption of HSD17B11catalytic activity. For example, can the author demonstrate that HSD17B11 S172L lysates are catalytically deficient compared with WT counterparts? What are the effects of this mutation on androsterone levels in this overexpression system?

In the SDR superfamily, 2 amino acids are critical for catalysis: a serine and a tyrosine, which would be S172 and Y185 on HSD17B11 (a 3rd amino acid playing a key role is a lysine, which would be K189 on HSD17B11, allowing protonation/deprotonation of the catalytic tyrosine). The identification of a mutation of the key S172 residue in one of our DACR clones was therefore a very striking result and prompted us to use this mutation in the rest of our study. Initially, we were concerned by the fact that this mutation strongly impacted on HSD17B11 expression levels. However, expression in the form of a C-terminally FLAG-GFP tagged protein under the control of the CMV promoter rescued the expression levels of the mutant. Clones with similar expression of

HSD17B11 WT and S172L mutants were used for rescue experiments and for immunoprecipitations to perform in vitro assays. For these ones, to normalize for the amount of enzyme between the WT and S172L mutant, an excess of HSD17B11 compared to the beads’ capacity was used.

Thanks to Reviewer 1 and 3 comments, we now provide molecular docking experiment showing that the DAC (*S*)-**3** hydroxyl group is properly oriented in the HSD17B11 catalytic domain to form hydrogen bonds with S172 and Y185, while the DAC (*R*)-**3** cannot form such contacts. These additional data provide a rational for the enantiospecific bioactivation.

Finally, the inclusion of an additional plasmid coding for a mutation outside of the catalytic domain in the complementation assay is a recommended control given the protein instability issues of from mutations.

Thanks to Reviewer 3’s comment, we also tested other mutants outside of the catalytic domain in rescue experiments. Two mutations outside of the catalytic domain, *id* in the N-terminal ER/lipiddroplet localization sequence, were identified in the DACR clones: L14P and V16D. Thanks to the C-terminal FLAG-GFP tag and/or the CMV-based overexpression, we were able to complement U2OS inactivated for HSD17B11 with these two mutants to levels close to the S172L and WT HSD17B11-GFP (see the novel Figure 1—figure supplement 5C). Strikingly, both mutants successfully complemented U2OS (Figure 1—figure supplement 5D) supporting that these mutations conferred a resistance to DACs in HAP1 cells by reducing the level of HSD17B11 proteins rather than by affecting its catalytic activity. The fact that, under the same conditions, HSD17B11 S172L did not complement is a strong argument supporting that this mutation affects the catalytic activity of HSD17B11. Note that all the plasmids to express these mutants have been deposited on Addgene, which should benefit to this field of research.

In response to Reviewer 3 comments, these data are now included in the result section:

"CRISPR/Cas9-mediated inactivation of HSD17B11 also conferred significant (*S*)-**3** resistance to U2OS cells, which was suppressed by wild-type HSD17B11-GFP but not by the S172L mutant or GFP alone (Figure 1—figure supplement 5A,B). In contrast, complementation with HSD17B11 carrying the L14P or V16D mutations, lying outside of the catalytic domain and identified in the DACR clones #B1 and #A1/#A2, respectively, restored (*S*)-**3** cytotoxic activity, in agreement with these mutations affecting HSD17B11 protein stability and not its catalytic activity (Figure 1—figure supplement 5C,D). These data also support that the C-terminal FLAG-GFP tag and/or the CMV promoter-based overexpression partly overcome the impact of these mutations on HSD17B11 expression level."

2. To support the major critique above, the authors should show by LC-MS or NMR that purified HSD17B11 incubated with (S)-9 under the appropriate biochemical assay conditions produces the dialkynylketone (DACone) 10 product. The authors should compare with the (R)-9 substrate to demonstrate enantio-selectivity. These data should be included to support the claims associated with Figure 2F.

We agree with Reviewer 3 that this would be a direct demonstration of the enantioselective bioactivation of (*S*)-DACs into DACones. However, as explained above, the enzymatically produced DACone reacts with the nearby proteins including HSD17B11 itself when no other protein is available. We think the experiments provided in Figure 2F is the best demonstration that we can currently perform of the HSD17B11-dependent production of a protein reactive species. This is now clarified in the text:

“Considering that HSD17B11 known activity is the NAD+-dependent oxidation of a secondary carbinol into a ketone and that the only hydroxyl group on (*S*)-**3** is the one occurring in the dialkynylcarbinol pharmacophore, this experiment strongly supports the notion that HSD17B11 enantiospecifically oxidizes (*S*)-**3** into the DACone **7**, which immediately reacts with nearby proteins, including HSD17B11-GFP itself as observed in Figure 2F. This high level of reactivity unfortunately precludes isolating the HSD17B11-produced DACones.”

3. Authors should show western blots for HSD17B11 detection in Figure 2F since they report issues with production of the full length S172L protein due to protein instability.

The experiments comparing the rescue of DAC cytotoxicity by complementing HSD17B11 deficient cells with the WT, L14P, V16D and S172L HSD17B11 already provide some clarification regarding the different impact of the identified mutations on HSD17B11 catalytic activity.

In our in vitro experiments, we took great care to use similar amounts of HSD17B11 WT and S172L catalytic mutant by using excess of HSD17B11-GFP-containing extracts as compared to the bead’s binding capacity. As suggested by Reviewer 3, we also analyzed the amount of enzyme on the beads at the end of the reaction depicted in Figure 2F, altogether with the presence of DACmodified proteins using click-based detection with the AlexaFluor647-azido fluorophore. This confirmed that equivalent amounts of WT and S172L HSD17B11-GFP were used in the assay, but also revealed that HSD17B11-GFP itself is modified by DACs. This modification was decreased when betalactoglobuline was added to the assay, suggesting a competition between the two proteins for modification by the DACones. In addition, no modification was observed with the S172L mutant (or without NAD+, not shown here).

With the complementation experiments and the molecular docking, these data support that the S172L mutant blocked the production of protein-reactive DACones by directly affecting HSD17B11 catalytic activity. We thank the Reviewer 3 for requesting this analysis which strengthens our findings.